



Comparing modified substrate induced respiration with selective inhibition
(SIRIN) and N₂O isotope approaches to estimate fungal contribution to
denitrification in three arable soils under anoxic conditions
Lena Rohe[1, 2, 3], Traute-Heidi Anderson[2], Heinz Flessa[2], Anette Giesemann[2], Dominika Lewicka-
Szczebak[2, 4], Nicole Wrage-Mönnig[5], Reinhard Well[2]
[1]Helmholtz Centre for Environmental Research – UFZ, Department Soil System Sciences, Theodor-
Lieser Str. 4, Halle, Germany
[2]Thünen Institute of Climate Smart Agriculture, Bundesallee 65, Braunschweig, Germany
[3]University of Göttingen, Department of Crop Sciences, Institute of Grassland Science, von-Siebold-
Str. 8, 37075 Göttingen, Germany
[4]University of Göttingen, Centre for Stable Isotope Research and Analysis, Büsgenweg 2, 37077
Göttingen, Germany
[5]University of Rostock, Agricultural and Environmental Faculty, Grassland and Fodder Sciences,
Justus-Liebig-Weg 6, Rostock, Germany
*Correspondence to*: Lena Rohe (lena.rohe@ufz.de)
Keywords: selective growth inhibition, $^{15}$N site preference, fungal denitrification, $C_2H_2$, isotope
endmember mixing approach, $SP_{N2O}$ mixing balance, SP/$\delta^{18}$O mapping approach
**Abstract**
Pure culture studies provide evidence of the ability of soil fungi to produce nitrous oxide (N₂O) during
denitrification. Soil studies with selective inhibition indicated a possible dominance of fungal
compared to bacterial N₂O production in soil, which drew more attention to fungal denitrification.
Analyzing the isotopic composition of N₂O, especially the $^{15}$N site preference of N₂O produced
($SP_{N2O}$), showed that N₂O of pure bacterial or fungal cultures differed in $SP_{N2O}$ values, which might
enable the quantification of fungal N₂O based on the isotopic endmember signatures of N₂O produced
by fungi and bacteria.
This study aimed to identify the fungal contribution to N₂O emissions under anaerobic conditions in
incubated repacked soil samples by using different approaches to disentangle sources of N₂O. Three
soils were incubated under anaerobic conditions to promote denitrification with four treatments of a
modified substrate induced respiration with selective inhibition (SIRIN) approach. While one
treatment without microbial inhibition served as a control the other three treatments were amended
with inhibitors to selectively inhibit bacterial, fungal or bacterial and fungal growth. These treatments
were performed in three varieties. In one variety the $^{15}$N tracer technique was used to estimate the
effect of N₂O reduction on N₂O produced, while two other varieties were performed under natural
isotopic conditions but with and without acetylene. Three approaches were established to estimate the
N₂O production by a fungal community in soil: i) A modification of the SIRIN approach was used to





calculate $N_2O$ evolved from selected organism groups, and ii) $SP_{N2O}$ values from the acetylated
treatment were used in the isotope endmember mixing approach (IEM), and iii) the $SP/\delta^{18}O$ mapping
approach ($SP/\delta^{18}O$ Map) was used to estimate the fungal contribution to $N_2O$ production and $N_2O$
reduction under anaerobic conditions from the non-acetylated treatment.
The three approaches tested revealed a small fungal contribution to $N_2O$ fluxes under anaerobic
conditions in the soils tested. Quantifying the fungal fraction with modified SIRIN was only possible
in one soil and totaled $0.28\pm0.09$. This was higher than the results obtained by IEM and $SP/\delta^{18}O$ Map,
which accounted zero to 0.20 of $N_2O$ produced to the fungal community.
To our knowledge, this study was the first attempt to quantify the fungal contribution to anaerobic $N_2O$
production by simultaneous application of three approaches, i.e. modified SIRIN, IEM and $SP/\delta^{18}O$
Map. While all methods coincided by suggesting a small or missing fungal contribution, further
studies under conditions ensuring larger fungal $N_2O$ fluxes and including alternative inhibitors are
needed to better cross-validate the methods.
## 1. Introduction
The greenhouse gas nitrous oxide ($N_2O$) contributes to global warming and to the depletion of the
ozone layer in the stratosphere (Crutzen, 1970; IPCC, 2013). The largest anthropogenic $N_2O$ emissions
originate from agricultural soils and are mainly produced during microbial nitrification, nitrifier
denitrification and denitrification (Firestone and Davidson, 1989; Bremner, 1997; IPCC, 2013; Wrage-
Mönnig et al., 2018). In order to find mitigation strategies for $N_2O$ emissions from arable soils, it is
important to understand $N_2O$ sources and sinks and thus improve knowledge about the production
pathways and the microorganisms involved.
For a long time, it was believed that solely bacteria are involved in $N_2O$ formation during
denitrification (Firestone and Davidson, 1989); however, also several fungi are capable of
denitrification (Bollag and Tung, 1972; Shoun et al., 1992). Denitrification describes the reduction of
nitrate ($NO_3^-$) to dinitrogen ($N_2$), with the intermediates nitrite ($NO_2^-$), nitric oxide (NO) and $N_2O$
(Knowles, 1982). While this entire reaction chain including the ability to reduce $N_2O$ to $N_2$ is found
among bacterial denitrifiers, most fungi lack $N_2O$ reductase (Nos) (Shoun et al., 1992; Shoun et al.,
2012; Higgins et al., 2018). Recently, pure culture studies showed that $N_2O$ from fungal denitrification
was often accompanied with $N_2O$ from abiotic production (Phillips et al., 2016a; Phillips et al.,
2016b), which may lead to overestimate the importance of fungal $N_2O$ production. Other studies
indicated that only some fungal species (e.g. *Fusarium* strains) performing respiratory denitrification
with substantial amounts of $N_2O$ production (Higgins et al., 2018; Keuschnig et al., 2020). Even
though only a few fungal species were identified to be capable of respiratory denitrification, $N_2O$
produced by fungi may contribute largely to $N_2O$ from denitrification in soil. Firstly, fungi dominate
the biomass in soil (up to 96%) compared to bacteria in general and thus fungi could potentially play a
dominant role in $N_2O$ production (Ruzicka et al., 2000; Braker and Conrad, 2011). Thus, a respiratory





fungal-to-bacterial (F:B) ratio of 4 is typical for arable soils (Anderson and Domsch, 1975;
Blagodatskaya and Anderson, 1998). Secondly, the fact that $N_2O$ is the major end product of fungal
denitrification led to the assumption that the potential activity of fungal $N_2O$ production in soil may
exceed that of bacteria, provided that both microbial groups have the same specific denitrification
activity (Shoun et al., 1992; Sutka et al., 2008). Thirdly, co-denitrification was found to often co-occur
with fungal denitrification (Shoun and Tanimoto, 1991; Tanimoto et al., 1992). During this fungal
pathway, a hybrid $N_2O$ is formed using one N atom from $NO_2^-$ and one N atom from compounds like
azide or ammonium ($NH_4^+$) for $N_2O$ production (Tanimoto et al., 1992; Shoun et al., 1992; Rohe et al.,
2017; Spott et al., 2011). A $^{15}N$ tracing approach was used to identify and quantify co-denitrification,
which contributed about 92% to $N_2O$ produced in an incubation experiment with a grassland soil under
anaerobic conditions (Laughlin and Stevens, 2002). This again stresses the large potential $N_2O$
production by fungi. However, in pure culture studies not only co-denitrification, but also abiotic $N_2O$
formation may co-occur with fungal denitrification (Phillips et al., 2016a; Phillips et al., 2016b; Rohe
et al., 2017) and pathway differentiation is still challenging.
Soil incubation experiments could serve to differentiate between $N_2O$ produced by fungi and bacteria
during denitrification by the application of two antibiotics: streptomycin and cycloheximide, which
inhibit bacterial or fungal growth, respectively, by inhibition of the protein biosynthesis. This method
is known as substrate induced respiration with selective inhibition (SIRIN) (Anderson and Domsch,
1975; Laughlin and Stevens, 2002; Crenshaw et al., 2008; Blagodatskaya et al., 2010; Long et al.,
2013). A few studies used a modification of this method for $N_2O$ analysis and found a greater decrease
of $N_2O$ production with fungal than with bacterial growth inhibition (e.g. 89 vs. 23% decrease
(Laughlin and Stevens, 2002)), indicating that fungi might dominate $N_2O$ production (Laughlin and
Stevens, 2002; McLain and Martens, 2006; Crenshaw et al., 2008; Blagodatskaya et al., 2010; Long et
al., 2013; Chen et al., 2014; Chen et al., 2015).
Analysing the isotopic composition of $N_2O$ might be a promising tool to distinguish between $N_2O$
from bacterial and fungal denitrification and other pathways. Especially, the isotopomer ratios of $N_2O$
(i.e. $N_2O$ molecules with the same bulk $^{15}N$ isotopic enrichment but showing different positions of $^{15}N$
in the linear $N_2O$ molecule (Ostrom and Ostrom, 2017)) in pure culture studies showed differences in
$N_2O$ of bacterial and fungal denitrification (Sutka et al., 2006; Sutka et al., 2008; Frame and Casciotti,
2010; Rohe et al., 2014a; Rohe et al., 2017) and might be suitable for distinguishing between $N_2O$
produced by bacteria or fungi under denitrifying conditions. Isotopomer ratios of $N_2O$ can be
expressed as $^{15}N$ site preference ($SP_{N2O}$), i.e. the difference between $\delta^{15}N$ of the central and terminal N-
position of the asymmetric $N_2O$ molecule (Toyoda and Yoshida, 1999). The $SP_{N2O}$ values of $N_2O$ of six
pure fungal cultures was between 16 and 37 ‰ (Sutka et al., 2008; Rohe et al., 2014a; Maeda et al.,
2015; Rohe et al., 2017), whereas several bacteria produced $N_2O$ with $SP_{N2O}$ values between -7.5 and
+3.5 ‰ during denitrification (Toyoda et al., 2005; Sutka et al., 2006; Rohe et al., 2017). However, the
$SP_{N2O}$ value of $N_2O$ produced by pure bacterial cultures during nitrification is approximately 33 ‰ and





interferes with $SP_{N2O}$ values of fungal denitrification (Sutka et al., 2006; Sutka et al., 2008; Rohe et al.,
2014a). This demonstrates the difficulty to use $SP_{N2O}$ values as an indicator for different organism
groups contributing to $N_2O$ production from soil, where different pathways may co-occur. Although
$SP_{N2O}$ values are independent of isotopic signatures of the precursors, $\delta^{15}N$ and $\delta^{18}O$ values of
produced $N_2O$ ($\delta^{15}N^{bulk}_{N2O}$ and $\delta^{18}O_{N2O}$, respectively) result from the isotopic signature of the
precursor and isotopic fractionation during $N_2O$ production (Toyoda et al., 2005; Frame and Casciotti,
2010). Interpretation of $\delta^{18}O_{N2O}$ values is even more complex, because O exchange during
denitrification between water and denitrification intermediates alters the final $\delta^{18}O_{N2O}$ value (Garber
and Hollocher, 1982; Aerssens et al., 1986; Kool et al., 2007; Rohe et al., 2014b; Rohe et al., 2017).
However, recently fungal and bacterial $N_2O$ showed different ranges for $\delta^{18}O_{N2O}$ values and this
isotopic signature may also be helpful in differentiation of these pathways (Lewicka-Szczebak et al.,
2016). Moreover, $\delta^{15}N^{bulk}_{N2O}$, $\delta^{18}O_{N2O}$ and $SP_{N2O}$ values are in the course of denitrification affected by
isotopic fractionation due to $N_2O$ reduction. During $N_2O$ reduction, the $^{14}N^{16}O$ bond is preferentially
broken compared to $^{14}N^{18}O$ or $^{15}N^{16}O$, resulting in residual $N_2O$, that is relatively isotopically enriched
in $^{15}N$ and $^{18}O$ and shows larger $SP_{N2O}$ values compared to $SP_{N2O}$ values of $N_2O$ from denitrification
without the reduction step (Popp et al., 2002; Ostrom et al., 2007). Quantification of $N_2O$ reduction to
$N_2$ during denitrification is possible by analyzing $^{15}N_2$ fluxes in $^{15}N$ tracing experiments using $^{15}N$
enriched substrates (Well et al., 2006; Lewicka-Szczebak et al., 2014). To quantify $N_2O$ reduction and
the pathways producing $N_2O$ based on $N_2O$ isotopocules (i.e. $N_2O$ with differing number or positions
of N or O isotopes (Ostrom and Ostrom, 2017)), the isotope mapping approach was developed using
isotope fractionation factors together with $\delta^{15}N^{bulk}$ values of $N_2O$ precursors ($\delta^{15}N_{NOx}$) as well as
$\delta^{15}N^{bulk}_{N2O}$ and $SP_{N2O}$ values of $N_2O$ produced (Toyoda et al., 2011). Recently, this isotope mapping
approach was further developed (SP/$\delta^{18}O$ Map) using $\delta^{18}O_{N2O}$ and $SP_{N2O}$ values of $N_2O$ and $\delta^{18}O$
values of precursors (Lewicka-Szczebak et al., 2014; Lewicka-Szczebak et al., 2017). This approach
uses different slopes of $N_2O$ reduction and mixing lines in the $\delta^{18}O$ – SP isotope plot and allows for
differentiation of isotope effects due to $N_2O$ reduction and admixture of fungal $N_2O$.
Based on the above cited ranges for the isotopomer endmembers of fungal and bacterial
denitrification, and assuming that only fungi and bacteria are responsible for $N_2O$ production the
fraction of fungal $N_2O$ can be calculated using the isotope endmember mixing approach (IEM) with
$SP_{N2O}$ values of $N_2O$ produced in soil ($SP_{prod}$), provided $N_2O$ reduction, which is altering $SP_{N2O}$ values
of emitted $N_2O$, does not occur (Ostrom et al., 2010; Ostrom and Ostrom, 2011). This can be ensured
in laboratory experiments by inhibiting $N_2O$ reduction to $N_2$ using acetylene ($C_2H_2$) during anaerobic
incubation experiments (Yoshinari and Knowles, 1976; Groffman et al., 2006; Well and Flessa, 2009;
Nadeem et al., 2013). Hence, $C_2H_2$ inhibition might be suitable to quantify $SP_{prod}$ values in soils
exhibiting significant $N_2O$ reduction and would thus allow quantification of fungal $N_2O$ fluxes based
on $SP_{prod}$ values. For the SP/$\delta^{18}O$ Map, the inhibition of $N_2O$ reduction is not needed. Hence, $N_2O$
reduction can be estimated together with the $N_2O$ mixing due to application of two isotopic signatures



of $N_2O$. While it is generally assumed that $SP_{prod}$ values of $N_2O$ produced by fungal pure cultures is
transferable to $N_2O$ produced by fungal soil communities, this has not yet been proven. Until now,
studies reporting possible ranges of fungal contributions to $N_2O$ fluxes from soil were based on $SP_{prod}$
values of pure cultures (Köster et al., 2013b; Zou et al., 2014; Lewicka-Szczebak et al., 2017;
Senbayram et al., 2018; Senbayram et al., 2020; Lewicka-Szczebak et al., 2014), but uncertainty of
this approach arose from the fact that the full range of $SP_{prod}$ values is between 16 and 37‰ (Sutka et
al., 2008; Maeda et al., 2015; Rohe et al., 2017). It would thus be useful to constrain fungal $SP_{prod}$
values for a specific soil or soil type.
So far, the described methods for distinguishing between fungal and bacterial $N_2O$ emission have not
been evaluated and compared in the same soil and their accuracy and possible bias remains unknown.
Therefore, this study aims at (i) determining the fungal contribution on $N_2O$ production by
denitrification under anoxic conditions and glucose addition using three arable soils and approaches:
modified SIRIN, IEM and the SP/$\delta^{18}O$ Map, (ii) to compare the fungal contribution on $N_2O$ production
determined by these approaches and thus assess factors of potential bias of the methods, and (iii) to
estimate the $SP_{N2O}$ values from a fungal soil community and thus to evaluate the transferability of the
pure culture range of the fungal $SP_{N2O}$ endmember values.
**2.  Materials and Methods**
2.1 Soil samples
All experiments were conducted with three arable soils differing in texture to provide different
conditions for denitrification. As one soil was sampled at two different time points, we conducted four
experiments: Experiment 1 with loamy sand sampled in December 2012, Experiment 2 with sand
sampled in January 2013, Experiment 3 with silt loam sampled in December 2012, and Experiment 4
with loamy sand sampled in June 2011.
Soil samples of the upper 30 cm were collected in plastic bags aerated via cotton wool stoppers and
stored at 6 °C for maximally two months. To get information about the initial soil status, total contents
of C and N in soil samples were analyzed by dry combustion of grinded samples (LECO TruSpec,
Germany). The soil pH was measured in 0.01 M $CaCl_2$. The mineral nitrogen content (Nmin) of soil
samples was determined before and after fertilization by extracting $NO_3^-$ and $NH_4^+$ with 0.01 M
calcium chloride dihydrate ($CaCl_2 \cdot 2 H_2O$) according to ISO 14255 and analyzing $NO_3^-$ and $NH_4^+$
concentrations in the extracts with a Continuous-Flow-Analyzer (SKALAR, Germany). The $\delta^{15}N$ and
$\delta^{18}O$ values of $NO_3^-$ and $NO_2^-$ ($\delta^{15}N_{NOx}$ and $\delta^{18}O_{NOx}$, respectively) in soil extracts (with 0.01 M calcium
chloride dihydrate ($CaCl_2 \cdot 2 H_2O$)) were analyzed by the bacterial denitrifier method (Casciotti et al.,
2002). Respiratory biomass of the three soils was analyzed with substrate induced respiration (SIR)
according to Anderson and Domsch (1978) and the respiratory F:B ratio was analyzed with substrate
induced respiration with selective inhibition (SIRIN) in summer 2010 by a computer-generated
selectivity analysis: "SIR-SBA 4.00" (Heinemeyer, copyright MasCo Analytik, Hildesheim, Germany)





(Anderson and Domsch, 1975). The scheme of glucose and growth inhibitor combinations is listed
below in section "Methodological approach". For further characteristics of the soils, see Table 1.
2.2 Methodological approach
2.2.1 SIRIN pre-experiment
As in most studies applying the SIRIN method on $N_2O$ emissions (e. g. Laughlin and Stevens, 2002;
Chen et al., 2014; Ladan and Jacinthe, 2016), a pre-experiment was conducted, in order to get
information about optimal substrate and inhibitor concentrations for substrate induced growth
inhibition. The SIR method (Anderson and Domsch, 1978) was used to get information about the
amount of respiratory biomass in soil. To this end, we added different concentrations of glucose (0.50,
0.75, 1.0, 1.5, 2.0, 3.0, 4.0, 5.0, 6.0 mg $g^{-1}$ dry weight (dw) soil) to find the optimal glucose
concentration ($c_{opt}$(glucose)), which is the glucose concentration that causes maximum initial
respiration rates (Anderson and Domsch, 1978). $C_{opt}$(glucose)) was 1.0 mg $g^{-1}$ for Experiment 2 (sand)
and 1.5 mg $g^{-1}$ for Experiments 1, 3 and 4 (loamy sand and silt loam). Glucose served as substrate to
initiate microbial growth (Anderson and Domsch, 1975).
We conducted SIRIN for determining the respiratory F:B ratio according to Anderson and Domsch
(1975). Selectivity of the inhibitor combinations of streptomycin (bacterial respiratory inhibitor) and
cycloheximide (fungal respiratory inhibitor) were tested with the following concentrations, 0.75, 1.0,
1.5 mg $g^{-1}$ dw, respectively. The optimal concentration for inhibition of fungal respiration was 0.75 mg
$g^{-1}$ dw soil cycloheximide ($c_{opt}$(cycloheximide)) and for bacterial respiratory inhibition 1.0 mg $g^{-1}$ dw
soil streptomycin ($c_{opt}$(streptomycin)).
**Table 1: Soil characteristics of three arable soils from Germany used for incubation experiments (Exp.)**
**(standard deviation in brackets).**

| Exp. (Year) | Soil texture | Soil type (WRB) | Location | C content [%] | N content [%] | $NH_4^+$ [mg N $L^{-1}$] | $NO_3^-$ [mg N $L^{-1}$] | pH ($CaCl_2$) | $\delta^{15}N_{NOx}$ [‰][e] | $\delta^{18}O_{N \; Ox}$ [‰][e] | F:B[f] | Biomass[g] [µg C $gdw^{-1}$ soil] |
|---|---|---|---|---|---|---|---|---|---|---|---|---|
| 1 (2012) 4 (2011) | Loamy sand | Haplic Luvisol | Braun-schweig[a] | 1.43 (<0.01) | 0.10 (<0.01) | 0.04 | 1.25 | 5.67 | 3.98 | -4.82 | 2.6 | 234 |
| 2 (2013) | Sand | Gleyic Podzol | Wenne-bostel[b] | 2.31 (0.04) | 0.14 (<0.01) | 0.02 | 0.56 | 5.54 | 0.73 | -2.68 | 2.6 | 161 |
| 3 (2013) | Silt loam | Haplic Luvisol | Götting-en[c] | 1.62 (0.02) | 0.13 (<0.01) | n.d.[d] | 2.05 | 7.38 | 4.18 | 2.32 | 4.9 | 389 |

[a]**Experimental Station of the Friedrich-Löffler Institute, Braunschweig, Germany**
[b]**private agricultural field North of Hannover, water protection area Fuhrberger Feld, Germany**
[c]**Reinshof Experimental Farm, Georg-August-University, Göttingen, Germany**
[d]**not detectable (i.e. below detection limit of 0.005 mg $L^{-1}$ $NH_4^+$-N)**
[e]**Isotopic values of natural soil $NO_3^-$ using the denitrifier method (Casciotti et al., 2002).**
[f]**Respiratory fungal-to-bacterial (F:B) ratio analyzed by SIRIN method (Anderson and Domsch, 1973,**
**1975)**
[g]**Respiratory biomass analyzed by $CO_2$ production from SIR method.**





2.2.2 Soil incubation with selective inhibition to determine $N_2O$ forming processes
The experimental design included two factors, (i.) microbial inhibition by fungal and/or bacterial
inhibitors and (ii.) activity of $N_2O$ reductase analyzed either by inhibition with $C_2H_2$ or quantification
by [15]N tracing. To address factor (i.), the SIRIN method for determination of the respiratory F:B ratio
based on $CO_2$ emission was modified to determine $N_2O$ production by microbial groups. However, in
contrast to previous studies by Laughlin and Stevens (2002), McLain and Martens (2006),
Blagodatskaya et al. (2010) and Long et al. (2013), we did not pre-incubate the soil with the growth
inhibitors, as this could result in changes of the microbial community (e.g. preferential growth of
selected organisms). We intended to disturb microbial communities as little as possible.
The soil was sieved (2 mm) and pre-incubated at 22 °C for five to seven days in the dark with cotton
wool stoppers to allow respiration and aerobic conditions in soil bags. Four microbial inhibitor
treatments (each in triplicate) with $c_{opt}$(glucose) for each soil were established:
A        Control, without growth inhibitors
B        With streptomycin sulfate ($C_{42}H_{84}N_{14}O_{36}S_3$) to inhibit bacterial growth
C        With cycloheximide ($C_{15}H_{23}NO_4$) to inhibit fungal growth
D        With streptomycin and cycloheximide, to inhibit bacterial and fungal growth
To address factor (ii.), all microbial inhibitor treatments were conducted in three $N_2O$ reductase
varieties, i.e.: with [15]N-$NO_3$ fertilizer (variety "*traced*") to quantify $N_2O$ reduction to $N_2$, with natural
abundance $NO_3^-$ and 10 kPa $C_2H_2$ in the headspace (variety "*+$C_2H_2$*") to block $N_2O$ reductase, and
with natural abundance $NO_3^-$ but without blocking $N_2O$ reductase, i.e. no $C_2H_2$ added (variety "*-
$C_2H_2$*"). In total, there were 48 experimental treatments and 144 vessels (four Experiments with four
inhibitor treatments (A, B, C, D) and three varieties (*traced, +$C_2H_2$* and *-$C_2H_2$,*) each in triplicates).
The soil was adjusted to 80% water filled pore space (WFPS) with distilled water and simultaneously
fertilized with $NO_3^-$ (varieties *-$C_2H_2$* and *+$C_2H_2$* with 50 mg N kg$^{-1}$ $KNO_3$ in Experiment 1, 2 and 3 and
with 60 mg N kg$^{-1}$ $NaNO_3$ in Experiment 4 and *traced* variety with 50 mg N kg$^{-1}$ [15]N-$KNO_3$ in
Experiment 1, 2 and 3 and 60 mg N kg$^{-1}$ [15]N-$KNO_3$ in Experiment 4 with a [15]N-labeling of 50 atom%
(at%)). For each treatment, we incubated 100 g dw soil in 850 mL preserving jars (J. WECK GmbH u.
Co KG, Wehr, Germany) with gas inlet and outlet equipped with three port luer lock plastic stopcocks
(Braun, Melsungen, Germany). According to the original SIRIN method (Anderson and Domsch,
1973, 1978) and a mixture of $c_{opt}$(glucose) and carrier material talcum (5 mg talcum g dw$^{-1}$) was added
to soil of treatment A and together with the growth inhibitors to the soil of treatments B, C and D. The
soil and additives of each treatment were mixed for 90 seconds with a handheld electric mixer. During
packing, the soil density was adjusted to a target soil density of 1.6 g cm$^{-3}$ in Experiment 1, 2 and 4
and of 1.3 g cm$^{-3}$ in Experiment 3. To ultimately achieve denitrifying conditions in all treatments and
to avoid catalytic NO decomposition in the *+$C_2H_2$* variety (Nadeem et al., 2013), the headspace of the
closed jars was flushed with $N_2$ to exchange the headspace 10 times. Directly following, 85 mL of the



gas in the headspace in variety *+C₂H₂* were exchanged by pure $C_2H_2$ resulting in 10 kPa $C_2H_2$ in the
headspace. The manual sample collection of 14 mL gas in duplicates with a plastic syringe was
performed after six, eight and ten hours (Experiment 1, 2 and 3) or two, four and eight (Experiment 4)
of incubation time, respectively. The removed gas was replaced by the same amount of $N_2$.

### 2.3 Gas analysis

Gas samples were analyzed for $N_2O$ and $CO_2$ concentrations (*c(N₂O)* and *c(CO₂)*) with gas
chromatography (GC, Agilent 7890A, Agilent, Böblingen, Germany). The detection limit of $N_2O$ was
0.04 ng N h⁻¹ with a measurement precision of 1% and for $CO_2$ the detection limit was 4 ng C h⁻¹ with
a measurement precision of 0.5%. As a control, $N_2$ and $O_2$ concentrations in the samples were analyzed
with GC to ensure anaerobic conditions during the incubation for $N_2O$ production from denitrification.
The $N_2O$ isotopic analysis of the gas samples of varieties *-C₂H₂* and *+C₂H₂* were performed on a pre-
concentrator (PreCon, Thermo–Finnigan, Bremen, Germany) interfaced with a GC (Trace Gas Ultra,
Thermo Scientific, Bremen, Germany) and analyzed by isotope ratio mass spectrometry (IRMS, Delta
V, Thermo Fisher Scientific, Bremen, Germany) (Brand, 1995; Toyoda and Yoshida, 1999; Köster et
al., 2013b). The analytical precision was 0.1 ‰, 0.2 ‰ and 1.5 ‰ for $\delta^{15}N^{bulk}_{N2O}$, $\delta^{18}O_{N2O}$ and $SP_{N2O}$
values, respectively.
The gas samples of variety *traced* from Experiment 1, 2, and 3 were analyzed for the 29/28 and 30/28
ratios of $N_2$ according to Lewicka-Szczebak et al. (2013) using a modified GasBench II preparation
system coupled to  IRMS (MAT 253, Thermo Scientific, Bremen, Germany). The gas samples of
variety *traced* from Experiment 4 were analyzed at the Centre for Stable Isotope Research and
Analysis (University of Göttingen, Germany). The $N_2$ produced was analyzed using an elemental
analyzer (Carlo Erba ANA 1500) that was coupled to dual inlet IRMS (Finnigan MAT 251) (Well et
al., 1998; Well et al., 2006). Isotopic values of $N_2O$ of Experiment 4 (variety *traced*) were analyzed in
the same lab using a pre-concentration unit coupled to IRMS (Precon-DeltaXP, Thermo Scientific,
Bremen, Germany) (Well et al., 2006). Isotope ratios were used applying the non-random distribution
approach to calculate the fraction of $N_2$ and $N_2O$ originating from the ¹⁵N-labelled N pool as well as
the ¹⁵N enrichment of that N pool (*aₚ*) (Bergsma et al., 2001; Spott et al., 2006).

### 2.4 Inhibitor effects

For interpretation of $N_2O$ or $CO_2$ production, the validity of the experimental results with respect to
fungal and bacterial $N_2O$ fluxes was checked using a flux balance comparing the sum of bacterial and
fungal inhibition effects (treatments B and C) to the dual inhibition effect (treatment D):
$$D = A - [(A - B) + (A - C)] \qquad \text{(Eq. 1)}$$
With *A, B, C* and *D* representing the $N_2O$ production rates of the last sampling time of treatment *A, B,*
*C* and *D*, respectively. Assuming that in the other three treatments (A, B and C) non-inhibitable $N_2O$





production was equal to treatment D, N$_2$O produced by bacteria or fungi should show the following
relation between the four treatments:
$(A - D) = (B - D) + (C - D)$               (Eq. 2)
The fungal contribution to N$_2$O production during denitrification with microbial inhibition ($F_{FDmi}$) can
be calculated, when N$_2$O production of treatment D is significantly smaller than N$_2$O production of
treatments A, B and C by:
$F_{FDmi} = \frac{(A-C)}{(A-D)}$                (Eq. 3)
2.5 Isotope methods
2.5.1 Isotope endmember mixing approach (IEM)
The fungal fraction ($F_{FD}$) contributing to N$_2$O production from denitrification in soil samples was
calculated according to the isotope mixing model (IEM) proposed by Ostrom et al. (2010), which was
established for calculating the bacterial fraction ($F_{BD}$) of N$_2$O production. Assuming that bacteria ($BD$)
and fungi ($FD$) are the only microorganisms responsible for denitrification in soil, the $^{15}$N site
preference values of produced N$_2$O ($SP_{prod}$) results from the $SP_{N2O}$ mixing balance:
$SP_{prod} = F_{FD} * SP_{FD} + F_{BD} * SP_{BD}$          (Eq. 4)
where $F_{FD}$ and $F_{BD}$ represent the fraction of N$_2$O produced by fungi and other N$_2$O sources than fungal
denitrification, respectively, and $SP_{FD}$ and $SP_{BD}$ are the respective $SP_{N2O}$ endmember values (Ostrom et
al., 2010; Ostrom and Ostrom, 2011). This calculation was based on the assumption that the sum of
$F_{BD}$ and $F_{FD}$ equals 1 and that N$_2$O reduction to N$_2$ is negligible. The mean $SP_{FD}$ value was assumed to
be 33.6 ‰ (Sutka et al., 2008; Maeda et al., 2015; Rohe et al., 2014a; Rohe et al., 2017) and the $SP_{BD}$
value from heterotrophic denitrification was assumed with minimum and maximum values from -7.5
to +3.7 ‰ (Yu et al., 2020). For this IEM approach, only results from variety +$C_2H_2$ could be used to
calculate the fungal fraction contributing to N$_2$O production ($F_{FD\_SP}$), as microorganisms of this variety
produce N$_2$O that is not affected by reduction to N$_2$. The $F_{FD\_SP}$ contributing to N$_2$O production during
denitrification was calculated from the measured $SP_{N2O}$ value from treatment A of variety +$C_2H_2$ as
$SP_{prod}$ value (Eq. 4). In case successful inhibition (modified SIRIN approach), Eq. 4 was solved for the
$SP_{FD}$ value using $F_{FD}$, $F_{BD}$, and $SP_{prod}$ values of the respective variety.
2.5.2 SP/$\delta^{18}$O isotope mapping approach (SP/$\delta^{18}$O Map)
The $F_{FD}$ contributing to N$_2$O production from denitrification in soil samples was also estimated with
the SP/$\delta^{18}$O Map ($F_{FD\_MAP}$) (Lewicka-Szczebak et al., 2017; Lewicka-Szczebak et al., 2020). This
method allows for estimation of both: the $F_{FD}$ and N$_2$O product ratio [N$_2$O/(N$_2$+N$_2$O)] (*product*
*ratio$_{Map}$*). For precise estimations, the $\delta^{18}$O values of soil water ($\delta^{18}$O$_{H2O}$) applied in the experiments
are needed and these values were not determined. However, since we have independent information on





the N$_2$O product ratio from the *traced* variety, we can calculate the possible $\delta^{18}O_{H2O}$ values of soil to
get the nearest N$_2$O product ratios in natural and $^{15}$N treatments. The fitting of values was performed
for mean, minimal und maximal values of $SP_{BD}$ (-1.9, -7.5 and 3.7‰, respectively) and aimed at
obtaining the minimal difference between *product ratio$_{Map}$* and measured in *traced* variety, *i.e.*, the
minimal value of (*product ratio$_{15N}$ - product ratio$_{Map}$*)$^2$ for -$C_2H_2$ and +$C_2H_2$ variety (for explanation of
the product ratio see next section). This further allows obtaining the possible ranges for $F_{FD}$ for
particular fitted values (Table 4). The calculations with this approach may be performed assuming two
different scenarios of the interplay between N$_2$O mixing and reduction (Lewicka-Szczebak et al.,
2017; Lewicka-Szczebak et al., 2020) but for this study both scenarios yield almost identical results
(maximal difference of 0.02 in N$_2$O product ratio and $F_{FD}$ was found), due to $F_{BD}$ near 1. Hence, we
only provide the results assuming the reduction of bacterial N$_2$O followed by mixing with fungal N$_2$O.
2.5.3 Product ratio [N$_2$O/(N$_2$+N$_2$O)] of denitrification
The variety *traced* served to assess N$_2$O reduction during denitrification in each experiment. The
product ratio of denitrification [N$_2$O/(N$_2$+N$_2$O)] as given by the variety *traced (product ratio$_{15N}$)* was
calculated as:
$product\ ratio_{15N} = \frac{^{15}N_{N2O}}{^{15}N_{N2} + ^{15}N_{N2O}}$            (Eq. 5)
with $^{15}N_{N2O}$ and $^{15}N_{N2}$ representing N$_2$O and N$_2$ produced in the $^{15}$N-labeled fertilizer pool. To check
the effectiveness of C$_2$H$_2$ to block the N$_2$O reduction, *product ratio$_{15N}$* was compared with *product*
*ratio$_{C2H2}$*, where the latter can be calculated from N$_2$O production rates of varieties -$C_2H_2$ and +$C_2H_2$:
$product\ ratio_{C2H2} = \frac{N_2O_{-C2H2}}{N_2O_{+C2H2}}$            (Eq. 6)
with $N_2O_{-C2H2}$ and $N_2O_{+C2H2}$ representing the N$_2$O produced in varieties -$C_2H_2$ and +$C_2H_2$, respectively.
If *product ratio$_{15N}$* and *product ratio$_{C2H2}$* were in agreement, a complete blockage of N$_2$O reduction
could be assumed. This enabled to estimate reduction effects on the isotopic signatures of N$_2$O by
comparing the isotopic values of N$_2$O produced without N$_2$O reduction effects of variety +$C_2H_2$ (*δ0*
values) with isotopic values of N$_2$O of variety -$C_2H_2$.
The information on the product ratio was used as an additional possibility to calculate the $F_{FD}$ also for
variety -$C_2H_2$. First, the Rayleigh-type model presented by Lewicka-Szczebak et al. (2017) and
Senbayram et al. (2018) for similar closed-system incubations, the $^{15}$N site preference values of
produced N$_2$O, i.e. without its reduction to N$_2$O ($SP_{prod}$), of variety -$C_2H_2$ was calculated by correcting
SP values of emitted N$_2$O, i.e. after partial reduction of produced N$_2$O ($SP_{N2O-r}$) from variety -$C_2H_2$
with the net isotope effect of N$_2$O reduction ($\eta r$) and the *product ratio$_{15N}$* as follows:
$SP_{prod} = SP_{N2O-r} + \eta r \ln(product\ ratio_{15N})$            (Eq. 7)
According to (Yu et al., 2020) the $\eta r$ was assumed to be -6‰. Secondly, Eq.4 was used to calculate the
$F_{FD}$ by using $SP_{prod}$ values of variety –$C_2H_2$ ($F_{FD\_SPcalc}$) obtained from Eq. 7



### 2.6 Sources of N₂O produced

Assuming that denitrification is the only process producing $N_2O$ in the incubation experiment, the expected $^{15}N$ enrichment in $N_2O$ produced ($^{15}N_{N2O\_exp}$) was given by

$$^{15}N_{N2O\_exp}\ [at\%] = \frac{(N_{soil}\ x\ ^{15}N_{nat}) + (N_{fert}\ x\ ^{15}N_{fert})}{N^{bulk}}$$     (Eq. 8)

with $N_{soil}$, $N_{fert}$ and $N^{bulk}$ describing the amount of N [mg] in unfertilized soil samples, fertilizer and fertilized soil samples, respectively and $^{15}N_{nat}$ and $^{15}N_{fert}$ is standing for $^{15}N$ enrichment under natural conditions (0.3663 at%) and in fertilizer (50 at%), respectively. Comparison of measured $^{15}N$ enrichment in $N_2O$ and $^{15}N_{N2O\_exp}$ gave information about the contribution of processes other than denitrification to $N_2O$ production.

### 2.7 Statistical Analysis

We conducted several three-way analyses of variance (ANOVA) to test significant effects of soil, experimental variety and treatment on $N_2O$ production, $CO_2$ production, and $SP_{N2O}$, $\delta^{15}N^{bulk}_{N2O}$ and $\delta^{18}O_{N2O}$ values. The pairwise comparison with Tukey's HSD test was made to find differences between soils, varieties and treatments influencing $N_2O$ production, $CO_2$ production, and isotopic values. Significant effects of soils and treatments on *product ratio*$_{C2H2}$ and *product ratio*$_{15N}$ were tested by two-way ANOVA, while differences between soils and treatments influencing the product ratios were tested with pairwise comparison with Tukey's HSD test. Effects of varieties *-C₂H₂* and *traced* on $N_2O$ and $CO_2$ production were tested by ANOVA. For this ANOVA, the $N_2O$ production rate had to be $\log_{10}$-transformed to achieve homogeneity of variance and normality. The significance level α was 0.1 for every ANOVA. For some ANOVAs treatments were excluded, when replicates were n < 3. The $N_2O$ or $CO_2$ production rates of variety *+C₂H₂* were followed over three sampling times by regression. For statistical analysis, we used the program R (R Core Team, 2013). Excel Solver tool was used to determine the $\delta^{18}O_{H2O}$ values in the application of SP/$\delta^{18}$O Map calculations.

## 3. Results

### 3.1 N₂O production rates

$N_2O$ and $CO_2$ production rates of all treatments were similar in magnitude in almost all cases and mostly indistinguishable (Table 2). $CO_2$ production rates were determined to get additionally information about the denitrifying process. $N_2O$ production rates exhibited increasing trends with ongoing incubation time for every soil with large variations within the treatments. Contrary to that, $CO_2$ production rates showed decreasing trends (Figure 1, exemplarily shown for data of variety *+C₂H₂*). Calculations of inhibitor effects were based on average $N_2O$ and $CO_2$ production rates of the entire incubation period, i.e. 10 hours of incubation time for Experiment 1, 2 and 3 and 8 hours for Experiment 4.



$N_2O$ and $CO_2$ production rates of all $+C_2H_2$ varieties differed significantly among soils ($P < 0.001$) and
$N_2O$ production rates differed also significantly among treatments ($P < 0.001$). Largest $N_2O$
production rate about 5.5 to 6.1 µg N $kg^{-1}h^{-1}$ was obtained in Experiment 1 and 3, while in Experiment
2 and 4 $N_2O$ production rates were lower (2.6 and 2.7 µg N $kg^{-1}h^{-1}$, respectively). $N_2O$ and $CO_2$
production rates were significantly larger in variety $+C_2H_2$ than in variety $-C_2H_2$ of Experiment 1, 3
and 4 ($P = 0.002$, $P < 0.010$ and $P < 0.010$ for $N_2O$ production rate and $P = 0.027$, $P < 0.010$ and
$P = 0.008$ for $CO_2$ production rate, respectively) (Table 2), while $-C_2H_2$ and $+C_2H_2$ varieties of
Experiment 2 did not differ in $N_2O$ and $CO_2$ production rates ($P = 0.402$ and $P = 0.288$, respectively).



**Figure 1: Time series of average $N_2O$ and $CO_2$ production rates during incubation of variety $+C_2H_2$ at the**
**three sample collection times of each soil (Experiment 1 - 4) for treatment A without growth inhibitors, B**
**with bacterial growth inhibition, C with fungal growth inhibition, and D with bacterial and fungal growth**
**inhibition; $P$-values for linear regressions (significance level $\alpha \leq 0.05$). For all significant regressions, $R^2$-**
**values were $\geq 0.46$ and in the case of non-significance, $R^2$-values were $\leq 0.40$.**
**n.d.: There was no detectable $CO_2$ production in Experiment 4 at the first sampling time after 2 hours.**
**(Figure is continued on next page)**

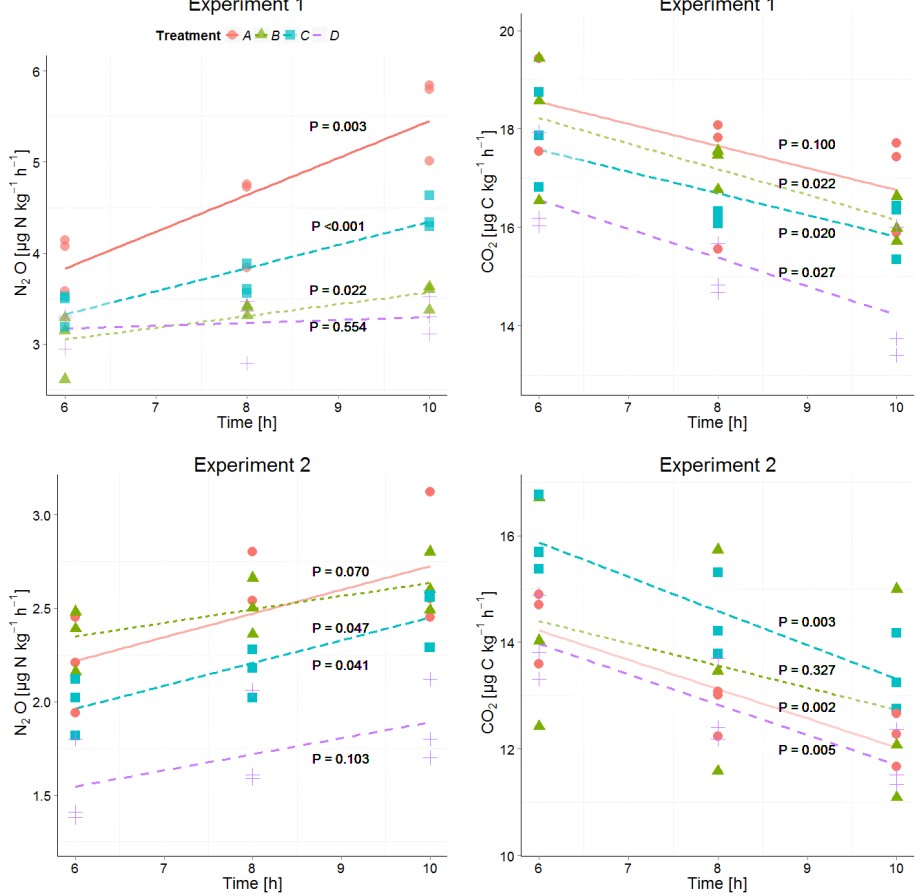


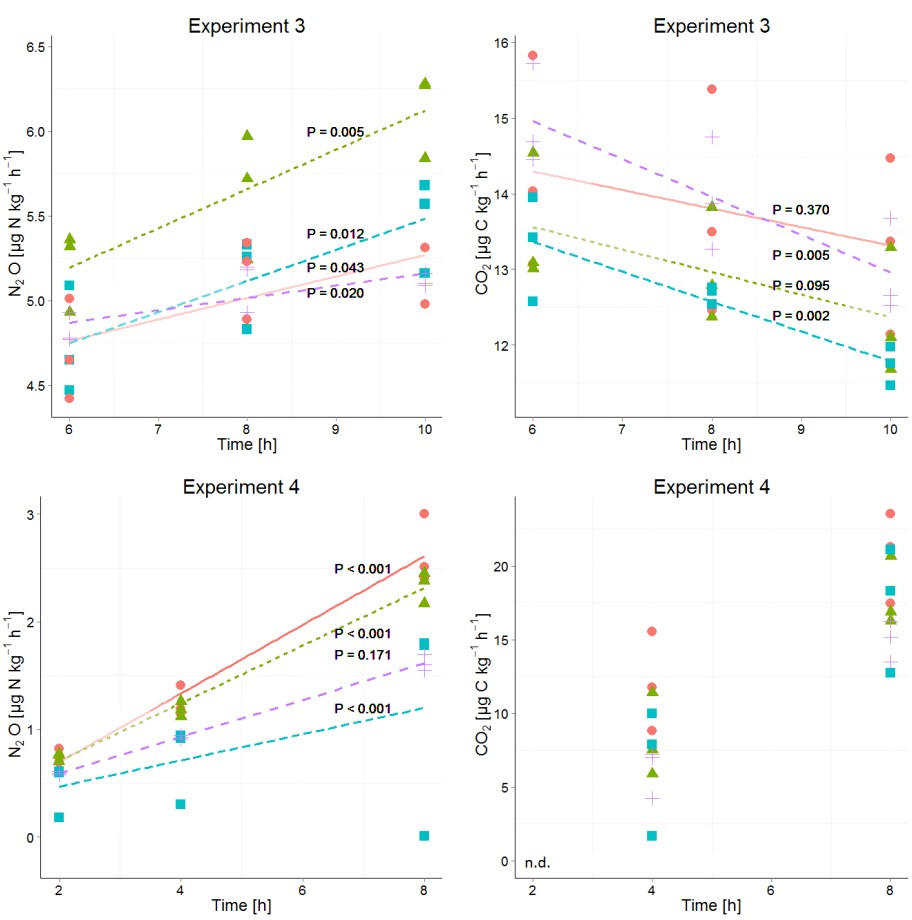

**Figure 1 continued.**

Without blockage of $N_2O$ reductase (variety $-C_2H_2$), $N_2O$ production rates of treatment A varied
significantly among experiments with mean values between 1.6 and 3.6 $\mu g$ N $kg^{-1}$ $h^{-1}$ ($P \leq 0.001$)
(Table 2). In Experiment 1, $N_2O$ production rate was significantly larger (2.7 $\mu g$ N $kg^{-1}$ $h^{-1}$) than in
Experiment 4 (1.6 $\mu g$ N $kg^{-1}$ $h^{-1}$) ($P = 0.028$) in variety $-C_2H_2$. The inhibitor application of each variety
revealed in most cases that treatment A (without growth inhibitors) produced most $N_2O$, followed by
either treatment B (bacterial growth inhibitor; more $N_2O$ compared to treatment C in Experiments 2, 3
and 4) or treatments C (fungal growth inhibitor; more $N_2O$ compared to treatment B in Experiment 1).
In varieties $-C_2H_2$, $+C_2H_2$ and *traced* varieties, non-inhibitable organisms (treatment D) showed
smallest $N_2O$ production rates in most cases (i. e. except of variety $-C_2H_2$ of Experiment 1, varieties -
$C_2H_2$ and traced of Experiment 3 and variety *traced* of Experiment 4). Microbial inhibitor treatments
differed significantly in $N_2O$ fluxes of variety $+C_2H_2$ of each experiment (always $P \leq 0.040$), while
this was not the case for inhibitor treatments of varieties $-C_2H_2$ and *traced* of Experiment 4 ($P = 0.154$
and $P = 0.154$, respectively). Significant deviations of treatments without (A) or with full inhibition



(D) were found in the following cases (Table 2): $N_2O$ production rate of treatment A was significantly
larger compared to the other three treatments of Experiment 1 ($+C_2H_2$ and $-C_2H_2$), Experiment 2
($+C_2H_2$) and Experiment 3 ($+C_2H_2$); treatment D was significantly smaller compared to the other three
treatments in Experiment 2 ($-C_2H_2$) only and compared to treatments A and C of Experiment 1
($+C_2H_2$). Comparing varieties $-C_2H_2$ and *traced*, $N_2O$ and $CO_2$ rates did not differ ($P = 0.991$ for $N_2O$
production rate and $P = 0.490$ for $CO_2$ production rate, respectively), confirming that [15]N-labeling did
not affect $N_2O$ and $CO_2$ processes.





**Table 2: Average $CO_2$ and $N_2O$ production rates and $N_2O$ isotopic values of $N_2O$ of the last sample collection with and without $C_2H_2$ application in the headspace (varieties -$C_2H_2$ and +$C_2H_2$) of each soil (Experiment 1 - 4) for treatments A without, B with bacterial, C with fungal, and D with bacterial and fungal growth inhibition, respectively (standard deviation in brackets, $n$ = 3).**

| Treatment/ variety | mean $N_2O$ [μg N $kg^{-1}$ $h^{-1}$] | mean $CO_2$ [μg C $kg^{-1}$ $h^{-1}$] | $\delta^{18}O_{N2O}$ [‰] | $\delta^{15}N^{bulk}_{N2O}$ [‰] | $SP_{N2O}$ [‰] |
|---|---|---|---|---|---|
| Experiment 1 (Loamy sand, winter 2012) | | | | | |
| A / -$C_2H_2$ | 2.7 (0.4)a | 12.3 (1.7)a | 13.1 (0.2)a | -21.9 (1.7)a | 1.6 (0.8)a |
| B / -$C_2H_2$ | 1.8 (0.2)b | 12.8 (1.6)a | 13.0 (<0.1)* | -24.2 (0.7)* | -1.3 (0.2)* |
| C / -$C_2H_2$ | 2.0 (0.1)b | 11.2 (0.5)a | 14.6 (0.4)a | -20.0 (0.8)a | -1.6 (0.5)a |
| D / -$C_2H_2$ | 2.1 (0.3)b | 13.7 (0.4)a | 15.2 (0.5)* | -20.2 (1.8)* | -0.3 (0.5)* |
| A / +$C_2H_2$ | 5.5 (0.5)a | 17.0 (1.0)a | 8.5 (0.1)a | -22.1 (0.3)a | -0.4 (0.3)a |
| B / +$C_2H_2$ | 3.5 (0.1)b | 16.1 (0.5)a | 7.5 (0.1)a | -26.1 (0.2)a | -1.2 (1.0)b |
| C / +$C_2H_2$ | 4.4 (0.2)c | 14.0 (0.6)a | 9.3 (0.2)a | -22.4 (0.4)a | -0.9 (0.4)b |
| D / +$C_2H_2$ | 3.3 (0.2)b | 14.4 (1.4)a | 7.8 (0.3)* | -24.2 (0.1)* | -2.3 (0.7)* |
| Experiment 2 (Sand, winter 2012) | | | | | |
| A / -$C_2H_2$ | 3.2 (0.4)a | 13.1 (1.0)a | 15.5 (1.8)a | -18.9 (2.6)a | -0.9 (2.5)a |
| B / -$C_2H_2$ | 2.4 (<0.1)b | 12.1 (0.2)a | 15.0 (1.3)a | -23.4 (2.5)a | -0.8 (<0.1)a |
| C / -$C_2H_2$ | 2.5 (0.2)b | 12.0 (0.5)a | 14.3 (0.1)a | -21.8 (0.2)a | -1.8 (0.2)a |
| D / -$C_2H_2$ | 2.0 (0.3)b | 11.0 (0.2)a | 13.4 (0.3)a | -24.5 (0.1)a | -1.2 (0.3)a |
| A / +$C_2H_2$ | 2.7 (0.4)a | 12.7 (2.0)a | 12.6 (0.3)a | -18.9 (4.6)a | -1.4 (0.3)a |
| B / +$C_2H_2$ | 2.6 (0.2)a | 13.4 (0.7)a | 12.3 (0.1)a | -24.6 (0.2)b | -2.0 (0.2)a |
| C / +$C_2H_2$ | 2.5 (0.2)a | 12.2 (0.5)a | 12.7 (0.1)* | -23.3 (0.2)* | -1.7 (0.4)* |
| D / +$C_2H_2$ | 1.9 (0.2)b | 11.7 (0.6)a | 12.2 (0.3)a | -26.0 (0.1)b | -1.5 (0.9)a |
| Experiment 3 (Silt loam, winter 2013) | | | | | |
| A / -$C_2H_2$ | 3.6 (0.2)a | 12.3 (1.0)a | 26.0 (0.5)a | -20.8 (0.5)a | -0.5 (0.4)a |
| B / -$C_2H_2$ | 3.3 (0.4)a | 11.6 (1.8)a | 24.1 (0.2)b | -22.0 (0.2)b | -0.1 (0.4)a |
| C / -$C_2H_2$ | 2.8 (0.1)a | 10.6 (0.6)a | 27.3 (0.1)b | -20.6 (0.3)a | 0.6 (0.2)a |
| D / -$C_2H_2$ | 2.9 (0.4)a | 11.2 (0.7)a | 26.3 (0.3)a | -21.0 (0.1)a | -0.04 (0.18)a |
| A / +$C_2H_2$ | 6.1 (0.3)a | 13.3 (1.2)a | 15.2 (0.1)a | -25.6 (0.8)a | -2.8 (0.2)a |
| B / +$C_2H_2$ | 5.5 (0.3)b | 12.4 (0.8)a | 14.9 (0.2)a | -26.3 (<0.1)a | -3.5 (0.4)a |
| C / +$C_2H_2$ | 5.2 (0.2)b | 11.7 (0.3)a | 16.2 (<0.1)* | -25.2 (0.1)* | -4.0 (0.4)* |
| D / +$C_2H_2$ | 5.1 (<0.1)b | 13.0 (0.6)a | 16.0 (0.1)b | -25.1 (0.1)a | -4.3 (0.5)a |
| Experiment 4 (Loamy sand, summer 2011) | | | | | |
| A / -$C_2H_2$ | 1.8 (0.1)a | 24.5 (1.4)a | 25.7 (0.3)a | -30.6 (0.2)a | 12.1 (1.6)a |
| B / -$C_2H_2$ | 1.2 (0.7)a | 20.9 (0.2)b | 28.0 (5.0)a | -32.3 (0.7)a | 7.7 (1.4)b |
| C / -$C_2H_2$ | 1.0 (0.05)a | 18.4 (1.9)b | 29.3 (0.1)a | -30.0 (0.5)a | 4.3 (1.0)c |
| D / -$C_2H_2$ | 0.7 (0.6)a | 16.3 (1.2)b | 28.9 (1.2)a | -31.8 (2.2)a | 3.4 (2.0)c |
| A / +$C_2H_2$ | 2.6 (0.3)a | 20.8 (3.1)a | 13.5 (0.5)* | -34.7 (0.1)* | -1.0** |
| B / +$C_2H_2$ | 2.3 (0.2)a | 17.9 (2.4)a | 14.3 (1.7)a | -33.8 (0.9)a | -4.9 (0.9)a |
| C / +$C_2H_2$ | 1.2 (1.0)a | 17.4 (4.2)a | 19.0 (7.0)a | -33.1 (2.8)a | -1.7 (2.7)b |
| D / +$C_2H_2$ | 1.6 (0.1)a | 15.0 (1.3)a | 14.8 (0.5)a | -35.7 (0.2)a | -4.9 (0.7)c |

**Letters denote significant differences (P < 0.1) among treatments and varieties within a soil.**
**Asterisks indicate that only two samples (*) or one sample (**) of triplicates were analyzable.**



3.2 Isotopologues of $N_2O$ produced in different varieties and treatments

434        3.2.1 Variety $+C_2H_2$

$SP_{N2O}$ values of all experiments, and all treatments of variety $+C_2H_2$ were within a narrow range
between -4.9 and -0.4 ‰ (Table 2), and differed only significantly among treatments of Experiment 4
($P = 0.002$). In general, there were only small differences among treatments: $SP_{N2O}$ values of
treatments A in variety $+C_2H_2$ differed significantly among soils ($P < 0.001$), with largest $SP_{N2O}$ values
in Experiment 1 (-0.4 ‰) and smallest $SP_{N2O}$ values in Experiment 3 (-2.8 ‰). $SP_{N2O}$ values of
treatment D in variety $+C_2H_2$ of all soils varied between -1.5 and -4.9 ‰, but only $SP_{N2O}$ values of
Experiment 2 differed significantly from $SP_{N2O}$ values of the other Experiments ($P = 0.006$). For
treatments B of variety $+C_2H_2$, $SP_{N2O}$ values differed only significantly between Experiment 1 and 4, 2
and 4, and 1 and 3 (each $P = 0.002$). $SP_{N2O}$ values from treatment C in variety $+C_2H_2$ did not differ
significantly ($P = 0.600$). For every soil we found significantly larger $\delta^{18}O_{N2O}$, $\delta^{15}N^{bulk}_{N2O}$ and $SP_{N2O}$
values in variety $-C_2H_2$ than in variety $+C_2H_2$ ($P < 0.001$), except for Experiment 2, where $\delta^{15}N^{bulk}_{N2O}$
values of variety $-C_2H_2$ were indistinguishable from those of variety $+C_2H_2$ ($P = 0.400$). However,
only in a few varieties there were significant differences in $\delta^{18}O_{N2O}$, $\delta^{15}N^{bulk}_{N2O}$ or $SP_{N2O}$ values
between treatments with fungal and bacterial inhibition (B and C, respectively) (Table 2). $N_2O$
reduction blockage in varieties $+C_2H_2$ was successful in most cases (Experiment 2, 3 and 4). $SP_{N2O}$
values of this variety are thus assumed to be valid estimates of $\delta 0$, i.e. $SP_{prod}$ values of $N_2O$ production,
and can thus be used for applying the IEM.

452        3.2.2 Variety $–C_2H_2$

$SP_{N2O}$ values of all experiments and inhibitor treatments of variety $–C_2H_2$ were within a range of -1.8
to 12.1 ‰ (Table 2) and did not differ among inhibitor treatments ($P = 0.037$). $SP_{N2O}$ values in variety
$-C_2H_2$ of Experiment 4 was particularly large (3.4 - 12.1 ‰) compared to the other experiments (1.6 to
-1.6 ‰). As already stated above, $SP_{N2O}$ values of variety $–C_2H_2$ were significantly larger than $SP_{N2O}$
values of variety $+C_2H_2$ (up to 2.4, 1.5, 4.6 and 4.1‰ in Experiment 1, 2, 3 and 4, respectively).
Generally, most $SP_{prod}$ values of variety $–C_2H_2$ (Eq. 7) were smaller than $SP_{N2O}$ values of variety $–C_2H_2$
but still larger than $SP_{N2O}$ values of variety $+C_2H_2$ and are presented in Table S1 (supplementary
Material).

461        3.2.3 Variety *traced*

The $^{15}N$-labeling of $N_2O$ ($^{15}N_{N2O}$) or $N_2$ produced ($^{15}N_{N2}$) gave information about the incorporated N
from $^{15}N$-labeled $NO_3^-$ into $N_2O$ or $N_2$ as well as about the $N_2O$ reduction to $N_2$. Microorganisms in
each treatment used the $^{15}N$-labeled $NO_3^-$ in variety *traced* (Table 3) and expected $^{15}N_{N2O}$ depended on
the initial N abundance in $NO_3^-$ of unfertilized soil (Eq. 7). Experiment 4 is the only one showing a





large discrepancy between measured (about 30 at%) and calculated $^{15}N_{N2O\_exp}$ (49 at%) in $N_2O$,
whereas the other experiments showed close agreement (Table 3).
3.3 Product ratios of denitrification and efficiency of $N_2O$ reductase blockage by $C_2H_2$
*Product ratio$_{C2H2}$* as well as *product ratio$_{15N}$* of Experiment 2 were significantly larger than of the other
experiments ($P \leq 0.001$) (Table 3). *Product ratio$_{15N}$* of treatment B was significantly larger than of
treatment C and D of Experiment 4 ($P = 0.032$), while all other treatments of other soils did not differ.
*Product ratio$_{C2H2}$* did not differ significantly among treatments ($P = 0.400$). In order to test the
efficiency of blockage of $N_2O$ reduction by $C_2H_2$ application, *product ratio$_{C2H2}$* (Eq. 5) was compared
with *product ratio$_{15N}$* (Eq. 6). In Experiment 1, *product ratio$_{C2H2}$* was by far smaller than *product*
*ratio$_{15N}$*, while both calculated product ratios were in similar ranges in the other three experiments and
thus a successful blockage of $N_2O$ reduction was assumed for those experiments.
**Table 3: Average $CO_2$ and $N_2O$ production rates of the last sample collection after 10 or 8 hours of variety**
***traced*, respectively, with $^{15}N$ labeling in $N_2O$ ($^{15}N$-$N_2O$) and the calculated *product ratio$_{15N}$* of variety *traced***
**and *product ratio$_{C2H2}$* calculated from $N_2O$ production rates of variety $-C_2H_2$ and $+C_2H_2$ of each soil**
**(Experiment 1 - 4) for treatments A without, B with bacterial, C with fungal, and D with bacterial and**
**fungal growth inhibition, respectively (standard deviation in brackets, $n = 3$).**

| Treatment | mean $N_2O$ [µg N kg$^{-1}$ h$^{-1}$] | mean $CO_2$ [µg N kg$^{-1}$ h$^{-1}$]* | $^{15}N_{N2O}$ [at%] | $^{15}N_{N2O\_exp}$ [at%][a] | Calc. total product ratio$_{15N}$[b]* | Calc. total product ratio$_{C2H2}$[c]* |
|---|---|---|---|---|---|---|
| Experiment 1 (Loamy Sand, 2012) | | | | | | |
| A | 2.6 (0.4) | 13.1 (1.7) | 36.8 (0.1) | | 0.80 (0.02) | 0.48 (0.07) |
| B | 1.5 (0.3) | 11.5 (2.4) | 36.4 (0.2) | 39 | 0.76 (0.02) | 0.48 (0.05) |
| C | 1.9 (1.5) | 12.2 (1.1) | 36.9 (<0.1) | | 0.72 (0.05) | 0.45 (0.04) |
| D | 1.5 (<0.1) | 12.5 (0.5) | 36.8 (0.1) | | 0.69 (0.02) | 0.54 (0.05) |
| Experiment 2 (Sand, 2012) | | | | | | |
| A | 2.4 (<0.1) | 12.9 (0.1) | 43.2 (<0.1) | | 0.94 (0.01) | 1.04 (0.10) |
| B | 1.9 (<0.1) | 11.6 (0.2) | 43.0 (0.1) | 44 | 0.94 (0.01) | 0.81 (0.04) |
| C | 2.4 (0.1) | 12.8 (0.6) | 43.2 (0.1) | | 0.95 (0.01) | 0.99 (0.09) |
| D | 1.7 (0.1) | 12.0 (0.3) | 42.7 (0.1) | | 0.93 (0.01) | 0.98 (0.04) |
| Experiment 3 (Silt loam, 2013) | | | | | | |
| A | 2.9 (0.2) | 10.4 (0.5) | 35.8 (<0.1) | | 0.62 (<0.01) | 0.52 (0.04) |
| B | 3.2 (0.2) | 12.0 (0.9) | 35.5 (<0.1) | 34 | 0.62 (0.01) | 0.59 (0.02) |
| C | 2.2 (0.3) | 9.8 (2.0) | 35.5 (<0.1) | | 0.59 (0.02) | 0.48 (0.04) |
| D | 2.3 (0.1) | 9.9 (0.7) | 35.3 (<0.1) | | 0.62 (0.01) | 0.51 (0.04) |
| Experiment 4 (Loamy Sand, 2011) | | | | | | |
| A | 1.6 (0.6) | 31.1 (12.5) | 31.1** | | 0.54 (0.05) | 0.63 (0.10) |
| B | 1.7 (<0.1) | 23.2 (3.0) | 26.5** | 49 | 0.59 (0.03) | 0.63 (0.17) |
| C | 1.2 (<0.1) | 17.9 (0.8) | 30.1* | | 0.50 (0.01) | 0.62 (0.02) |
| D | 1.2 (<0.1) | 17.1 (0.4) | 33.5* | | 0.50 (0.01) | 0.53 (0.12) |

**Asterisks indicate that only two samples (*) or one sample (**) were analyzed.**
**[a]$^{15}N_{N2Oexp}$ [at%] was calculated from Eq. 7.**
**[b]*product ratio$_{15N}$* = [$N_2O/(N_2+N_2O)$] with $N_2O$ or $N_2$ production rates from variety *traced*; see Eq. 5**
**[c]*product ratio$_{C2H2}$* = [$N_2O_{-C2H2}/N_2O_{+C2H2}$] with $N_2O$ production rate from varieties $-C_2H_2$ and $-C_2H_2$; see Eq.**
**6, cf. Table 2**



### 3.4 Fungal contribution to N₂O production from denitrification by microbial inhibitor approach (modified SIRIN)

When calculating $F_{FDmi}$, $N_2O$ production rates of treatment D must be significantly smaller compared to the other treatments and the flux balance according to Eq. 1 and 2 must be consistent. This was only the case in Experiment 2 of variety $+C_2H_2$. The calculated $F_{FDmi}$ (Eq. 3) was $0.28 \pm 0.90$ (Table 5). The respective flux of fungal $N_2O$ was $0.24 \pm 0.08$ µg N kg$^{-1}$ h$^{-1}$. For all other experiments calculation of $F_{FDmi}$ was not possible.

### 3.5 Fungal contribution to N₂O production from denitrification by the SP endmember mixing approach (IEM) and SP/δ¹⁸O isotope mapping approach (SP/δ¹⁸O Map)

When applying SP/$\delta^{18}$O Map, we can assess the plausibility of the determined $F_{FD}$ values based on the $\delta^{18}O_{H2O}$ values obtained from the fitting ($\delta^{18}O_{H2O}$ value in Table 4) and the fitting outcome, i.e. the difference between *product ratio$_{15N}$* and *product ratio$_{MAP}$* (*Diff* in Table 4). The most probable $\delta^{18}O_{H2O}$ value for our experiments can be assumed based on the fact that Braunschweig tap water was added to soil and the original soil water also represent the isotope characteristics typical for this region which is about -7.4‰ (long-term mean Braunschweig precipitation water (Stumpp et al., 2014)). Depending on the season and evaporative losses, this value may slightly vary and the most possible range of soil water in our experiments may vary from about -11 to -4‰ as observed in other experiments conducted in our laboratory in similar conditions (Lewicka-Szczebak et al., 2014; Rohe et al., 2014a; Lewicka-Szczebak et al., 2017; Rohe et al., 2017). Taking this into account, we can say that for Experiment 1, the fungal contribution must be below 0.02, because to obtain any larger $F_{FD}$ values unrealistically small $\delta^{18}O_{H2O}$ values (of -14.9‰) must be fitted (see Table 4). For Experiment 2 both the smaller $F_{FD\_MAP}$ values of 0.01 and the larger ones up to 0.15 are possible, since they are associated with very realistic $\delta^{18}O_{H2O}$ values (of -6.3 and -10.1, respectively) and identical *Diff* of 0.04 (Table 4). For Experiment 3 the only plausible fitting can be obtained for the smallest $SP_{BD}$ values, which are associated with a $\delta^{18}O_{H2O}$ value of -5.6‰ (Table 4). Although the *Diff* for this fitting is slightly higher, the other fittings must be rejected due to unrealistic $\delta^{18}O_{H2O}$ values (of -1.7 and +3.7‰), hence $F_{FD\_MAP}$ values must be 0.04-0.09. Similarly, for Experiment 4, the only plausible fitting can be obtained for the smallest $SP_{BD}$ values, which are associated with a $\delta^{18}O_{H2O}$ value of -6.8‰ (Table 4) and indicate $F_{FD\_MAP}$ values from 0.11 to 0.20. Here this fitting also shows clearly the smallest *Diff* of only 0.01 (Table 4). However, except for Experiment 4, where the *Diff* is smallest for the last fitting, the *Diff* values for other experiments are very similar for different fittings with the largest values in Experiment 3. A better fit (showing smaller *Diff* values) was not possible with any other $SP_{BD}$ and $\delta^{18}O_{H2O}$ values. The $F_{FD\_SP}$ ranged between 0 and approximately 0.15 (Table 5). The results obtained from SP/$\delta^{18}$O Map show $F_{FD\_MAP}$ reaching up to 0.14, 0.15, 0.09 and 0.20 for Experiments 1, 2, 3, and 4 respectively (Table 4, Table 5).





**Table 4: Summary of the results provided by SP/$\delta^{18}$O Map for fraction of fungal denitrification ($F_{FD\_MAP}$) and N$_2$O product ratio (*product ratio$_{MAP}$*) in the acetylated (+$C_2H_2$) and non-acetylated (-$C_2H_2$) treatments for 3 possible $SP_{N2O}$ values from bacterial denitrification ($SP_{BD}$): mean (-1.9‰), maximal (3.7‰), and minimal (-7.5‰). The $\delta^{18}$O values of soil water ($\delta^{18}O_{H2O}$) were fitted to get the lowest difference (*Diff*) between product ratio determined with $^{15}$N treatment and SP/$\delta^{18}$O Map (*product ratio$_{15N}$* and *product ratio$_{MAP}$*). The most plausible fittings are bolded (see discussion for reasons of this choice).**

| Experiment | Variety | *product ratio$_{15N}$* | $SP_{BD}$ [‰] | $\delta^{18}O_{H2O}$ [‰] | *product ratio$_{MAP}$* | Diff | $F_{FD\_MAP}$ |
|---|---|---|---|---|---|---|---|
| 1 | **-$C_2H_2$** | **0.66** | **-1.9** | **-11.2** | **0.66** | **0.00** | **-0.01** |
|  | **+$C_2H_2$** | **1** | **-1.9** | **-11.2** | **1.00** | **0.00** | **0.02** |
|  | -$C_2H_2$ | 0.66 | 3.7 | -6.1 | 0.65 | 0.01 | -0.14 |
|  | +$C_2H_2$ | 1 | 3.7 | -6.1 | 1.00 | 0.00 | -0.16 |
|  | -$C_2H_2$ | 0.66 | -7.5 | -14.9 | 0.66 | 0.00 | 0.08 |
|  | +$C_2H_2$ | 1 | -7.5 | -14.9 | 1.00 | 0.00 | 0.14 |
| 2 | **-$C_2H_2$** | **0.94** | **-1.9** | **-6.3** | **0.90** | **0.04** | **0.01** |
|  | **+$C_2H_2$** | **1** | **-1.9** | **-6.3** | **1.04** | **0.04** | **0.01** |
|  | -$C_2H_2$ | 0.94 | 3.7 | -1.2 | 0.90 | 0.04 | -0.16 |
|  | +$C_2H_2$ | 1 | 3.7 | -1.2 | 1.04 | 0.04 | -0.18 |
|  | **-$C_2H_2$** | **0.94** | **-7.5** | **-10.1** | **0.90** | **0.04** | **0.13** |
|  | **+$C_2H_2$** | **1** | **-7.5** | **-10.1** | **1.04** | **0.04** | **0.15** |
| 3 | -$C_2H_2$ | 0.61 | -1.9 | -1.7 | 0.54 | 0.07 | -0.03 |
|  | +$C_2H_2$ | 1 | -1.9 | -1.7 | 1.04 | 0.04 | -0.05 |
|  | -$C_2H_2$ | 0.61 | 3.7 | 3.7 | 0.54 | 0.07 | -0.14 |
|  | +$C_2H_2$ | 1 | 3.7 | 3.7 | 1.03 | 0.03 | -0.24 |
|  | **-$C_2H_2$** | **0.61** | **-7.5** | **-5.6** | **0.53** | **0.08** | **0.04** |
|  | **+$C_2H_2$** | **1** | **-7.5** | **-5.6** | **1.04** | **0.04** | **0.09** |
| 4 | -$C_2H_2$ | 0.60 | -1.9 | -3.3 | 0.66 | 0.06 | 0.15 |
|  | +$C_2H_2$ | 1 | -1.9 | -3.3 | 0.96 | 0.04 | -0.03 |
|  | -$C_2H_2$ | 0.60 | 3.7 | 1.5 | 0.72 | 0.12 | 0.08 |
|  | +$C_2H_2$ | 1 | 3.7 | 1.5 | 0.91 | 0.09 | -0.21 |
|  | **-$C_2H_2$** | **0.60** | **-7.5** | **-6.8** | **0.61** | **0.01** | **0.20** |
|  | **+$C_2H_2$** | **1** | **-7.5** | **-6.8** | **0.99** | **0.01** | **0.11** |







**Table 5: Ranges of the fraction of N₂O produced by fungi ($F_{FD}$) from four soil experiments using four different approaches: Fungal fraction was calculated using a) the microbial inhibitor approach (modified SIRIN) ($F_{FDmi}$), b) the isotopomer endmember mixing approach (IEM) by SP isotope mixing balance ($F_{FD\_SP}$), c) the IEM by $SP_{N2O}$ isotope mixing balance (IEM) for results from variety -$C_2H_2$ with reduction correction to calculate the $SP_{N2O}$ values ($F_{FD\_SPcalc}$), and d) the $\delta^{18}O/SP$ Map ($F_{FD\_MAP}$) with $\delta^{18}O_{N2O}$ and $SP_{N2O}$ values from variety -$C_2H_2$ and variety +$C_2H_2$. Negative values by IEM and $\delta^{18}O/SP$ Map are assumed to be zero.**

| Experiment | $F_{FDmi}$[a] | $F_{FD\_SP}$[b] | $F_{FD\_SPcalc}$[c] | $F_{FD\_MAP}$[d] |
|---|---|---|---|---|
| 1 | n.d. | 0-0.15 | 0-0.19 | 0-0.02 |
| 2 | 0.19-0.37 | 0-0.14 | 0-0.15 | 0.01-0.15 |
| 3 | n.d. | 0-0.09 | 0-0.18 | 0.04-0.09 |
| 4 | n.d. | 0-0.11 | 0-0.21 | 0.11-0.20 |

[a]Fungal fraction on N₂O production calculated Eq. 3.

[b]Fungal fraction on N₂O production calculated by Eq. 4 for variety +$C_2H_2$ with assuming $SP_{N2O}$ values of N₂O produced by bacteria were 3.7 ‰ (resulting in negative fraction and therefore set to zero) or -7.5 ‰.

[c]Eq. 4 to solve for fungal fraction in variety -$C_2H_2$ with assuming $SP_{N2O}$ values of N₂O produced by bacteria was 3.7 (resulting in negative fraction and therefore set to zero) or -7.5 ‰ and using reduction correction with $\eta_r$=-6 ‰ to calculate $SP_{prod}$ values (Senbayram et al., 2018; Yu et al., 2020).

[d]Fungal fraction on N₂O production calculated by SP/$\delta^{18}O$ Map with assuming most probable $SP_{N2O}$ values from bacterial denitrification (according to Table 4)

n.d.-not determined because of insufficient inhibition.

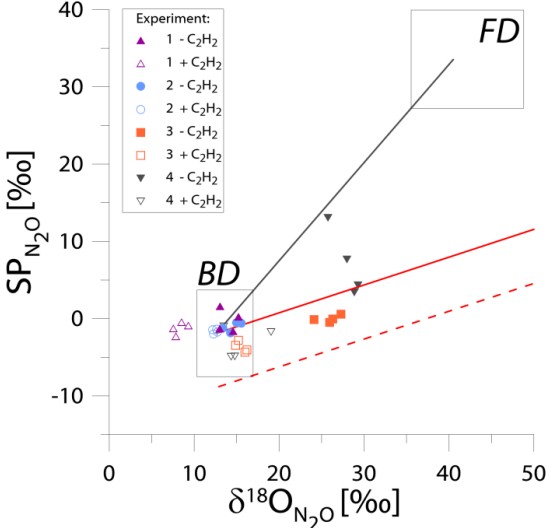

**Figure 2: SP/$\delta^{18}O$ isotope mapping approach (SP/$\delta^{18}O$ Map) to estimate the contribution of bacteria or fungi to N₂O produced according to Lewicka-Szczebak et al. (2017) and Lewicka-Szczebak et al. (2020). The isotopic values for natural abundance treatments with acetylene addition (+$C_2H_2$, empty symbols) and without acetylene addition (-$C_2H_2$, corresponding filled symbols) are shown for four experiments (1-4). The grey rectangles indicate expected ranges of isotopic signatures for heterotrophic bacterial denitrification (BD) and fungal denitrification (FD) (Yu et al. 2020). The black solid line is the mixing line connecting the average expected values for BD and FD, while the red solid line is the mean reduction (for the mean SP values for BD) line and the red dashed line is the minimum reduction line (for the minimal $SP_{N2O}$ values for BD).**



582        3.6 $SP_{N2O}$ values of N$_2$O produced by the fungal soil community

Solving Eq. 4 for $SP_{FD}$ enables to calculate $SP_{N2O}$ values from the fungal soil community for
Experiment 2 (Table 6). Estimates for the ranges of $F_{FD}$ and $F_{BD}$ from the results (+C$_2$H$_2$) of the
modified SIRIN were obtained ($F_{FDmi}$=0.19-0.37 and $F_{BD}$=1- $F_{FDmi}$ resulted in a range between 0.63
and 0.81, respectively, see section "*3.4 Fungal contribution to N$_2$O production from denitrification by*
*microbial inhibitor approach (modified SIRIN)*"). The $SP_{prod}$ values of N$_2$O ($SP_{prod}$ = -1.4 ‰) of the
respective treatment A (Table 2, variety +C$_2$H$_2$) served to calculate $SP_{N2O}$ values for fungal
denitrification for Experiment 2. Assuming -7.5 or 3.7 ‰ for the bacterial $SP_{N2O}$ endmember values of
N$_2$O (Toyoda et al., 2005; Sutka et al., 2006; Yu et al., 2020) resulted in $SP_{FD}$ values between -10 ‰
($SP_{BD}$ = 3.7 ‰) and 25 ‰ ($SP_{BD}$ = -7.5 ‰) (Table 6). The respective $SP_{FD}$ value for variety –C$_2$H$_2$ was
in a very similar range between -17 ‰ and 27 ‰ (Table 6) using $SP_{prod}$ values ($SP_{prod}$ = -1.0 ‰) of the
respective treatment A (Table S1).

**Table 6: $SP_{FD}$ values (i.e. $SP_{N2O}$ values of N$_2$O produced by fungi) by solving Eq. 4 using $F_{FDmi}$ and $F_{BD}$**
**from results of modified SIRIN approach and using $SP_{prod}$ values of varieties +C$_2$H$_2$ and -C$_2$H$_2$ of**
**Experiment 2.**

| Treatment | $SP_{prod}$ [‰] | $SP_{BD}$ [‰][a] | $F_{FDmi}$[b] | $F_{BD}$[b] | $SP_{FD}$ [‰] |
|---|---|---|---|---|---|
| | | -7.5 | 0.19 | 0.81 | 25 |
| +C$_2$H$_2$ | -1.4 | 3.7 | 0.19 | 0.81 | -23 |
| | | -7.5 | 0.37 | 0.63 | 9 |
| | | 3.7 | 0.37 | 0.63 | -10 |
| | | -7.5 | 0.19 | 0.81 | 27 |
| -C$_2$H$_2$ | -1.0 | 3.7 | 0.19 | 0.81 | -17 |
| | | -7.5 | 0.37 | 0.63 | 10 |
| | | 3.7 | 0.37 | 0.63 | -9 |

**$SP_{N2O}$ endmember values of bacterial denitrification were taken for calculation (Eq. 4) according to**
**studies with pure cultures (Toyoda et al., 2005; Sutka et al., 2006; Yu et al., 2020).**
**[b]Ranges of $F_{FDmi}$ and $F_{BD}$ were calculated using the modified SIRIN approach.**

601        **4. Discussion**

To our knowledge, this was the first attempt to determine $SP_{N2O}$ values by fungi or bacteria from soil
communities using microbial growth inhibitors with a modification of SIRIN and comparing microbial
inhibitor and isotopic approaches (IEM and SP/δ$^{18}$O Map) to estimate fungal contribution to N$_2$O
production from denitrification in anoxic incubation. Using IEM revealed that the fungal contribution
to N$_2$O production was small ($F_{FD\_SP} \leq 0.15$ or $F_{FD\_MAP} \leq 0.20$) in the three soils tested (Table 5). Only
one experiment with modified SIRIN allowed the calculation of the fungal fraction producing N$_2$O
during denitrification ($F_{FDmi}$ between 0.19 and 0.37 in Experiment 2), which was larger than the $F_{FD}$ by
two isotope approaches ($\leq 0.20$). While the three approaches coincided in showing dominance of
bacterial denitrification, the isotopic approaches yielded similar estimates of $F_{FD}$ and thus did not
confirm largest $F_{FD}$ of Experiment 2. The strict application of the SIRIN method prescribes proof of



selectivity of the inhibitors (i.e., streptomycin should not inhibit fungi and cycloheximide should not
inhibit bacteria). The SIRIN results obtained with respect to $N_2O$ production by the fungal or bacterial
fraction were rather unsatisfactory and led to unsolved questions, which are discussed in the following
sections.
4.1 Experimental setup
Inhibitor effects, expressed by smaller $N_2O$ production with selective inhibitors (treatments B, C and
D) compared to treatments without inhibitors (A), were only minor in the present study. Previous
studies found much larger inhibitor effects by pre-incubating the soil with selective inhibitors
(Laughlin and Stevens, 2002; Blagodatskaya et al., 2010; Long et al., 2013; Chen et al., 2014). The
experimental design of our incubation setup was, however, in agreement with the original SIRIN
method for respiration (Anderson and Domsch, 1975, 1978) without soil pre-incubation with selective
inhibitors to minimize disturbance of the soil microbial community. Another study performing similar
experiments without pre-incubation with inhibitors did not find effectiveness of application of both
antibiotics during long-term application (up to 48 h) (Ladan and Jacinthe, 2016). Inhibitor application
without pre-incubating with inhibitors was contrary to previous studies focusing on $N_2O$ production
(Laughlin and Stevens, 2002; Blagodatskaya et al., 2010; Long et al., 2013) and we suppose that pre-
incubation with selective inhibitors changes the F:B ratio compared to the undisturbed soil
considerably more than soil incubation without this pre-incubation step. Additionally, although
Blagodatskaya et al. (2010) did not find more inhibitor efficiency after a period of 1 to 20 hours of
pre-incubation with streptomycin, they found greater inhibitor effects of cycloheximide with pre-
incubation phases. This could indicate that microbial distribution changed after exposition to this
inhibitor. Anderson and Domsch (1975) stated already that $CO_2$ production of initially active
organisms can only be ensured up to six or eight hours of experimental duration and biomass activity
is changed by both inhibitors.
It has to be noticed that pre-incubation in previous studies was without glucose, while $N_2O$ production
was analyzed after the addition of glucose as substrate in the present as well as previous studies
(Laughlin and Stevens, 2002; McLain and Martens, 2006; Blagodatskaya et al., 2010; Long et al.,
2013). Glucose initiates the growth of active heterotrophic organisms. Pre-incubation under
denitrifying conditions is not needed for microorganisms to produce denitrifying enzymes as pure
cultures synthesized enzymes capable of denitrification within two to three hours (USEPA, 1993). We
started gas sample collection after two or four hours, when organisms should have produced
denitrifying enzymes and microbial growth of initially active organisms should have started. With
incubation time production rates of $CO_2$ decreased, probably because experimental incubation
conditions provoked unfavorable conditions and physiological changes, e.g. increasing partial pressure
within the closed jars.





The conventional practice of SIRIN implies determination of $c_{opt}$(glucose), $c_{opt}$(streptomycin) or
$c_{opt}$(cycloheximide) with an "Ultragas 3" $CO_2$ analyzer (WösthoffCo., Bochum) (Anderson and
Domsch, 1973) with continuous gas flow and we used this method to determine optimal
concentrations for SIRIN and used these concentrations for the modified SIRIN approach as well. This
optimization procedure was not used in other studies (Laughlin and Stevens, 2002; Blagodatskaya et
al., 2010; Long et al., 2013). We supposed that optimal concentrations for $CO_2$ respiration could work
as well for denitrification, if both inhibitors are apt to inhibit the denitrification process as well. SIRIN
has so far been tested with isolated cultures and soils for microbial growth on agar and CO2
production (Anderson and Domsch, 1975, 1973), but information on $N_2O$ producing processes,
especially denitrification, is still lacking and should be investigated in further studies.
**4.2 Inhibitor effects**
Even with both growth inhibitors (treatment D) $N_2O$ production was large in all experiments, i.e.,
often not significantly smaller than in the other three treatments. Thus, we suppose similar
contributions of non-inhibitable organisms in all treatments. Non-inhibitable organisms could be, for
example, bacteria or fungi that are not in growth stage or may be not affected by inhibitors. These
organisms could be archaea as well, which are also known to be capable of denitrification (Philippot et
al., 2007; Hayatsu et al., 2008). It is known, that archaea are not affected by streptomycin or
cycloheximide (Seo and DeLaune, 2010). However, effects of archaeal occurrence in soil or secondary
effects on fungi or bacteria were not tested in this study. As stated before, Ladan and Jacinthe (2016)
did not find effective inhibition of denitrification by either inhibitor for denitrification although
streptomycin and cycloheximide are commonly used to inhibit denitrification of selective groups.
Thus, similar experiments with different inhibitors, such as the bactericide bronopol and the fungicide
captan presented by Ladan and Jacinthe (2016), should be conducted to evaluate inhibition approaches
and isotopic endmember approaches.
**4.3 Is SIRIN without $C_2H_2$ suitable to examine the fungal contribution to $N_2O$ production in soil?**
In order to determine $SP_{N2O}$ values without alteration by partial reduction of $N_2O$ to $N_2$, $C_2H_2$ was used
to quantitatively block $N_2O$ reduction during denitrification. We found the expected effect of $C_2H_2$
application, i.e. larger $N_2O$ production rates in variety $+C_2H_2$ compared to variety $-C_2H_2$. Calculated
product ratios varied between 0.5 and 0.95 (*product ratio$_{15N}$*) in all soils, showing that $N_2O$ reduction
can have significant effects on measured $N_2O$ production and isotopic values. The product ratio is
controlled by the reaction rate or by the activity of enzymes capable of $N_2O$ reduction (Nos) in the
system. The calculated *product ratio$_{C2H2}$* was within the same range as *product ratio$_{15N}$* in Experiment
2, 3 and 4 (maximal 9% difference), providing the effective blockage of $N_2O$ reductase in variety
$+C_2H_2$. Only in Experiment 1 *product ratio$_{15N}$* and *product ratio$_{C2H2}$* differed by about 34% with larger
calculated reduction in the *tracer* variety, which might be explained by potential incomplete inhibition



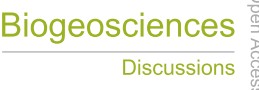

by the $C_2H_2$ method. Nadeem et al. (2013) found some artifacts with $C_2H_2$, which resulted in smaller
$N_2O$ production rates due to NO oxidation accelerated by $C_2H_2$ application in the presence of very
small oxygen (O) amounts ($\geq 0.19$ mL $L^{-1}$). Moreover incomplete $C_2H_2$ diffusion into denitrifying
aggregates might also lead to incomplete $N_2O$ reductase blockage (Groffman et al., 2006). Both
potential methodological errors cannot be excluded for Experiment 1. For the other three experiments
(2, 3 and 4) it can be supposed that the isotopic signature of $N_2O$ of variety $+C_2H_2$ showed isotopic
signatures of produced $N_2O$ without influences of $N_2O$ reduction. By comparing varieties $-C_2H_2$ and
$+C_2H_2$, isotopologue values of all soils (except $\delta^{15}N^{bulk}_{N2O}$ values of Experiment 2) of variety $-C_2H_2$
were significantly larger than those that of variety $+C_2H_2$. The enrichment of residual $N_2O$ in heavy
isotopes results from the isotope effect associated with $N_2O$ reduction (Jinuntuya-Nortman et al., 2008;
Well and Flessa, 2009; Lewicka-Szczebak et al., 2014). This explains why $C_2H_2$ application is
essential for analyzing $N_2O$ produced by different microbial organism groups from soil using solely
the modified SIRIN approach without additional isotopic approaches.
Moreover, when applying SIRIN without quantifying $N_2O$ reduction, fungal denitrification is
potentially overestimated due to the impact of SIRIN inhibitors on $N_2O$ reduction. It is evident that
$N_2O$ fluxes represent net $N_2O$ production, i. e. the difference between gross $N_2O$ production by the
microbial community and $N_2O$ reduction, mainly by heterotrophic bacterial denitrifiers (Müller and
Clough, 2014). The goal of SIRIN application has been to determine the contribution of fungi and
bacteria, respectively, to net $N_2O$ production. It has been shown that $N_2O$ released by microorganisms
to air filled pore space can be partially consumed by denitrifiers before being emitted (Clough et al.,
1998). This means that fungal $N_2O$ can also be subject to reduction by bacterial denitrifiers.
Consequently, inhibiting bacterial denitrification by SIRIN would lead to an overestimation of fungal
contribution to $N_2O$ production. Until now, this effect has not been considered in previous SIRIN
papers on fungal $N_2O$. This effect can only be evaluated by measuring $N_2O$ reduction in all inhibitor
treatments as in our study. If true, the $N_2O$ reduction with bacterial inhibition should be smaller than
that of the treatments without inhibition or with fungal inhibition. Though, with fungal inhibition, $N_2O$
reduction is also assumed to be smaller than that without inhibition, because $N_2O$ produced by fungi is
missed for bacterial reduction. The product ratio is a measure for the $N_2O$ reduction to $N_2$. However,
regarding the *product ratio$_{15N}$*, there was no evidence of different $N_2O$ reduction effects between the
SIRIN treatments. The *product ratio$_{C2H2}$* also revealed indistinguishable values between SIRIN
treatments in Experiment 1 and 4, but it was slightly larger in Experiment 3 with bacterial inhibition
compared to the other treatments. However, this effect was very small, which would only cause small
overestimation of fungal contribution. The smallest $N_2O$ reduction was found in Experiment 2
(*product ratio$_{C2H2}$* values near 1), with smallest *product ratio$_{C2H2}$* with bacterial inhibition (0.81). This
could result in an overestimation of bacterial contribution, since with blockage of $N_2O$ reduction, gross
$N_2O$ production of bacteria is measured. The *product ratio$_{15N}$* and *product ratio$_{C2H2}$* were between 0.5
and 1 and $N_2O$ reduction was thus never consuming most of the produced $N_2O$. Hence both the $C_2H_2$





and Streptomycin effects on SIRIN results were probably low. But the product ratio in soil
denitrification exhibits the full range from 0 to 1, meaning that this effect can be quite relevant and
must thus be considered in future studies.
4.4 $SP_{N2O}$ values of $N_2O$ produced by microbial communities
The $SP_{N2O}$ values of each soil indicated predominantly bacteria to be responsible for $N_2O$ production
during denitrification, assuming that results of $SP_{N2O}$ values of denitrification by pure bacterial
cultures is transferable to bacteria of soil communities contributing to denitrification. The latter
assumption has been confirmed repeatedly in soil incubation studies, where in absence of $N_2O$
reduction smallest $SP_{N2O}$ values have been found that were within the range of bacterial pure cultures
(Lewicka-Szczebak et al., 2015; Lewicka-Szczebak et al., 2017; Senbayram et al., 2018). Therefore,
there was no unequivocal evidence of fungi contributing to $N_2O$ production during denitrification,
although the isotopic approaches revealed a fungal contribution up to 0.20 on $N_2O$ production during
denitrification. The $SP_{N2O}$ values of treatment A within variety $+C_2H_2$ showed that the signature of
produced $N_2O$ was not affected by reduction effects and might give evidence of the microbial
community contributing to $N_2O$ production regarding differences in $SP_{N2O}$ values of pure bacterial or
fungal culture studies (Sutka et al., 2006; Sutka et al., 2008; Frame and Casciotti, 2010; Rohe et al.,
2014a). However, variations in $SP_{N2O}$ values of treatments A of variety $+C_2H_2$ are very small and do
not give a clear evidence of any differences in microbial soil community producing $N_2O$. Lewicka-
Szczebak et al. (2014) analyzed $SP_{N2O}$ values of denitrification with blockage of $N_2O$ reduction by
$C_2H_2$ for the same soils as used in the present study for Experiment 1 and 4 as well as Experiment 3
and revealed $SP_{N2O}$ values between -3.6 and -2.1 ‰, which is similar to the respective $SP_{N2O}$ values of
the present study from -4.9 to -0.4 ‰. This reinforces the conclusion that bacteria dominate gross $N_2O$
production under anoxic conditions in both these soils. However, other studies found larger $SP_{N2O}$
values of produced $N_2O$ unaffected by the reduction effect of up to +6 ‰ (Köster et al., 2013a) most
probably as a result of larger contributions of fungi to $N_2O$ production. However, those results were
obtained in an experimental setup with ambient oxygen concentration, without glucose amendment
and without $C_2H_2$ inhibition of $N_2O$ reduction since $N_2$ gas fluxes were directly measured. It was also
discussed before that short-time incubations under static conditions as presented here, may promote
bacterial over fungal growth, which may also be transferable to denitrification activity by both
organism groups (Lewicka-Szczebak et al., 2017; Lewicka-Szczebak et al., 2014). Additionally to this,
the selection of glucose as substrate in the selected concentration may promote bacteria compared to
fungi even more (Koranda et al., 2014; Reischke et al., 2014).
4.5 $\delta^{18}O_{N2O}$ values
The analysis of $\delta^{18}O_{N2O}$ values can give information about O exchange between water and
denitrification intermediates by various microorganisms (Aerssens et al., 1986; Kool et al., 2007; Rohe





et al., 2014b; Rohe et al., 2017). The range of $\delta^{18}O_{N2O}$ values in our study for variety $+C_2H_2$ (7.5 to
19.0 ‰) was quite similar to the range found by Lewicka-Szczebak et al. (2014) for the same soils
(4.8 to 16.3 ‰), where almost complete O exchange with soil water was documented. Hence, for this
study the O exchange was probably also very high. However, there were no remarkable differences in
$\delta^{18}O_{N2O}$ values among treatments within one variety and soil and therefore we assume no differences
in O exchange among the treatments.
The information on $\delta^{18}O_{N2O}$ values combined with known $\delta^{18}O_{H2O}$ values is also precious information
for differentiation between $N_2O$ mixing and reduction processes (Lewicka-Szczebak et al., 2017).
However, for this study, $\delta^{18}O_{H2O}$ values were not analyzed. However, due to parallel *traced* variety
experiments, we could determine possible $\delta^{18}O_{H2O}$ values for the particular $SP_{N2O}$ values of bacterial
denitrification mixing endmember (Table 4). Since the $\delta^{18}O_{H2O}$ value for the particular geographic
region can be assessed based on the known isotopic signatures of meteoric waters (Lewicka-Szczebak
et al., 2014; Stumpp et al., 2014; Lewicka-Szczebak et al., 2017; Buchen et al., 2018) the most
plausible ranges of $\delta^{18}O_{H2O}$ values can be used to indicate the plausible ranges of $F_{FD\_MAP}$ values. In
case of precisely determined $\delta^{18}O_{H2O}$ values, the calculated $F_{FD\_MAP}$ values could be more precise,
however, here we show that in case of missing $\delta^{18}O_{H2O}$ values but known product ratio, the $SP/\delta^{18}O$
Map can also provide information on $N_2O$ production pathway contributions.

771        4.6 Co-denitrification

The influence of co-denitrification, which is predominantly associated to fungi (Spott et al., 2011),
may have a large impact on $N_2O$ production, since Laughlin and Stevens (2002) found $N_2O$ production
in their experiment derived to 92% from co-denitrification and only 8% from denitrification. So far,
there is no study on $SP_{N2O}$ values of $N_2O$ produced by co-denitrification. Co-denitrification could have
been a contributing process in Experiment 4. When N in $N_2O$ originates only from $^{15}$N-labeled soil
$NO_3^-$, measured $\delta^{15}N^{bulk}_{N2O}$ values as well as the $^{15}$N enrichment of the labelled N pool producing $N_2O$
($a_p$) should show identical $^{15}$N enrichment to the labeled soil $NO_3^-$. During co-denitrification, when
one N atom in $N_2O$ originates from labeled $NO_3^-$ and the other one from another unlabeled and
unknown N source, this results in $a_p$ values and $^{15}$N enrichment of produced $N_2O$ smaller than the
respective enrichment of the $NO_3^-$ pool. The $^{15}$N enrichment of soil $NO_3^-$ was about 60% larger than
the analyzed $^{15}$N enrichment in $N_2O$, leading to the assumption that $N_2O$ was produced not only by
denitrification. We also calculated $a_p$ values of the other three experiments (data not shown) which
coincided with the $^{15}$N enrichment of $N_2O$ (Table 3). Since $a_p$ would not be affected by contributions
of unlabelled $N_2O$ we can thus exclude the possibility that this smaller enrichment could be caused by
dilution of enriched $N_2O$ from denitrification by $N_2O$ production from an unknown N source and thus
verified that this was due to formation of hybrid $N_2O$, probably via co-denitrification (Spott et al.,
2011). In the other experiments there was no indication of co-denitrification being relevant for $N_2O$
production since $^{15}$N enrichments of $NO_3^-$ and $N_2O$ coincided. The question arises, why hybrid $N_2O$



formation was only found when the loamy sand was sampled in summer (June, Experiment 4) but not
when it was sampled during winter (December, Experiment 1). Information on substrates for co-
denitrification, i.e. $NO_2^-$ and $NH_4^+$ or certain organic N compounds could have been different due to
seasonal effects. Moreover, seasonal impacts on microbial community could have been relevant. Since
these possible factors were not assessed in our study and their impact on co-denitrification is still
poorly understood, it is currently not possible to give an answer here. Thus, only the $SP_{N2O}$ values in
Experiment 4 might be influenced by co-denitrification. But since $SP_{N2O}$ values of the acetylated
treatments of Experiment 4 coincided with the $SP_{N2O}$ value range of bacterial denitrification and also
with $SP_{N2O}$ values of the other experiments, our data give no indication that co-denitrification produces
$N_2O$ with $SP_{N2O}$ values differing from bacterial denitrification.
4.7 Calculating the fungal fraction contributing to $N_2O$ production and $SP_{FD}$ values
Due to the inefficiency of microbial inhibition regarding $N_2O$ production in most cases, calculation of
$F_{FDmi}$ contributing to $N_2O$ production was only possible for Experiment 2. Comparing the modified
SIRIN with the isotopic approaches revealed that the fungal fraction contribution to $N_2O$ production
was smaller (about 0.28 in modified SIRIN, ≤0.15 with IEM, ≤0.20 with $SP/\delta^{18}O$ Map) than the
bacterial fraction. Although we did not obtain a very clear picture of various microorganisms
contributing to $N_2O$ production due to the large uncertainties of the calculated fractions, all approaches
coincided by showing dominance of bacterial $N_2O$. In contrast to SIRIN, the isotopic approaches
yielded similar estimates of $F_{FD}$ for all experiments.
In some soil studies using helium incubations the $SP_{Prod}$ values obtained by correction for the
reduction effect on $SP_{N2O}$ values showed significantly larger values than $SP_{N2O}$ of bacterial
denitrification (Köster et al., 2013a; Lewicka-Szczebak et al., 2017; Lewicka-Szczebak et al., 2014;
Senbayram et al., 2018; Senbayram et al., 2020). Therefore, it can be supposed that based on the
isotopic approaches various soils may largely differ in the microbial community that contributes to
$N_2O$ from denitrification. The three tested soils seemed to contain a microbial community where fungi
have minor contributions to $N_2O$ emissions from denitrification compared to bacteria. However, this
may also be due to the applied experimental setup favoring bacterial denitrification by static and
strictly anoxic conditions and due to the choice of glucose as substrate. Senbayram et al. (2018) could
show in an incubation experiment with sufficient $NO_3^-$ supply, that fungal contribution to
denitrification was larger with straw compared to a control without straw addition.
The fungal $SP_{FD}$ values (section 3.6 "*SP of $N_2O$ produced by the fungal soil community*") by SIRIN
were highly variable with values between -23 and +25 ‰, which is smaller than the $SP_{N2O}$ range of
$N_2O$ known from pure cultures (16 - 37 ‰) (Sutka et al., 2008; Rohe et al., 2014a). Unfortunately,
both ranges exhibit a large overlap but also some discrepancy, which precludes a clear conclusion
whether or not Experiment 2 yielded valid estimates of fungal $SP_{N2O}$ values. There may be different
reasons why estimating the $SP_{N2O}$ values using SIRIN of the fungal community was imprecise: the





fungal fraction contributing to denitrification of the tested soils was only small compared to that of
bacteria, $SP_{N2O}$ values were estimated using a large endmember range known from pure culture studies
only, and possible SIRIN artefacts may have occurred as discussed above. The isotopic approaches
should thus be further investigated with soils, where presumable fungi contribute largely to $N_2O$
production during (e. g. acid forest soils, or litter-amended arable soils) (Senbayram et al., 2018) and
using SIRIN with suitable inhibitors (Ladan and Jacinthe, 2016). The critical question whether the
isotopic signatures of fungal $N_2O$ determined in pure culture studies are transferable to natural soil
conditions cannot be fully answered with this study due to large uncertainties associated with the
results of the SIRIN method.
**5. Conclusions**
Selective inhibitor and isotopic approaches coincided in showing dominance of bacterial
denitrification. Neither the modified SIRIN approach, nor IEM or SP/$\delta^{18}$O Map approaches yielded
larger contributions of the fungal $N_2O$ fraction in any experiment. Both selective growth inhibitors of
modified SIRIN confirmed the expected effect on $N_2O$ production only in one out of four experiments,
and $SP_{N2O}$ values of fungal $N_2O$ calculated from this treatment did not appear to be a valid estimate of
this value and need further evaluation. There might be several artefacts in the modified SIRIN, where
further studies should focus on, e.g. including the effectiveness of inhibitors, changes in microbial
community during pre-incubation with inhibitors and effects of bacterial consumption of $N_2O$
produced by fungi in the presence of bacterial growth inhibitors. The present study could show that
consideration of $N_2O$ reduction in further studies is inevitably necessary. Further studies should also
determine the range of $SP_{N2O}$ values known from fungal denitrification as well as the effect of specific
inhibitors on microbial groups producing $N_2O$ and reducing $N_2O$ during denitrification.
*Data availability.* Gas emission and isotopic data are available from the authors on request.
*Author contribution.* HF, NWM, RW and THA designed the experiment. LR carried out the
experiment at Thünen Institute for Climate-Smart Agriculture in Braunschweig. AG, DLS and RW
helped with isotopic analysis and DLS performed the $\delta^{18}$O/SP Map. LR, RW and DLS prepared the
manuscript with contributions from all co-authors.
*Competing interests.* The authors declare that they have no conflict of interest.
*Acknowledgements.* Many thanks are due to Jens Dyckmans for [15]N analysis and to Martina Heuer for
$N_2O$ isotopic analyses. This joint research project was financially supported by the State of Lower-



Saxony and the Volkswagen Foundation, Hanover, Germany. Further financial support was provided
by the German Research Foundation (grant LE 3367/1-1 to DLS, grant WE 1904/8-1 to LR and RW,
and the research unit 2337: "Denitrification in Agricultural Soils: Integrated Control and Modeling at
Various Scales (DASIM)", grant WE 1904/10-1 to RW and WR 211/1-2 to NWM).

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
