# Peer review of "Comparing modified substrate induced respiration with selective inhibition (SIRIN) and $N_2O$ isotope approaches to estimate fungal contribution to denitrification in three arable soils under anoxic conditions"

_Biogeosciences, 2020_

## Referee Comment (RC1) · Eliza Harris (Referee) · 6 Oct 2020

Review of bg-2020-285:
**Comparing modified substrate induced respiration with selective inhibition 2 (SIRIN) and N$_2$O isotope approaches to estimate fungal contribution to 3 denitrification in three arable soils under anoxic conditions**

Fungal denitrification can make a significant contribution to N$_2$O production in soils, however emissions are poorly constrained. This study uses a variety of approaches to attempt to quantify the proportion of N$_2$O produced by fungal denitrification under anaerobic conditions. The methods are carefully applied however the complex treatment design is challenging to follow and a better overview is needed. The interpretation and statistical analysis is careful but somewhat basic and empirical – each of the methods is considered separately, and results from one are often used in another (eg. product ratios) which makes reasoning circular, and assumptions and uncertainties hard to follow. This type of multipronged approach would hugely benefit from a more complex statistical analysis, such as a Bayesian methodology whereby the results from all experiments as well as the uncertainties in many critical parameters from previous studies could all be brought together to gain a much clearer and more robust picture of the results and implications. It would be a great benefit to the paper if the authors would take the opportunity to use such methods to improve the results at this stage, although I suspect they may consider this beyond the scope of the paper and review. The use of English in the paper is not too bad, but would really improve following careful copyediting by a native speaker – it is often awkward and difficult to follow. Overall the paper is of a good scientific quality and worthy of publication, which I recommend once the comments in this review have been addressed.

- **Specific comments:**
  - L266: How did you calibrate N$_2$O isotopic values? Where values and/or precision dependent on N$_2$O concentration? Was interference from or dependency of isotope ratios on CO$_2$, H$_2$O or any other gas observed?
  - L290: I guess D is abiotic production, eg. chemodenitrification and similar. But if D is abiotic production and not any kind of artefact, why does it matter if D is lower than A, B and C for this calculation? And why is the denominator A-D? The equation then surely gives fungal production as a proportion of biotic denitrification production rather than as a proportion of total production, which would be more relevant?
  - L379: Why would production rates change with time throughout the incubation? Why did you only use the 10 h time to compare?
  - Table 2 / Results S 3.1: Rates for D are clearly not negligible, in fact usually on the order of around half of the total N2O production. I don't see this as a big problem for Eq. 3, as I stated earlier, but it is a significant problem for the use of Eq. 4, which assumes mixing of only FD and BD endmembers.
  - L451: Yes, it sounds like they are a valid estimate of emitted N$_2$O ie. without reduction, however the IEM still suffers from the problem of unrepresented processes as evidenced by significant fluxes from D.
  - L458: This maybe suggests a problem with either the product ratio or the fractionation factor?
  - L474: If inhibition was not successful, there would be less N2O following inhibition than was really produced (eg. lower denominator of Eq. 6), and the calculated product ratio would be larger than it should be. This seems to be the case in most of Expt 2 and in Expt 4 but in Expt 1 and 3 the opposite is observed. Why would you observe this effect, which is really strong for Expt 1? An unaccounted for process in tracing? Or an additional impact of +C2H2 on N cycling that is not just due to reduction? Also, it seems like you don't have complete inhibition for 2 and 4 – maybe 10% not inhibited – how much may this affect results?

o S3.4: This suggests that inhibition may have downstream effects on N cycling, eg. through inhibited processing of N species that are important as substrates for other processes. This could be a really significant problem for all your experiments, which all rely to some extent on SIRIN, and warrants a great deal more discussion.

o S3.5: As a rule of thumb, I would have thought that the further the points are from the BD origin, the more FD would be calculated. This appears to be the case for 4 -C2H2 but for 3 -C2H2 the calcuted $F_{FD}$ values are very low. Why is this? Also, most of your points are close to the origin of BD. Can you use uncertainties in isotope measurements and in endmember values to put uncertainty ranges on the $F_{FD}$ estimates? And can you give a minimum $F_{FD}$ that you would detect by this approach? I think given the uncertainties in every term you would need a relatively strong contribution, eg. 20%, for it to be visible.

o S3.6: These are much lower than the endmember you used for FD. How does this impact your other results? If the fungal endmember was lower than you assumed, the FFD from both IEM and mapping approaches would have been underestimated. Indeed following half your calculated endmembers (4 of 8 are negative) FD and BD could be indistinguishable isotopically. Why do you think your endmembers are so low? Could this relate to underexpression when substrates are limiting, or some other effect?

o L723: Well, except that the FD endmembers you found were much lower than expected…?

o L768: I don't think you do show this, because you had really large variability in your FD map values, and no clear quantitative answer for $f_{FD}$ because you had no clear endmember for soil water.

- **Minor comments:**
  o L60: This description of denitrification should be the first sentence in the paragraph.
  o L149-156: This discussion of whether fungal soil and pure culture values agree seems logically to fit before the more detailed introduction to IEM and mixing line approaches. Overall the introduction is a little hard to follow – it would be good to really think about the logical flow of the concepts from least to most complex and structure the intro accordingly.
  o L216-236: A table summarising the treatments and abbreviations used throughout would be very useful here. It is very confusing at the moment and needs to laid out much more clearly.
  o L239: The word "Experiment" here is confusing since it is really four different soils, right? It would be better to call the different soils "Soil 1" and so on. Also, why does Soil 4 get more fertiliser added?
  o L300: $f$ rather than $F$ would be a more common abbreviation for fraction. Also, this assumes no abiotic denitrification.
  o S2.5.1 is very hard to follow because of the treatment designations. Again, a table earlier in the methods is needed, much more clearly linking each specific treatment combination to a clear abbreviation code.
  o L322: *Product ratio* is much too long to be used repeatedly as a variable, maybe just $f_{red}$ or $P$ similar?
  o S3.2.2: -C2H2 is basically a control compared to +C2H2 and it seems like logically it should be discussed first.
  o L645: Partial pressure effects would potentially also be expected to affect N2O production, but you saw an increase in N2O production with time?
  o L4.2: Also potentially abiotic production.

---

## Referee Comment (RC2) · Anonymous Referee #2 · 15 Oct 2020

The manuscript submitted here presents an interesting combination of approaches for assessing the contribution of fungal denitrification to the N2O. By using some SIRIN and two isotopic techniques (endmember mixing) and SP/delta18O mapping, they conclude that the fungal contribution to N2O fluxes under anaerobic conditions in the three investigated soils is modest. In general, the manuscript is well written, and the methods are well elaborated. I however miss a clear rationale for the study. As a consequence, the reader is not guided through the work, so that it is hard to get the main conclusions

of the work, and how the different pieces of the work fit together (i.e. how the methods compare to each other). Below, you can find some critical parts which should be revisited, with regard to objectives, experimental design and methods and discussion of the results. At the end, some more specific comments.

I think the manuscript would benefit from a more straightforward formulation of the objectives. As it is now, the three objectives (L155-162) are hard to differentiate from each other (ie, using three approaches to determine the fungal contribution to N2O efflux; compare the fungal contribution obtained by the three, and evaluate the use of SPN2O values); even more importantly, in the abstract I don't see a connection with such objectives. You can think on hypotheses -e.g. methods (do not) perform equal-, and ways to test them.

It is also not clear to me the reasoning for the selection of the three soils and how this relates to the objectives, so it is hard to evaluate the suitability of the approach. Is it just to get an idea of variability? In L165 you stated that the soils differ in texture to provide different conditions for denitrification, so that might be the reason behind. The experiments take place under anoxic conditions, so texture might not be that relevant, and one may argue that, for example, different C sources for denitrification maybe more important, or different proportion of fungal vs. bacterial microbial biomass. However, the variability across soils, or the potential role of texture on the results is not discussed at all. Furthermore, one of the soils was sampled twice. Why? What is the difference between experiment 1 and 4? Is it about seasonality? Which kind of information did you want to obtain, and which kind of lessons you learned in hand of the results? I am missing this information in the discussion.

As you said, the microbial inhibitors did not have the expected effects. This is evident not only for N2O, but also for CO2. I understand this kind of results are disappointing when investing large efforts in conducting the analysis. But this is a key issue which deserves more attention, since it has important implications for the relevance of the whole study. For example, L609-610 read: "the SIRIN results [. . .] were rather

unsatisfactory and led to unsolved questions" (a similar statement at the end of the discussion, L827-830). This is quite a statement which, to be honest, it is not reflected in the abstract, which describes that, for the one soil where it was possible to quantify the fungal contribution, this was 28%, higher than what obtained by the other methods. Thus, as a reader I would infer that SIRIN might overestimate the fungal contribution to the N2O fluxes, which is quite a different conclusion compared to "SIRIN results were rather unsatisfactory". On the contrary, the conclusions have a totally different approach, focusing almost only in the caveats of the SIRIN approach. By the way, you said you tested the concentrations of inhibitors applied; thus, were the preliminary tests performing better than the "real runs"? How did you test the optimal concentration? In general, when presenting the results of the different methods and discussing them, I missed a profound analysis on which method should be applied, what the cons and pros are and whether methods provide complementary information, which would support the simultaneous use. As it is now, they are presented almost separately, thus failing in in the objective which can be derived from the tile: "comparing . . .approaches to estimate fungal contribution to denitrification . . ."). And I honestly consider this is a serial issue, especially because you are making use of some of the outputs of selected methods as input for the rest of the methods, making all of them dependent to each other. It is only a suggestion, but consider including some table or graph with the main features of each methods and the key info, so that a reader can get an overview at first glance

Specific comments:

L48: What do you mean by "under conditions ensuring larger fungal N2O fluxes"?

L165-169: A more detailed description of the soil sampled will help. In general, the use of experiment/soil/treatment and variety is confusing, e.g. the same soil is used in two "experiments" (see above) and variety might refer to the use of c2h2 or 15N tracer.

Table 1: inorganic N is expressed in mg/L, which is fine for solutions, but not for describing soils. The usual unit for me is mg N /kg soil. Please check consistence of these numbers. See my comments below, these values can have high temporal variability, with consequences on your analysis. Further to the table: what is "natural soil NO3-"? In general, how stable are these numbers for the Braunschweig soils, and for the rest of the sites? They are arable soils, probably subject to fertilization.

L243: What is the rationale of having target soil densities? Do they correspond to the field bulk density?

L248/L378/Figure 1: How did you calculate the N2O (and CO2) fluxes? Since you flushed with N2, I presume that, for t=0, you used a background concentration of 0 for both N2O and CO2 and then calculated the rate of change after 6, 8 or 10 hours (or 2, 4 and 8 in experiment 4). However, you mentioned average production rates (L378, L390); thus, where does the average come from? As you show in Figure 1, rates vary in some cases by more than 100% depending on the incubation length. So, what is your view on this and, more importantly, what is your suggestion for future experiments?

L254: How can you have a fixed measurement precision across different incubation lengths? How was this calculated? The precision is some orders of magnitude lower than the calculated fluxes, but there are some large variations within the same treatment. Is this solely due to spatial variability? Or are you presenting only the analytical precision and leaving out some other sources of uncertainty?

L289: I am not sure I understood this section. I suspect you used two approaches, but they are presented in a mixed way. With the IEM, one calculates the fungal contribution to N2O by solving the equation 4 using the SP of the N2O produced in the acetylene treatment (variety A) and the assumed SP for fungal and bacteria (33.6 for fungi, and -7.5 to +3.7 per mil for bacteria). The N2O from the acetylene is used to eliminate the distortion coming from N2O reduction in the non-acetylene treatment. But according to L304-307, you solve the equation for SP of fungi using FFD. Please clarify this, since it is highly misleading as it is now.

L352: Where did you get the amount of N in unfertilized soils from? Table 1?

L445: How did you assess the success of the acetylene blockage? A rough look to table 2 suggests that is experiment 2 which did not work. In general, this is a fundamental problem, since you don't know the n2o reduction rate a priori (it is precisely the info you want to obtain), unless e.g. application of 15N labelled substrate is combined with N2 isotopic analyses (what you did in section 3.2.3, but I don't see results for the acetylene treatment, or for N2).

L450: Significance level was established at 0.1 (L366) so p = 0.037 is significant

L461: What are the implications for exp. 4? Was there a significant amount of NO3- available in the soil which may compromise your results? Interestingly, many replicates were not analysed; why? For those values coming from two replicates, why didn't you include the standard deviation (as you did in Table 2)?

L479: you probably mean eq. 8

Table 4 and 5: Why did you set the negative values to 0 in Table 5, but not in Table 4?

Further to table 5: How are the ranges calculated? Are they coming from the different replicates, or from different SP and delta18O, or both?

Section 4.1 and 4.2 should be better streamlined. Actually, section 4.1 refers almost completely to inhibitors (which is section 4.2)

L642: It is not clear to me whether high partial pressure induces physiological changes or rather provokes methodological artifacts (or both). In the former case, respiratory effects might also influence denitrification activity. In the latter case, do you expect an effect on diffusion?

L653: The role of abiotic processes should be briefly discussed here.

L674-678: Experiments 1 and 4 were performed on the same soils, but you got completely different results. And this applies for the mismatch between tracers and acetylene method (only in treatment 1, as you say), but also for the tracers results per se (70-80% N2O production ratios in exp. 1, 50-60% in exp. 4). How do you explain this?

L701: Include the papers you refer to.

L702-706: The explanation is right, but I suspect you have many uncertainties in the application of inhibitors. Take into account that CO2 release was not affected by the inhibitors, and N2O not as high as expected, so you may have a significant contribution of non-inhibitable organisms, so that the substrate effect on N2O reduction rates may not be that important

L724-727: The whole sentence is contradictory. Is the SP not an isotopic approach? Please, clarify

---

## Author Comment (AC1) · 26 Nov 2020

***Review response on** "**Comparing modified substrate induced respiration with selective inhibition (SIRIN) and N₂O isotope approaches to estimate fungal contribution to denitrification in three arable soils under anoxic conditions**" **by Lena Rohe et al.***

**Referee #1**

> *We thank the reviewer for the overall positive critique and for the good comments that will largely improve the manuscript.*
> *The authors' answer is in italic font.*

Comparing modified substrate induced respiration with selective inhibition 2 (SIRIN) and N2O isotope approaches to estimate fungal contribution to 3 denitrification in three arable soils under anoxic conditions

Fungal denitrification can make a significant contribution to N2O production in soils, however emissions are poorly constrained. This study uses a variety of approaches to attempt to quantify the proportion of N2O produced by fungal denitrification under anaerobic conditions.

The methods are carefully applied however the complex treatment design is challenging to follow and a better overview is needed.

> *To represent the different methods applied with various measures derived from these methods more comprehensible, we will prepare a scheme to illustrate the methodological procedure. We will change the variables for product ratios of the different methods to $r_{15N}$, $r_{C2H2}$, and $r_{MAP}$ and also for fraction (F) to the more common "f" in the revised version. In the revised version we will also provide a table in supplementary material showing the variables and abbreviations.*

[Figure]

The interpretation and statistical analysis is careful but somewhat basic and empirical – each of the methods is considered separately, and results from one are often used in another (eg. product ratios)

which makes reasoning circular, and assumptions and uncertainties hard to follow. This type of multipronged approach would hugely benefit from a more complex statistical analysis, such as a Bayesian methodology whereby the results from all experiments as well as the uncertainties in many critical parameters from previous studies could all be brought together to gain a much clearer and more robust picture of the results and implications. It would be a great benefit to the paper if the authors would take the opportunity to use such methods to improve the results at this stage, although I suspect they may consider this beyond the scope of the paper and review.

> *The reviewer is correct, that the manuscript and evaluation of the different measures derived from analysis would benefit from a more detailed data analysis including estimations of uncertainties of the different methods tested. However, as stated in the text the methods tested had high uncertainties that could not clearly be quantified with the presented approaches due to too few data. Only one out of four modified SIRIN experiments revealed a result for the fungal fraction contributing to denitrification. For Bayesian probability the very small number of values and the large uncertainties would result in a very wide probability distribution. We are aware that the different approaches have high uncertainties, especially deriving from partly ineffective inhibition of microbial groups, but we think that a further analysis of uncertainties would not contribute to improve understanding of the present data as we have only a small data set to test and compare different methods in parallel.*

The use of English in the paper is not too bad, but would really improve following careful copyediting by a native speaker – it is often awkward and difficult to follow.

> *We apologize for linguistic errors. We will revise the manuscript carefully.*

Overall the paper is of a good scientific quality and worthy of publication, which I recommend once the comments in this review have been addressed.

• Specific comments:

o L266: How did you calibrate N2O isotopic values? Where values and/or precision dependent on N2O concentration? Was interference from or dependency of isotope ratios on CO2, H2O or any other gas observed?

> *The isotopic analysis will be described in more detail in the revised version as follows: "A laboratory standard $N_2O$ gas was used for calibration, having $\delta^{15}N^{bulk}_{N2O}$, $\delta^{18}O_{N2O}$ and $SP_{N2O}$ values of -1.06 ‰, 40.22 ‰, and -2.13 ‰, respectively, in three concentrations (5, 10 and 20 ppm)." Additional information on traps used will be added: "$H_2O$ and $CO_2$ were trapped with magnesium perchlorate and Ascarite, respectively, to prevent any interference with $N_2O$ analysis."*

o L290: I guess D is abiotic production, eg. chemodenitrification and similar. But if D is abiotic production and not any kind of artefact, why does it matter if D is lower than A, B and C for this calculation? And why is the denominator A-D? The equation then surely gives fungal production as a proportion of biotic denitrification production rather than as a proportion of total production, which would be more relevant?

> *The equation for calculating fractions of sources is adapted from the original SIRIN method by Anderson & Domsch 1973. The calculation is based on the assumption that the fraction of $N_2O$ of treatment D is present in all other treatments as well (A, B, C), representing non-inhibitable sources. Thus, calculating (A-D) as dominator enables to calculate the contribution of $N_2O$ production by bacteria or fungi to the proportion of $N_2O$ from bacteria plus fungi. The*

*method is also based on the assumption that only/mainly bacteria and fungi contribute to $N_2O$ production.*

*The reviewer is correct, that abiotic $N_2O$ production may be one source in modified SIRIN treatment D. Additionally to that source, we also cannot exclude $N_2O$ production from organisms that were either not inhibited by the antibiotics (e.g. archaea or incompleteness of selective inhibition) or ineffectiveness when organisms are active but not growing. Both inhibitors block the protein biosynthesis and thus are ineffective for ongoing processes. This was discussed in detail in section 4.2. As stated in the Material and Method section (l. 281 ff.), the dominator has to be A-D underlying the assumption that the proportion of undefined sources in D contribute to $N_2O$ in the other three treatments as well. In the Method and Result section, we will include a reference to the discussion section: "A detailed discussion of inhibitor effects and difficulties with organisms that were not inhibited or abiotic sources is presented in section 4.1 and 4.2".*

o L379: Why would production rates change with time throughout the incubation? Why did you only use the 10 h time to compare?

*The incubation time of the presented study was relatively short (10 hours) compared to other studies focussing on denitrification. However, when using inhibitors it is absolutely necessary to keep incubation time as short as possible to avoid changes in microbial communities due to species development of non-inhibited organisms. This was already described by Anderson & Domsch 1973 and mentioned in l. 631 ff.:"Anderson and Domsch (1975) stated already that $CO_2$ production of initially active organisms can only be ensured up to six or eight hours of experimental duration and biomass activity is changed by both inhibitors." The reason for this is, that inhibitors can also be used as C sources for microbial growth. As stated before, antibiotics inhibit the protein biosynthesis, and therefore an increase in microbial growth should be reached by changing the environment. We will expand this by "Thus, short-time incubation should cause changes in environment for microorganisms and initiate growth on the one hand, while it should avoid the use of inhibitors as C sources by organisms on the other." However, it is well known from a previous study (Ladan & Jancinthe 2016) that incubations with selective growth inhibitors over a too long period result in non-plausible artefacts. We will include this point in the discussion section.*

o Table 2 / Results S 3.1: Rates for D are clearly not negligible, in fact usually on the order of around half of the total N2O production. I don't see this as a big problem for Eq. 3, as I stated earlier, but it is a significant problem for the use of Eq. 4, which assumes mixing of only FD and BD endmembers.

*The reviewer is correct; the large amount of $N_2O$ produced in treatment D is clearly problematic to interpret data of the other treatments. As presented in Eq. 3, the fungal or bacterial proportion is estimated by taking the production of $N_2O$ from treatment D into account. However, this can only be estimated for $N_2O$ production but not for isotopic values of different treatments. As stated in the results and discussion (section 4.4), the $SP_{N2O}$ values did not largely differ between the SIRIN treatments A-D. Thus we are aware of an uncertainty that is difficult to be estimated and we will describe this in more detail in the revised version.*

o L451: Yes, it sounds like they are a valid estimate of emitted N2O ie. without reduction, however the IEM still suffers from the problem of unrepresented processes as evidenced by significant fluxes from D.

*In section 4.2 the inhibitor effects on $N_2O$ production and interpretation of data were discussed with focussing on treatment D. It is correct that the large $N_2O$ production of non-inhibitable sources (D) was too large in Experiment 1, 3, and 4 to estimate $f_{FD}$. Calculation of $f_{FD}$ resulted in a large range and was of course uncertain, as $N_2O$ production in treatment D was large, although it was significantly smaller than that of treatment A. However, we decided to clearly state in the manuscript that SIRIN was not successful, because we have the large amount of non-inhibitable production and the only result for Experiment 2 is actually very unsure. Thus, we also decided to delete the attempt to calculate $SP_{FD}$ using the SIRIN results by resolving Eq. 4 for $SP_{FD}$ (section 2.5.2) since this yielded biased result.*

*As stated in section 4.2, we assumed a similar presence and activity of non-inhibitable $N_2O$ sources on all four treatments and only small variations of $SP_{N2O}$ values among the four modified SIRIN treatments indicated, that bacteria mainly contributed to $N_2O$ production. This was discussed in detail in section 4.4. The IEM, however, relies on isotopic values ($SP_{N2O}$) known from pure culture studies.*

o L458: This maybe suggests a problem with either the product ratio or the fractionation factor?

*The fractionation factor for $N_2O$ reduction (-6‰) was adapted from published data (Yu et al. 2020) and not directly estimated in the present study. Thus your suggestion might be true and the fractionation factor for $N_2O$ reduction may slightly deviate from the literature value. Apart from that we calculated with average values of $SP_{N2O}$ and product ratio$_{15N}$, which of course contributes to deviations between measured and calculated values. A comment on this will be added to section 4.4. The fractionation factor of about -6‰ is an estimate representing a range of measured fractionation factors in soil and pure culture experiments (e.g. Ostrom et al. 2007). Decreasing this average fractionation factor (-6‰) leads to increasing $SP_{prod}$ values, what in turn would result in values more similar to $SP_{N2O}$ values of variety $-C_2H_2$. We will describe this possible uncertainty of the fractionation factor used in the present study in more detail in the discussion section of the revised version.*

o L474: If inhibition was not successful, there would be less N2O following inhibition than was really produced (eg. lower denominator of Eq. 6), and the calculated product ratio would be larger than it should be. This seems to be the case in most of Expt 2 and in Expt 4 but in Expt 1 and 3 the opposite is observed. Why would you observe this effect, which is really strong for Expt 1? An unaccounted for process in tracing? Or an additional impact of +C2H2 on N cycling that is not just due to reduction? Also, it seems like you don't have complete inhibition for 2 and 4 – maybe 10% not inhibited – how much may this affect results?

*It is true, from Eq. 6 (product ratio = $N_2O_{-C2H2}$ / $N2O_{+C2H2}$) unsuccessful blockage of $N_2O$ reduction would result in smaller $N_2O_{+C2H2}$ values, resulting in larger product ratios. It is well known that $N_2O$ blockage with $C_2H_2$ is very challenging, but due to the comprehensive experimental setup we did not conduct a control of effective blockage of $N_2O$ reduction using $C_2H_2$ with $^{15}N$ labelling. However, in the present study we used the comparison of product ratios derived from approaches with and without acetylene and the $^{15}N$ tracing approach. Comparison of both calculations of product ratio as well as possible artefacts of $C_2H_2$ blockage was discussed in section 4.3 (NO oxidation and incomplete diffusion of $C_2H_2$). To clarify this,*

*we will add "It was possible to assess the completeness of blockage of $N_2O$ reduction by $C_2H_2$ with the experimental setup by comparing product ratios among methods." to l. 337. Although estimated from parallel incubations, i. e. different incubation vessels, both product ratios ($^{15}N$ and $C_2H_2$) were in similar ranges for Experiment 2, 3, and 4, while only Experiment 1 revealed differences in the two calculated product ratios. Additionally, the microbial activity may slightly differ among replicates. Regarding the standard deviations, both product ratios were indistinguishable for treatments (Exp. 2, experiment 4 treatment A, B, D).*

o S3.4: This suggests that inhibition may have downstream effects on N cycling, eg. through inhibited processing of N species that are important as substrates for other processes. This could be a really significant problem for all your experiments, which all rely to some extent on SIRIN, and warrants a great deal more discussion.

*This is true and is a common problem of all inhibitor approaches. Treatments B, C and D revealed that a large number of organisms or processes were not fully inhibited in the presented experiment. This was discussed in detail in the discussion section (section 4.2) and this discussion will be expanded. However, the estimations based on stable isotope approaches do not rely on SIRIN results. As the results of both approaches are only compared, this is not a problem for this approach. This will be further clarified in the discussion.*

*It is correct, that inhibitors can also be used as C sources for microbial growth and therefore it is absolutely necessary to keep incubation time as short as possible to prevent changes in microbial communities due to species development, also of non-inhibited organisms. This was already described by Anderson & Domsch 1973 and mentioned in l. 631 ff.:"Anderson and Domsch (1975) stated already that $CO_2$ production of initially active organisms can only be ensured up to six or eight hours of experimental duration and biomass activity is changed by both inhibitors."*

o S3.5: As a rule of thumb, I would have thought that the further the points are from the BD origin, the more FD would be calculated. This appears to be the case for 4 -C2H2 but for 3 -C2H2 the calcuted FFD values are very low. Why is this? Also, most of your points are close to the origin of BD. Can you use uncertainties in isotope measurements and in endmember values to put uncertainty ranges on the FFD estimates? And can you give a minimum FFD that you would detect by this approach? I think given the uncertainties in every term you would need a relatively strong contribution, eg. 20%, for it to be visible.

*The very precise uncertainty analysis of the isotope mapping approach (SP/$\delta^{18}O$ Map) is a complex issue and was published recently (Wu et al., 2019). The uncertainties are indeed large when we take into account all the possible sources of errors. We will include this information with the relevant citation in the discussion.*

*The points for 4-C2H2 and 3-C2H2 are the values for the treatments without inhibition of $N_2O$ reduction. So, these points are shifted from the bD field mostly due to $N_2O$ reduction. SP/$\delta^{18}O$ Map allows for differentiation of $N_2O$ reduction and $N_2O$ fungal admixture. And for these treatments, we calculate that the possible $f_{FD}$ fraction is up to 9 and 20%, respectively. Table 4 presents a summary of the calculation results of different cases (for the range of literature values for $SP_{BD}$ values and possible different $\delta^{18}O(H_2O)$ values. This table can be used as estimation of the possible uncertainty of this approach. The range for calculated $f_{FD}$*

*values varies up to 20% (for Exp4), so this is the absolute uncertainty of this method in this case study. This discussion with assessment of uncertainty will be added in the manuscript.*

o S3.6: These are much lower than the endmember you used for FD. How does this impact your other results? If the fungal endmember was lower than you assumed, the FFD from both IEM and mapping approaches would have been underestimated. Indeed following half your calculated endmembers (4 of 8 are negative) FD and BD could be indistinguishable isotopically. Why do you think your endmembers are so low? Could this relate to underexpression when substrates are limiting, or some other effect?

> *Fungal endmember values obtained by modified SIRIN were biased by the high uncertainty of the SIRIN approach. As stated above we therefore decided to delete the attempt to calculate $SP_{FD}$ values. Therefore, we had to rely on the values known from literature and further experiment would be needed to compare these values with $SP_{N2O}$ values known from selective organisms or from a soil culture. These could be investigated by mixing various fungal species known to occur in soil or by isolating fungal communities from soil and conduct experiments under anoxic conditions with supply of electron acceptors and C sources to investigate denitrification. With these incubations, parallel $^{15}N$ tracing experiments should be conducted to confirm denitrification as the dominating process. However, we think we should not vary $SP_{N2O}$ values for isotope mapping or IEM, but will discuss this point in section 4.4. As stated in section 4.4, we assume bacterial dominance in the denitrifying community and thus a large uncertainty in the estimated fungal fraction. Using a fungal fraction with high uncertainty would thus results in imprecise $SP_{N2O}$ values.*

o L723: Well, except that the FD endmembers you found were much lower than expected…?

> *This paragraph focusses on treatment A (without inhibition), and we interpreted the low $SP_{N2O}$ values as indicative for bacterial dominance of $N_2O$ production. We will clarify this in the revised version. The $^{15}N$ tracing approach revealed that other processes than denitrification played no or only a minor role and. However, to clarify we will change the beginning of this section (4.4) as follows: "As discussed above, all modified SIRIN treatments of Experiment 1, 3 and 4 were largely affected by $N_2O$ from non-inhibitable organisms or processes which of course has an impact on $SP_{N2O}$ values of all SIRIN. This made it inappropriate to calculate $SP_{N2O}$ values for active bacteria or fungi (modified SIRIN B-C).*

o L768: I don't think you do show this, because you had really large variability in your FD map values, and no clear quantitative answer for fFD because you had no clear endmember for soil water.

> *This is correct, the variability among experiments (soils) was large, although it was much smaller among samples from one experiment, with exception of Experiment 4 ($-C_2H_2$). In the revised version we will point out that in the presented application of $SP/\delta^{18}O$ MAP we fitted $\delta^{18}O$ values of water, but calculation would be more precise when measuring $\delta^{18}O$ values of water during comparable experiments.*

• Minor comments:

o L60: This description of denitrification should be the first sentence in the paragraph.

> *We will change this in the revised version.*

o L149-156: This discussion of whether fungal soil and pure culture values agree seems logically to fit before the more detailed introduction to IEM and mixing line approaches. Overall the introduction is

a little hard to follow – it would be good to really think about the logical flow of the concepts from least to most complex and structure the intro accordingly.

*We will revise the introduction. Thank you for the constructive comment.*

o L216-236: A table summarising the treatments and abbreviations used throughout would be very useful here. It is very confusing at the moment and needs to laid out much more clearly.

*Thanks for this constructive and helpful suggestion. We will insert such a table in the supplementary material of the revised version and provide a scheme showing approaches and different measured values in the material and method section (as shown above).*

o L239: The word "Experiment" here is confusing since it is really four different soils, right? It would be better to call the different soils "Soil 1" and so on. Also, why does Soil 4 get more fertiliser added?

*In the revised version the term "Experiment" will be changed throughout the text, tables and figures to "Soil".*

o L300: f rather than F would be a more common abbreviation for fraction. Also, this assumes no abiotic denitrification.

*Thank you for this advice. We will change it in the revised version. We will also include the information that non-inhibitable organisms contributing to $N_2O$ production and abiotic processes are not included in the assumptions of Eq. 4.*

o S2.5.1 is very hard to follow because of the treatment designations. Again, a table earlier in the methods is needed, much more clearly linking each specific treatment combination to a clear abbreviation code.

*Thank you, as described above we will insert such a table.*

o L322: Product ratio is much too long to be used repeatedly as a variable, maybe just fred or P similar?

*Thank you for this advice. We will change it to $r_{15N}$, $r_{C2H2}$, and $r_{MAP}$ in the revised version.*

o S3.2.2: -C2H2 is basically a control compared to +C2H2 and it seems like logically it should be discussed first.

*You are right. We will change the order of 3.2.1 and 3.2.2 in the revised version.*

o L645: Partial pressure effects would potentially also be expected to affect N2O production, but you saw an increase in N2O production with time?

*This is true and unfortunately we did not analyse the partial pressure of $CO_2$ during incubation. However, due to the inhibitor application, the incubation time was rather short and we can only assume that the $N_2O$ production was not largely affected since we found increasing production rates over time.*

o L4.2: Also potentially abiotic production.

*In the revised version we will include information on this as follows: "Additionally, abiotic $N_2O$ production cannot be quantified with the experimental setup, but might be contributing to each inhibitor treatment."*

---

## Author Comment (AC2) · 26 Nov 2020

***Review response on*** "**Comparing modified substrate induced respiration with selective inhibition (SIRIN) and N$_2$O isotope approaches to estimate fungal contribution to denitrification in three arable soils under anoxic conditions**" ***by*** **Lena Rohe et al.**

**Anonymous Referee #2**

> *We thank the reviewer for the critical comments on the structure of the manuscript and for the good comments that will largely improve the manuscript.*
> *The authors' answer is in italic font.*

The manuscript submitted here presents an interesting combination of approaches for assessing the contribution of fungal denitrification to the N2O. By using some SIRIN and two isotopic techniques (endmember mixing) and SP/delta18O mapping, they conclude that the fungal contribution to N2O fluxes under anaerobic conditions in the three investigated soils is modest. In general, the manuscript is well written, and the methods are well elaborated. I however miss a clear rationale for the study. As a consequence, the reader is not guided through the work, so that it is hard to get the main conclusions of the work, and how the different pieces of the work fit together (i.e. how the methods compare to each other). Below, you can find some critical parts which should be revisited, with regard to objectives, experimental design and methods and discussion of the results. At the end, some more specific comments.

I think the manuscript would benefit from a more straightforward formulation of the objectives. As it is now, the three objectives (L155-162) are hard to differentiate from each other (ie, using three approaches to determine the fungal contribution to N2O efflux; compare the fungal contribution obtained by the three, and evaluate the use of SPN2O values); even more importantly, in the abstract I don0t see a connection with such objectives. You can think on hypotheses -e.g. methods (do not) perform equal-, and ways to test them.

> *To represent the different methods applied with various measures derived from these methods more comprehensible, we will prepare a scheme to illustrate the methodological procedure. We will change the variables for product ratios of the different methods to $r_{15N}$, $r_{C2H2}$, and $r_{MAP}$ and also for fraction (F) to the more common "f" in the revised version. In the revised version we will also provide a table in supplementary material showing the variable and abbreviations. Furthermore, the term "Experiment" will be changed throughout the text, tables and figures to "Soil".*

[Figure]

*We will revise the abstract and introduction. The abstract will be revised by including "Three approaches were established (modified SIRIN approach, endmember mixing approach (IEM) and the SP/$\delta^{18}$O mapping approach (SP/$\delta^{18}$O Map) to estimate the $N_2O$ production by a fungal community in soil: i) A modification of the SIRIN approach was used to calculate N2O evolved from selected organism groups, ii) $SP_{N2O}$ values from the acetylated treatment were used in the isotope endmember mixing approach (IEM), and iii) the SP/$\delta^{18}$O mapping approach (SP/$\delta^{18}$O Map) was used to estimate the fungal contribution to $N_2O$ production and $N_2O$ reduction under anaerobic conditions from the non-acetylated treatment to investigate the fungal fraction contributing to $N_2O$ from denitrification in different soils independently. Furthermore, experiments with clear results in determined fungal fraction contributing to $N_2O$ from denitrification using SIRIN will be used to compare $SP_{N2O}$ values of the fungal fraction with fungal $SP_{N2O}$ endmember values previously reported in the literature." in l. 37 ff.*

*We will include the following hypothesises to the introduction (l. 156 ff.):" We hypothesized that the fungal fraction contributing to $N_2O$ from denitrification in different soils using a modified SIRIN approach and isotopic methods will be correlated but not match exactly due to limited inhibitability of microbial communities and variability in $SP_{N2O}$ endmember values. Furthermore, experiments with clear results in determined fungal fraction contributing to $N_2O$ from denitrification using SIRIN will yield fungal $SP_{N2O}$ endmember values within the range of values previously reported in the literature.". Thus, we will change the objectives of the study to (l. 159 ff.): "Therefore, this study aims at (i) determining the fungal contribution to $N_2O$ production by denitrification under anoxic conditions and glucose addition using three arable soils and approaches (modified SIRIN, IEM and the SP/$\delta^{18}$O Map), and to assess the reliability in soil studies and thus assess factors of potential bias of the methods and (ii) to estimate the $SP_{N2O}$ values from a fungal soil community and thus to evaluate the transferability of the pure culture range of the fungal $SP_{N2O}$ endmember values."*

*In the abstract section, we will also clarify the conclusion (l. 43 ff.): "All three approaches tested revealed a small fungal contribution to $N_2O$ fluxes ($F_{FD}$) under anaerobic conditions in the soils tested. Quantifying the fungal fraction with modified SIRIN was in most cases not successful. In only one soil, $F_{FD}$ of modified SIRIN was 0.28±0.09, which was possibly overestimated as it was higher than the results obtained by IEM and SP/$\delta^{18}$O Map (FD of 0 and 0.20, respectively)."*

*According to this, the Conclusion section will be changed to (l. 836 ff.): "However, it has to be pointed out, that quantifying the fungal fraction with modified SIRIN was only possible with one soil and was possibly overestimated. According to this, the $SP_{N2O}$ values of fungal $N_2O$ calculated from the SIRIN treatment did not appear to be a valid estimate of this value and need further evaluation.".*

It is also not clear to me the reasoning for the selection of the three soils and how this relates to the objectives, so it is hard to evaluate the suitability of the approach. Is it just to get an idea of variability? In L165 you stated that the soils differ in texture to provide different conditions for denitrification, so that might be the reason behind. The experiments take place under anoxic conditions, so texture might not be that relevant, and one may argue that, for example, different C sources for denitrification maybe more important, or different proportion of fungal vs. bacterial microbial biomass. However, the variability across soils, or the potential role of texture on the results is not discussed at all. Furthermore, one of the soils was sampled twice. Why? What is the difference between experiment 1 and 4? Is it about seasonality? Which kind of information did you want to obtain, and which kind of lessons you learned in hand of the results? I am missing this information in the discussion.

*Three different soils were assumed to provide various conditions for denitrification and thus also different environments for microbial communities. Thus, the three soils were not selected to analyse effects of land-use type or soil types, but aimed to compare the different methods (modified SIRIN and isotope approaches) to analyse denitrification. Therefore three soils*

*harbouring different microbial communities were used to estimate differences in results of the used methods among soils (as described in section 2.1).*

*The three soils differed in texture, but also in C content, C/N ratio of $C_{org}$ and pH (Table 1) and we identified differing microbial biomass values. Thus we assumed variable community structures and as a consequence, differences in fungal to bacterial abundances were assumed. We thus did not focus on textural effects on denitrification, but aimed to find differences in fungal abundance in soil. We will include this as follows: "Three soils differing in texture, $C_{org}$ content, C/N ratio of $C_{org}$ and pH were chosen assuming that the soils harbour different denitrifying communities, i.e., different fractions of bacteria and fungi contributing to denitrification."*

As you said, the microbial inhibitors did not have the expected effects. This is evident not only for N2O, but also for CO2. I understand this kind of results are disappointing when investing large efforts in conducting the analysis. But this is a key issue which deserves more attention, since it has important implications for the relevance of the whole study. For example, L609-610 read: "the SIRIN results [: : :] were rather unsatisfactory and led to unsolved questions" (a similar statement at the end of the discussion, L827-830). This is quite a statement which, to be honest, it is not reflected in the abstract, which describes that, for the one soil where it was possible to quantify the fungal contribution, this was 28%, higher than what obtained by the other methods. Thus, as a reader I would infer that SIRIN might overestimate the fungal contribution to the N2O fluxes, which is quite a different conclusion compared to "SIRIN results were rather unsatisfactory". On the contrary, the conclusions have a totally different approach, focusing almost only in the caveats of the SIRIN approach.

*We agree to clearly state in the manuscript that SIRIN was not successful, because we have the large amount of non-inhibitable production and the only result for Experiment 2 is actually very unsure. Thus, we decided to delete the attempt to calculate $SP_{FD}$ using the SIRIN results by resolving Eq. 4 for $SP_{FD}$ (section 2.5.2) since this yielded biased results. Consequently, section 3.6 ($SP_{N2O}$ values of $N_2O$ produced by the fungal soil community ) will be deleted. We will focus on $SP_{N2O}$ values from the fungal fraction in the discussion as follows (l. 829): "The critical question whether the isotopic signatures of fungal $N_2O$ determined in pure culture studies are transferable to natural soil conditions cannot be fully answered with this study due to large uncertainties associated with the results of the SIRIN method making it inappropriate to calculate the $SP_{N2O}$ values of the fungal soil fraction. Further experiments would be needed with improved selective inhibition to assure that $SP_{N2O}$ values known from pure cultures or soil isolates (Sutka et al., 2008; Rohe et al., 2014a; Maeda et al., 2015) are true for fungi in selective groups in soil as well.*

*As mentioned above we will change the abstract as follow (l. 43 ff.): "All three approaches tested revealed a small fungal contribution to $N_2O$ fluxes ($F_{FD}$) under anaerobic conditions in the soils tested. Quantifying the fungal fraction with modified SIRIN was in most cases not successful. In only one soil, $F_{FD}$ of modified SIRIN was 0.28±0.09, which was possibly overestimated as it was higher than the results obtained by IEM and SP/$\delta^{18}O$ Map (FD of 0 and 0.20, respectively)."*

By the way, you said you tested the concentrations of inhibitors applied; thus, were the preliminary tests performing better than the "real runs"? How did you test the optimal concentration? In general, when presenting the results of the different methods and discussing them, I missed a profound analysis on which method should be applied, what the cons and pros are and whether methods provide complementary information, which would support the simultaneous use. As it is now, they are presented almost separately, thus failing in in the objective which can be derived from the tile: "comparing : : :approaches to estimate fungal contribution to denitrification : : :"). And I

honestly consider this is a serial issue, especially because you are making use of some of the outputs of selected methods as input for the rest of the methods, making all of them dependent to each other. It is only a suggestion, but consider including some table or graph with the main features of each methods and the key info, so that a reader can get an overview at first glance

> *As described in section 2.2.1 the pre-experiments were conducted as described in the original method to analyse F:B ratio by substrate induced respiration with selective inhibition. Unfortunately, only $CO_2$ production was analysed under oxic conditions and we did not test the optimum conditions under anoxic conditions. Additionally, due to the oxic conditions, $N_2O$ production was not measured in this pre-experiment. We will describe the differences between this pre-experiment and the incubation experiment presented in more detail in the method section. Regarding the different methods used (modified SIRIN and isotope approaches), we want to emphasize that the both isotope approaches, IEM and $SP/\delta^{18}O$ Map, were independent on results of the modified SIRIN approach. While $SP_{N2O}$ values from the acetylated treatment were used in IEM, the $SP/\delta^{18}O$ Map was used to estimate the fungal contribution to $N_2O$ production and $N_2O$ reduction from the non-acetylated treatments. As described above we will point on this in the revised objectives of the study.*
>
> *We will revise the conclusion section and include: "Based on the presented results we conclude that the modified SIRIN approach presented here is not appropriate to estimate the contribution of selected communities (bacteria or fungi) on denitrification from soil. Both isotope approaches (IEM and $SP/\delta^{18}O$ Map) revealed similar and reasonable results of the fungal fraction contributing to denitrification and thus could be recommended in future studies. However, further studies would be needed to cross-validate methods, e. g. with such as improved inhibitor approaches or molecular-based methods."*

Specific comments:

L48: What do you mean by "under conditions ensuring larger fungal N2O fluxes"?

> *When supplying C sources other than glucose, the fungal growth might be greater compared to that with glucose. Apart from that, in a future study one could analyse the microbial community first and by this identify soils with approved high fungal abundance or maybe even a high fungal denitrifier abundance before applying such experiments. We will clarify this in the revised text by including "…by adding C sources preferred by fungi…" to the sentence in the abstract.*

L165-169: A more detailed description of the soil sampled will help. In general, the use of experiment/soil/treatment and variety is confusing, e.g. the same soil is used in two "experiments" (see above) and variety might refer to the use of c2h2 or 15N tracer.

> *As described in a comment above the different soils were not chosen to analyse soil factors controlling denitrification, but were chosen to provide variable microbial communities with assumingly variations in fungal and bacterial ratios contributing to denitrifying community. In the revised version the term "Experiment" will be changed throughout the text, tables and figures to "Soil".*

Table 1: inorganic N is expressed in mg/L, which is fine for solutions, but not for de-scribing soils. The usual unit for me is mg N /kg soil. Please check consistence of these numbers.

> *Thanks for pointing this out. In the revised version, this will be corrected.*

See my comments below, these values can have high temporal variability, with consequences on your analysis. Further to the table: what is "natural soil NO3-"? In general, how stable are these numbers for the Braunschweig soils, and for the rest of the sites? They are arable soils, probably subject to fertilization.

> *This is correct. The measurements for Braunschweig soil were performed in samples from 2012. Especially these values will vary within one year in arable soils. However, we amended the soil with C and N, thus changing the current state of the soil before incubation. Although*

*soil properties and microbial community or biomass may have changed over time, we thus assumed pre-incubating the soil for seven days, applying C and N, and changing the environmental conditions during denitrification induced a rapid growth of specific organisms. Consequently, we were aware, that the denitrifying community and the abundance of these organisms in incubation experiments may differ from the community in the field. We will include a discussion on this point to section 4.1 in the discussion section (l. 621. ff.).*

L243: What is the rationale of having target soil densities? Do they correspond to the field bulk density?

*We did not analyse the bulk density of the tested soils. We repacked the soils according to the expected bulk densities based on texture, i.e. 1.6 g cm$^{-3}$ for a sandy soil and 1.3 g cm$^{-3}$ for a silt loam. We will change the respective sentence in the Material and Methods section as follows (l. 244 ff.): "During packing, the soil density was adjusted to an expected target soil density of 1.6 g cm$^{-3}$ in Experiment 1, 2 and 4 and of 1.3 g cm$^{-3}$ in Experiment 3."*

L248/L378/Figure 1: How did you calculate the N2O (and CO2) fluxes? Since you flushed with N2, I presume that, for t=0, you used a background concentration of 0 for both N2O and CO2 and then calculated the rate of change after 6, 8 or 10 hours (or 2, 4 and 8 in experiment 4). However, you mentioned average production rates (L378, L390); thus, where does the average come from? As you show in Figure 1, rates vary in some cases by more than 100% depending on the incubation length. So, what is your view on this and, more importantly, what is your suggestion for future experiments?

*We calculated the N$_2$O production rates by averaging the measured N$_2$O production over 6, 8 or 10 hours (or 2, 4 and 8 in experiment 4). As you described, we calculated rates between the time point of flushing with N2 (t=0) and 6, 8 or 10 hours (or 2, 4 and 8 in experiment 4). Thus, we did not calculate the difference in rates between two time points, but averaged over the time point of incubation (0-6, 0-8 or 0-10 hours or 0-2, 0-4 and 0-8 in experiment 4). However, although we calculated these average values, the production rates differed largely between the time points.*

*As we modified the conditions for microorganisms directly before the incubation and accumulation of N$_2$O started (i.e. mixing the soil, adding water, nitrate, CO$_2$, flushing the headspace with N$_2$) we expected high activity of a large fraction of microorganisms. The antibiotics inhibit protein biosynthesis, and therefore we aimed to increase microbial growth by changing the environment. We will explain this in more detail in section 4.1: "Thus, short-time incubation should cause changes in environment for microorganisms and initiate growth on the one hand, while it should prevent the use of inhibitors as C sources by organisms on the other." However, it is well known from a previous study (Ladan & Jancinthe 2016) that incubations with selective growth inhibitors over a too long period result in non-plausible artefacts. We will include this point in the discussion section.*

*The incubation time of the presented study was relatively short (10 hours) compared to other studies focussing on denitrification. However, when using inhibitors it is absolutely necessary to keep incubation time as short as possible to prevent changes in microbial communities due to species development of non-inhibited organisms. This was already described by Anderson & Domsch 1973 and mentioned in l. 631 ff.:"Anderson and Domsch (1975) stated already that CO$_2$ production of initially active organisms can only be ensured up to six or eight hours of experimental duration and biomass activity is changed by both inhibitors." The reason for this is that inhibitors can also be used as C sources for microbial growth.*

*In consequence, we would aim for improved inhibitor effectiveness in future studies, but would recommend relatively short incubation times to avoid that microorganisms could use the inhibitors or dead cells as energy sources.*

L254: How can you have a fixed measurement precision across different incubation lengths? How was this calculated? The precision is some orders of magnitude lower than the calculated fluxes, but

there are some large variations within the same treatment. Is this solely due to spatial variability? Or are you presenting only the analytical precision and leaving out some other sources of uncertainty?

*The precision for GC and IRMS analysis is the analytical precision of measurements derived from analysing laboratory standards of different concentrations. We will describe this in more detail in the revised version. Variations within treatments derive from spatial variations and replicate incubation that may differ in microbial activity and thus denitrifying activity.*

L289: I am not sure I understood this section. I suspect you used two approaches, but they are presented in a mixed way. With the IEM, one calculates the fungal contribution to N2O by solving the equation 4 using the SP of the N2O produced in the acetylene treatment (variety A) and the assumed SP for fungal and bacteria (33.6 for fungi, and -7.5 to +3.7 per mil for bacteria). The N2O from the acetylene is used to eliminate the distortion coming from N2O reduction in the non-acetylene treatment. But according toL304-307, you solve the equation for SP of fungi using FFD. Please clarify this, since itis highly misleading as it is now.

*This is correct. We will put more emphasis on the precise description of calculations.*

L352: Where did you get the amount of N in unfertilized soils from? Table 1?

*You are correct; these data are provided in Table 1. We will include this information.*

L445: How did you assess the success of the acetylene blockage? A rough look to table 2 suggests that is experiment 2 which did not work. In general, this is a fundamental problem, since you don't know the n2o reduction rate a priori (it is precisely the info you want to obtain), unless e.g. application of 15N labelled substrate is combined with N2 isotopic analyses (what you did in section 3.2.3, but I don0t see results for the acetylene treatment, or for N2).

*Unfortunately, Table 2 does not give information on completeness of blockage since in natural soils the product ratio can vary between the full range of 0 and 1. With $^{15}N$ tracing we did not conduct a treatment with acetylene, but only without acetylene to estimate the $N_2O$ reduction to $N_2$ by analysing $^{15}N$ in $N_2O$ and $N_2$. Nevertheless, we were able to assess the completeness of blockage of $N_2O$ reduction by $C_2H_2$ with the experimental setup. This was done by comparing product ratios calculated from $+C_2H_2$ and $-C_2H_2$ treatments with product ratio calculated from $^{15}N$ treatments (section 3.3 Table 3). To clarify this, we added "It was possible to assess the completeness of blockage of $N_2O$ reduction by $C_2H_2$ with the experimental setup by comparing product ratios among methods." to l. 337. Although estimated from parallel incubations, i. e. different incubation vessels, both product ratios ($^{15}N$ and $C_2H_2$) were in similar ranges for Experiment 2, 3, and 4, while only Experiment 1 revealed differences in the two calculated product ratios.*

L450: Significance level was established at 0.1 (L366) so p = 0.037 is significant

*Thank you for the remark. We will change it in the revised version.*

L461: What are the implications for exp. 4? Was there a significant amount of NO3- available in the soil which may compromise your results? Interestingly, many replicates were not analysed; why? For those values coming from two replicates, why didn't you include the standard deviation (as you did in Table 2)?

*The soil of experiment 1 and 4 was investigated two times with the aim to identify differences in fungal and bacterial contribution of denitrification. Unfortunately we did not analyse $NO_3^-$ or microbial biomass in 2011 (exp. 4). However, we supplied $NO_3^-$ in excess and we can assume a homogeneous distribution of $NO_3^-$ added due to the experimental procedure. Thus, $NO_3^-$ supply should not affect fungal or bacterial contribution on denitrification in this setup. Thank you for the remark. We will include the standard deviation of the two samples in the revised version and indicate that only two samples were analysed.*

L479: you probably mean eq. 8

*Yes, thanks for spotting this. We will correct this in the revised version.*

Table 4 and 5: Why did you set the negative values to 0 in Table 5, but not in Table 4? Further to table 5: How are the ranges calculated? Are they coming from the different replicates, or from different SP and delta18O, or both?

*We agree that this was imprecisely described and thus included information on the resulting ranges in the table description. Additionally, we will uniformly set negative fractions to zero in the tables.*

Section 4.1 and 4.2 should be better streamlined. Actually, section 4.1 refers almost completely to inhibitors (which is section 4.2)

*This will be done in the revised version.*

L642: It is not clear to me whether high partial pressure induces physiological changes or rather provokes methodological artifacts (or both). In the former case, respiratory effects might also influence denitrification activity. In the latter case, do you expect an effect on diffusion?

*Higher partial pressure could result in lower diffusivity of gases from the soil. Unfortunately, we did not analyse the partial pressure of $CO_2$ during incubation. However, the incubation time was rather short and thus we can assume from published values and own experience that $N_2O$ production was not largely affected since we found increasing production rates over time.*

L653: The role of abiotic processes should be briefly discussed here.

*Thank you for this remark. Although it is known that abiotic denitrification may occur under the presented conditions it is not possible to quantify $N_2O$ produced from abiotic processes with the used setup. We will include the possibility of co-occuring abiotic $N_2O$ production in section 4.1 as follows: "Additionally, abiotic $N_2O$ production cannot be quantified with the experimental setup, but might be contributing to each inhibitor treatment."*

L674-678: Experiments 1 and 4 were performed on the same soils, but you got completely different results. And this applies for the mismatch between tracers and acetylene method (only in treatment 1, as you say), but also for the tracers results per se (70-80% N2O production ratios in exp. 1, 50-60% in exp. 4). How do you explain this?

*Samples were taken at different time points and microbial community may change in seasons during the year. We assume that variations in microbial communities and abundances may be the reasons for the differences in results. We will include this information in the discussion section.*

*We already discussed possibilities of variations in microbial groups for differences in pathways between Experiment 1 and 4 in section 4.6 as follows: "The question arises, why hybrid $N_2O$ formation was only found when the loamy sand was sampled in summer (June, Experiment 4) but not when it was sampled during winter (December, Experiment 1). Amounts of substrates for co-denitrification, i.e. $NO_2^-$ and $NH_4^+$ or certain organic N compounds, could have been different due to seasonal effects. Moreover, seasonal impacts on microbial communities could have been relevant. Since these factors were not assessed in our study and their impact on co-denitrification is still poorly understood, it is currently not possible to give an answer here."*

L701: Include the papers you refer to.

*We will include the requested papers: "(e. g. Laughlin and Stevens, 2002; Ladan and Jacinthe, 2016; Chen et al., 2014)".*

L702-706: The explanation is right, but I suspect you have many uncertainties in the application of inhibitors. Take into account that CO2 release was not affected by the inhibitors, and N2O not as high

as expected, so you may have a significant contribution of non-inhibitable organisms, so that the substrate effect on N2O reduction rates may not be that important

> *Thank you, we will describe this point rather generally and rephrase the sentence as "Consequently, inhibiting bacterial denitrification by SIRIN would lower the flux of fungal $N_2O$". To clarify we will include the references as requested in the comment above: (e. g. Laughlin and Stevens, 2002; Ladan and Jacinthe, 2016; Chen et al., 2014).*

L724-727: The whole sentence is contradictory. Is the SP not an isotopic approach? Please, clarify

> *We apologize for the imprecise description. The sentence will be changed as follows: "In many soil incubation studies with inhibited $N_2O$ reduction very small $SP_{N2O}$ values have been found that were within the range of bacterial pure cultures (Lewicka-Szczebak et al., 2015; Lewicka-Szczebak et al., 2017; Senbayram et al., 2018).*

---

## Referee Report (RR1)

Re-review of bg-2020-285:
**Comparing modified substrate induced respiration with selective inhibition (SIRIN) and N$_2$O isotope approaches to estimate fungal contribution to denitrification in three arable soils under anoxic conditions**

This manuscript has benefitted greatly from the first review round, and now it is close to ready for publication, following the comments listed below. The premise and experimental work is good, but the manuscript is often confusingly written and not concise enough.

- **Specific comments:**
    - Abstract: The abstract is confusing to read due to the complex set up and somewhat inconclusive results. There seems to be too much focus on SIRIN, with less focus on the other approaches. I would suggest to rewrite the abstract to (briefly!) answer the two main questions of this study: i) comparing the three approaches, first in method application and then in results, and ii) quantify importance of fungal denit. The current abstract is also perhaps too long; most of the first paragraph is intro and most of the last is conclusions – both should be shortened to just one concise sentence in the abstract.
    - L72: This is confusing – did L&S 2002 think fungal or co denitrification contributed 92%? Did they distinguish the two? The link here to codenitrification is not properly explained. Even if they often co-occur, surely that doesn't mean codenitrification is an indicator fungal denitrification – co-denit can occur under many conditions without FD also.
    - Introduction: The introduction is quite hard to follow and still somewhat unclear. I recommend to restructure to something very easy to follow (considering the complexity of the topic), eg.:
        - an introduction to the topic as a whole,
        - followed by a single paragraph about each of the three methods (how the method is applied, examples from prev studies, strengths, risks and weaknesses),
        - followed by a synthesis of comparability of the methods as well as things like acetylene inhibition (it is currently very unclear how this relates to the three methods), and
        - finally a summary of what you hope to achieve, questions, hypotheses.
      The exact structure is of course up to you but at the moment it is very hard to follow, and for a complex topic such as this a clear intro is very important.
    - L165: This naming is super confusing. The soil sampled twice should be called 1.1 and 1.2, not 1 and 4, to clarify throughout that these are "more similar" that soils 2 and 3.
    - Figure 1: This table is an improvement but still extremely confusing. Why is acetylene only mentioned at the very right hand side? The formatting in the second-from-right boxes is really odd. The use of acronyms (esp in the second-from-right boxes) means the table is not a good overview for a reader who has not yet finished reading the paper. The linkages eg. between expt varieties is odd. This figure still needs a lot of work – it should function as an overview to guide a reader into the experiments and clarify exactly what was applied to which soils in which order. Using colour would probably help to separate different aspects.
    - L201-216: These results should be shown in a figure in the SI.
    - L216: Is this a valid assumption, considering these are pretty different processes? What are the implications if conditions are not optimal for denitrification?
    - L250: Why more fertiliser for soil 4? And different sample timing (L264)?
    - L307: I understand from your response to the initial review that A-D is conventionally used as the denominator in this equation, ie. you determine FD

relative to FD + inhibitable BD, rather than as a proportion of total denitrification. All the same, I find that using just A as a denominator and calculating FD as a proportion of total denit would be more appropriate in this case, as it is more comparable to the isotope approaches, which are essentially finding FD as a proportion of total denit. I think it would be good to calculate f-FD with A as a denominator in addition to the more traditional A-D. It is well within the scope of this paper to try to improve these methods rather than just apply them as previously done.

- S2.5.1: I think you could consider an addition to this section, which would allow you to also use the non-acetylene experiments with the IEM approach. FD, nitrification, and reduction all act to increase SP. Only denitrification (bacterial, codenit, nitrifier denit…) produce low SP $N_2O$. Therefore, you can calculate a "minimum" denitrification contribution, and thus a "maximum" FD contribution. If you measure, for example, SP = 2 permil, with endmembers of 0 and 33 permil for BD and FD/nitrif (this is just a quick example, of course you want to consider uncertainties!), using Eq. 4 you get a minimum denitrification contribution of (2-0)/(33-0)=94% and thus a maximum FD of 6%. This approach gives a relatively strong constraint on fungal denitrification when you measure a low SP (can't have much FD) and very little or no constraint on FD when you measure a high SP.

- In S2.5.1 I'm also unsure why you completely discount nitrification. I know you have wettish soils and flush with $N_2$ but potentially there is still aerobic microsites, oxygen production in the soils from other processes, … Do you have an estimate or previous study to show nitrification is truly negligible?

- L386: Why 0.1? This is a high significance level. And at L405, why do you sometimes report the P (eg. 0.002, 0.008, 0.027) and sometimes just the comparison to the sig level (P < 0.010)? It would be better to always report the P, unless it is very small, eg. P<0.001.

- Table 4: I would think that the errors causing negative values would also apply to positive values, eg. this simply reflects a high uncertainty of the mapping approach. Simply rejecting negative values will therefore bias any interpretations or aggregations towards an overestimation of fungal denitrification using this technique.
  - This is also reflective of the fact that the mapping approach is extremely uncertain as your points are mostly very close together, providing little constraint on the gradient. The mapping approach would presumably be better suited only to samples with larger ranges (a couple of yours have more range but most not).

- L605-619: SIRIN could be an overestimation simply by definition, as it does not include non-inhibitable in the definition, and FD-MAP could be an overestimation because of the rejection of negative values – so I would say the endmember approach gives the highest confidence.

- S4.4: You have very little evidence at all relating to codenitrification and I think it could be more appropriate to remove this section, or at the very least reduce it to a short paragraph at the end of a different section.

- Discussion: The discussion is much improved from the previous version, and provides a good overview now. The authors could still reread the discussion and ensure (like the introduction) that it is a clear and concise coverage of the results in relation to the research questions and hypotheses.

- **Minor comments:**
  - L99: change to *…interferes with quantification of FD based on SP values…* or something similar.
  - L480: Change thanof to than of
  - S4.2: Should the title be *with* and not *without*?

- L744: I think it would be more appropriate to change "revealed" to "were consistent with". Also, this sentence is long and a bit repetitive and could be improved.

---

## Editor Decision (ED1)

bg-2020-285

Comments by Reviewer 3

Dear Editor,
the named study is a valuable contribution. I support publication. Please find my review below:

The study by Rohe et al. is well written and sheds light on a timely topic, namely the role of fungal denitrification in N2O production. Though the results are limited to arable soils, the study has been devised wisely and brings together state-of-the art methods. It is a valuable contribution to the scientific community and well suited for a journal like BG.
I support publication and have only minor comments, which are given below.
Title
ok
Abstract
L41: the term in brackets doesn't add additional information on the mapping approach. I suggest deleting the brackets.
L45: units unclear. I guess this is the fraction, but for clarity I suggest converting in % at this point.

Introduction
L62: describes is inappropriate. I suggest: Denitrification is the stepwise reduction of nitrate to …
L70: sentence is incomplete: I guess it was …performing respiratory denitrification produce substantial amounts of N2O.
L118: I don't agree that the interpretation is more complex. In situations in which oxygen exchange with water is complete, this stabilizes d18O-NO3, since the 18O in water is more stable than in a nitrate pool that is replenished and consumed through nitrification and denitrification. Thus, an assumption of a constant endmember value becomes possible, which has helped immensely with regard to SP 18O-mapping. Please work out that exchange may stabilize 18O-NO3.

Materials and Methods
L191: the term "substrate induced growth inhibition" is confusing. Please clarify in how far substrate and not inhibitor is responsible for limiting growth.
L282-291: How FFDmi is calculated depends on how well the calculated D of eq 1 and the measured D agree. In other words, at this point of the manuscript, one cannot assess if eq. 3 makes sense because D could be the sum of remaining N2o due to nitrification, uninhibited fungal denitrification, uninhibited bacterial denitrification and abiotic processes. Due to the experimental setup, nitrificatory contributions and abiotic processes are likely to have little relevance, which is in agreement with the author's notion. But that's also why I don't understand why the denominator is A-D and not A. Why is it more sensible to calculate fungal contribution to denitrification with microbial inhibition compared to inhibitable fungal denitricfication ((A-C)/A)?
L306: Please also give the mean SP_BD value, why is only the range given in contrast to SP_FD?
L318-324: section is unclear what values were fitted? Please clarify

Results
Section 3.1: Based on Eq. 1 and 2, the production rate in A = B+C-D. This should be used as a quality criterion for the assumptions met. From table 2, it seems that the assumptions made in deriving Eq 1 and 2 were not valid for N2O production rates. I suggest including the term B+C-D in table 2 and present this as result as well.

Discussion
L632: distribution or community? Please clarify.
L780: please define ap.

L821: larger instead of "smaller than the SPn2o range ...

---

## Author Response (AR3)

***Review response on*** **"Comparing modified substrate induced respiration with selective inhibition (SIRIN) and N₂O isotope approaches to estimate fungal contribution to denitrification in three arable soils under anoxic conditions"**

**First round of review:**

**Referee #1**

Review of bg-2020-285:

> *We thank the reviewer for the overall positive critique and for the good comments that largely improved the manuscript.*
> *The authors' answers are shown in italics.*

Comparing modified substrate induced respiration with selective inhibition 2 (SIRIN) and N2O isotope approaches to estimate fungal contribution to 3 denitrification in three arable soils under anoxic conditions

Fungal denitrification can make a significant contribution to N2O production in soils, however emissions are poorly constrained. This study uses a variety of approaches to attempt to quantify the proportion of N2O produced by fungal denitrification under anaerobic conditions.

The methods are carefully applied however the complex treatment design is challenging to follow and a better overview is needed.

> *To represent the different methods applied with various measures derived from these methods more comprehensibly, we prepared a scheme to illustrate the methodological procedure. We changed the variables for product ratios of the different methods to $r_{15N}$, $r_{C2H2}$, and $r_{MAP}$ in the revised version. In the revised version, we also provided a table in the supplementary material showing all variables and abbreviations.*

[Figure]

The interpretation and statistical analysis is careful but somewhat basic and empirical – each of the methods is considered separately, and results from one are often used in another (eg. product ratios) which makes reasoning circular, and assumptions and uncertainties hard to follow. This type of multipronged approach would hugely benefit from a more complex statistical analysis, such as a Bayesian methodology whereby the results from all experiments as well as the uncertainties in many critical parameters from previous studies could all be brought together to gain a much clearer and more robust picture of the results and implications. It would be a great benefit to the paper if the authors would take the opportunity to use such methods to improve the results at this stage, although I suspect they may consider this beyond the scope of the paper and review.

> *The reviewer is correct, that the manuscript and evaluation of the different measures derived from analysis would benefit from a more detailed data analysis including estimations of uncertainties of the different methods tested. However, as stated in the text the methods tested had high uncertainties that could not clearly be quantified with the presented approaches due to too few data. Only one out of four modified SIRIN experiments yielded a result for the fungal fraction contributing to denitrification. For Bayesian probability, the very small number of values and the large uncertainties would result in a very wide probability distribution. We are aware that the different approaches have high uncertainties, especially deriving from partly ineffective inhibition of microbial groups, but we think that a further analysis of uncertainties would not contribute to improved understanding of the present data as we have only a small data set to test and compare different methods in parallel. To include estimates of the precision of the IEM and SP/$\delta^{18}O$ Map possible ranges using minimum and maximum endmember values were used and thus we could present also ranges or possible maximum values for $f_{FD}$. To give more information on the uncertainty of the SP/$\delta^{18}O$ Map, we now inserted to the result section 3.5 (l. 632 ff.): "Since the precision of $r_{15N}$ (expressed in standard deviation in Table 3) was always $\leq 0.05$, this uncertainty of $r_{15N}$ did not reduce the precision of the fitting (compare large ranges of $\delta^{18}O_{H2O}$ and $r_{MAP}$ values, respectively, in Table 4)."*

The use of English in the paper is not too bad, but would really improve following careful copyediting by a native speaker – it is often awkward and difficult to follow.

> *We apologize for linguistic errors. We carefully revised the manuscript.*

Overall the paper is of a good scientific quality and worthy of publication, which I recommend once the comments in this review have been addressed.

• Specific comments:

o L266: How did you calibrate N2O isotopic values? Where values and/or precision dependent on N2O concentration? Was interference from or dependency of isotope ratios on CO2, H2O or any other gas observed?

> *The isotopic analysis was described in more detail in the revised version as follows (l. 323 ff.): "A laboratory standard $N_2O$ gas was used for calibration, having $\delta^{15}N^{bulk}_{N2O}$, $\delta^{18}O_{N2O}$ and $SP_{N2O}$ values of -1.06 ‰, 40.22 ‰, and -2.13 ‰, respectively, in three concentrations (5, 10 and 20 ppm)." Additional information on traps used was also added (l. 326f.): "$H_2O$ and $CO_2$ were trapped with magnesium perchlorate and ascarite, respectively, to prevent any interference with $N_2O$ analysis."*

o L290: I guess D is abiotic production, eg. chemodenitrification and similar. But if D is abiotic production and not any kind of artefact, why does it matter if D is lower than A, B and C for this calculation? And why is the denominator A-D? The equation then surely gives fungal production as a

proportion of biotic denitrification production rather than as a proportion of total production, which would be more relevant?

> *The equation for calculating fractions of sources is adapted from the original SIRIN Method by Anderson & Domsch 1973. The calculation is based on the assumption that the fraction of $N_2O$ of treatment D is present in all other treatments as well (A, B, C), representing non-inhibitable sources. Thus calculating A-D as dominator enables to calculate the contribution of $N_2O$ production by bacteria or fungi to the proportion of $N_2O$ from bacteria plus fungi. The method is also based on the assumption that only/mainly bacteria and fungi contribute to $N_2O$ production.*

> *The reviewer is correct that abiotic $N_2O$ production may be one source in modified SIRIN treatment D. Additionally to that source, we also cannot exclude $N_2O$ production from organisms that were either not inhibited by the antibiotics (e.g. archaea or incompleteness of selective inhibition) or ineffectiveness when organisms are active but not growing. Both inhibitors block the protein biosynthesis and thus are ineffective for ongoing processes. This is now discussed in detail in section 4.2 (4.1 in the revised version). As stated in the Material and Method section (l. 276 ff.), the dominator has to be A-D underlying the assumption that the proportion of undefined sources in D contribute to $N_2O$ in the other three treatments as well. In the Method and Result sections, we included a reference to the discussion section (l. 364 f.): "A detailed discussion of inhibitor effects and difficulties with organisms that were not inhibited or abiotic sources is presented in section 4.1".*

o L379: Why would production rates change with time throughout the incubation? Why did you only use the 10 h time to compare?

> *The incubation time of the presented study was relatively short (10 hours) compared to other studies focussing on denitrification. However, when using inhibitors, it is absolutely necessary to keep incubation time as short as possible to avoid changes in microbial communities due to species development of non-inhibited organisms. This was already described by Anderson & Domsch 1973 and mentioned in l. 733 ff.:"However, in accordance to Anderson and Domsch (1975) experimental duration should be as short as possible to ensure the $CO_2$ production of initially active organisms." The reason for this is that inhibitors can also be used as C sources for microbial growth. As stated before, antibiotics inhibit the protein biosynthesis, and therefore an increase in microbial growth should be reached by changing the environment. We expanded this by "Thus, short-time incubation is recommended when conducting a modified SIRIN approach, as it should cause changes in conditions for microorganisms and initiate growth on one hand, while it should avoid the use of inhibitors as C sources by organisms."*

o Table 2 / Results S 3.1: Rates for D are clearly not negligible, in fact usually on the order of around half of the total N2O production. I don't see this as a big problem for Eq. 3, as I stated earlier, but it is a significant problem for the use of Eq. 4, which assumes mixing of only FD and BD endmembers.

> *The reviewer is correct; the large amount of $N_2O$ produced in treatment D is clearly problematic to interpret data of the other treatments. As presented in Eq. 3, the fungal or bacterial proportion is estimated by taking the production of $N_2O$ from treatment D into account. However, this can only be estimated for $N_2O$ production but not for isotopic values of different treatments. As stated in the results and discussion (section 4.3 in the revised version), the $SP_{N2O}$ values did not largely differ between the SIRIN treatments A-D. Thus we are aware of an uncertainty that is difficult to be estimated and we described this in more detail in the discussion section 4.3 of the revised version as follows (l. 889 ff.): "As discussed above, all modified SIRIN treatments of Soil 1, 3 and 4 were dominated by $N_2O$ from non-inhibitable organisms or processes. This made it impossible to calculate $SP_{N2O}$ values for active bacteria or fungi*

*(modified SIRIN B-C), also with Soil 2, where a relatively large $N_2O$ production was observed with treatment D (Sutka et al., 2008; Rohe et al., 2014a; Maeda et al., 2015) (see section 4.4)."*

o L451: Yes, it sounds like they are a valid estimate of emitted N2O ie. without reduction, however the IEM still suffers from the problem of unrepresented processes as evidenced by significant fluxes from D.

*The IEM, however, relies on isotopic values ($SP_{N2O}$) known from pure culture studies. Unfortunately, there is no more information on other pathways or sources and its specific isotopic values of $N_2O$ produced. Thus, as in isotopic methods in general, only known pathways and sources can be included and this also applies for the IEM used in the present study. In section 4.1 (revised version) the inhibitor effects on $N_2O$ production and interpretation of data were discussed with focussing on treatment D. It is correct that the large $N_2O$ production of non-inhibitable sources (D) was too large in Experiment 1, 3, and 4 to estimate $f_{FD}$. Calculation of $f_{FD}$ resulted in a large range and was of course uncertain, as $N_2O$ production in treatment D was large, although it was significantly smaller than that of treatment A. However, we decided to clearly state in the manuscript (section 3.4) that SIRIN was not successful, because we have the large amount of non-inhibitable production and the only result for Experiment 2 is actually very unsure. This was inserted as follows (l. 602 ff.): "Although $N_2O$ production rate of treatment D was smaller than that of treatment A (Soil 2), it must be pointed out, that due to the large amount of non-inhibitable production (treatment D), even the result for Soil 2 is actually very unsure. For all other Soils, calculation of $f_{FDmi}$ was not possible, i.e., SIRIN was not successful." As a consequence, we also decided to delete the attempt to calculate $SP_{FD}$ using the SIRIN results by resolving Eq. 4 for $SP_{FD}$ (section 2.5.2) since this yielded biased result.*

*As stated in the revised version, section 4.1, we assumed a similar presence and activity of non-inhibitable $N_2O$ sources in all four treatments and the small variations of $SP_{N2O}$ values among the four modified SIRIN treatments indicated that bacteria mainly contributed to $N_2O$ production. This was discussed in detail in section 4.3.*

o L458: This maybe suggests a problem with either the product ratio or the fractionation factor?

*The fractionation factor for $N_2O$ reduction (-6‰) was adapted from published data (Yu et al. 2020) and not directly estimated in the present study. Thus your suggestion might be true and the fractionation factor for $N_2O$ reduction here may slightly deviate from the literature value. Apart from that we calculated with average values of $SP_{N2O}$ and product $ratio_{15N}$, which of course contributes to deviations between measured and calculated values. A comment on this was added to section 4.3. The fractionation factor of about -6‰ is an estimate representing a range of measured fractionation factors in soil and pure culture experiments (e.g. Ostrom et al. 2007), i.e. deviations from this average factor are not uncommon. Decreasing this average fractionation factor (-6‰) leads to increasing $SP_{prod}$ values, which in turn would result in values more similar to $SP_{N2O}$ values of variety $-C_2H_2$. We described this possible uncertainty of the fractionation factor used in the present study in more detail in discussion section 4.3 of the revised version as follows (l. 617 ff.): "$SP_{prod}$ values (variety $-C_2H_2$) differed from $SP_{N2O}$ values (variety $+C_2H_2$), which may result from deviations between the actual fractionation factor that was not estimated in the present study and the used fractionation factor of -6‰ adapted from the literature (Yu et al., 2020). If so, we could assume smaller fractionation effects in the present study as decreasing this average fractionation factor (-6‰) would lead to increasing $SP_{prod}$ values, which in turn would result in values more similar to $SP_{N2O}$ values of variety $-C_2H_2$."*

o L474: If inhibition was not successful, there would be less N2O following inhibition than was really produced (eg. lower denominator of Eq. 6), and the calculated product ratio would be larger than it should be. This seems to be the case in most of Expt 2 and in Expt 4 but in Expt 1 and 3 the opposite is

observed. Why would you observe this effect, which is really strong for Expt 1? An unaccounted for process in tracing? Or an additional impact of +C2H2 on N cycling that is not just due to reduction? Also, it seems like you don't have complete inhibition for 2 and 4 – maybe 10% not inhibited – how much may this affect results?

> It is true, from Eq. 6 (product ratio = $N_2O_{-C2H2}/N_2O_{+C2H2}$) unsuccessful blockage of $N_2O$ reduction would result in smaller $N_2O_{+C2H2}$ values, resulting in larger product ratios. It is well known that $N_2O$ blockage with $C_2H_2$ is very challenging, but due to the comprehensive experimental setup we did not conduct a control of effective blockage of $N_2O$ reduction using $C_2H_2$ with $^{15}N$ labelling. However, in the present study we used the comparison of product ratios derived from approaches with and without acetylene and the $^{15}N$ tracing approach. Comparison of both calculations of product ratio as well as possible artefacts of $C_2H_2$ blockage was discussed in section 4.2 (NO oxidation and incomplete diffusion of $C_2H_2$). To clarify this, we added "It was possible to assess the completeness of blockage of $N_2O$ reduction by $C_2H_2$ with the experimental setup. If $r_{15N}$ and $r_{C2H2}$ were in agreement, a complete blockage of $N_2O$ reduction could be assumed. " to l. 396 ff. Although estimated from parallel incubations, i.e. different incubation vessels, both product ratios ($^{15}N$ and $C_2H_2$) were in similar ranges for Soil 2, 3, and 4, while only Soil 1 revealed differences in the two calculated product ratios. Additionally, the microbial activity may slightly differ among replicates. Regarding the standard deviations, both product ratios were indistinguishable for treatments (Soil 2, Soil 4 treatment A, B, D).

o S3.4: This suggests that inhibition may have downstream effects on N cycling, eg. through inhibited processing of N species that are important as substrates for other processes. This could be a really significant problem for all your experiments, which all rely to some extent on SIRIN, and warrants a great deal more discussion.

> The concern about downstream effects of inhibitors is true and is a common and known problem of all inhibitor approaches. Treatments B, C and D revealed that a large number of organisms or processes were not fully inhibited in the presented experiment. This was discussed in detail in the discussion section (section 4.1 in revised version) and this discussion was expanded. However, the estimations based on stable isotope approaches do not rely on $N_2O$ production of modified SIRIN results. As the results of both approaches are only compared, this is not a problem for this approach. This was further clarified in the discussion section 4.5 as follows (l. 984 ff.): "Due to the inefficiency of the inhibition of microbial $N_2O$ production in most cases, calculation of $f_{FDmi}$ contributing to $N_2O$ production was possible for Soil 2 only, although even this calculated value included inaccuracies. The isotopic approaches, however, which are independent of modified SIRIN results, yielded similar estimates of $f_{FD}$ for all Soils.".
> Furthermore, inhibitors can also be used as C sources for microbial growth and therefore it is absolutely necessary to keep incubation time as short as possible to prevent changes in microbial communities due to species development, also of non-inhibited organisms. This was already described by Anderson & Domsch 1975 and mentioned in l. 732 ff.:"However, in accordance to Anderson and Domsch (1975) experimental duration should be as short as possible to ensure the $CO_2$ production of initially active organisms."

o S3.5: As a rule of thumb, I would have thought that the further the points are from the BD origin, the more FD would be calculated. This appears to be the case for 4 -C2H2 but for 3 -C2H2 the calcuted FFD values are very low. Why is this? Also, most of your points are close to the origin of BD. Can you use uncertainties in isotope measurements and in endmember values to put uncertainty ranges on the FFD estimates? And can you give a minimum FFD that you would detect by this approach? I think given

the uncertainties in every term you would need a relatively strong contribution, eg. 20%, for it to be visible.

> *The uncertainty analysis of the isotope mapping approach (SP/$\delta^{18}$O Map) is a complex issue and was published recently (Wu et al., 2019). The uncertainties are indeed large when we take into account all the possible sources of errors. We included this information with the relevant citation in the discussion 4.5 as follows (l. 988 ff.): "As recently published (Wu et al., 2019), uncertainty analysis is a complex issue and large uncertainties of the results from the SP/$\delta^{18}$O Map approach ca be assumed when all the possible sources of errors are taken into account.". The points for 4-C$_2$H$_2$ and 3-C$_2$H$_2$ are the values for the treatments without inhibition of N$_2$O reduction. So, these points are shifted from the bD field mostly due to N$_2$O reduction. SP/$\delta^{18}$O Map allows for differentiation of N$_2$O reduction and N$_2$O fungal admixture. And for these treatments, we calculate that the possible $f_{FD}$ is up to 9 and 20%, respectively. Table 4 presents a summary of the calculation results of different cases (for the range of literature values for SP$_{BD}$ values and possible different $\delta^{18}$O(H$_2$O) values). This Table can be used as estimation of the possible uncertainty of this approach. The range of calculated $f_{FD}$ values varies up to 20% (for Soil 4), so this is the absolute uncertainty of this method in this case study. Ranges for $f_{FD}$ for each Soil can be used for estimating the absolute uncertainty of this approach, as a summary of the calculation results of different cases (presenting the range of literature values for SP$_{BD}$ values and resulting $\delta^{18}O_{H2O}$ values (Table 5).*

o S3.6: These are much lower than the endmember you used for FD. How does this impact your other results? If the fungal endmember was lower than you assumed, the FFD from both IEM and mapping approaches would have been underestimated. Indeed following half your calculated endmembers (4 of 8 are negative) FD and BD could be indistinguishable isotopically. Why do you think your endmembers are so low? Could this relate to underexpression when substrates are limiting, or some other effect?

> *Fungal endmember values obtained by modified SIRIN were biased by the high uncertainty of the SIRIN approach. As stated above we therefore decided to delete the attempt to calculate SP$_{FD}$ values. Therefore, we had to rely on the values known from literature and further experiment would be needed to compare these values with SP values known from selective organisms or from a soil culture. These could be investigated by mixing various fungal species known to occur in soil or by isolating fungal communities from soil and conduct experiments under anoxic conditions with supply of electron acceptors and C sources to investigate denitrification. With these incubations, parallel $^{15}$N tracing experiments should be conducted to confirm denitrification as the dominating process. Because our experiments did not yield useful results for fungal endmembers, we kept literature values of fungal endmembers of mapping or IEM. As stated in section 4.3, we assume bacterial dominance in the denitrifying community and thus a large uncertainty in the estimated fungal fraction. Using a fungal fraction with high uncertainty thus results in imprecise SP values.*

o L723: Well, except that the FD endmembers you found were much lower than expected…?

> *This paragraph focusses on treatment A (without inhibition), and we interpreted the low SP$_{N2O}$ values as indicative for bacterial dominance of N$_2$O production. We clarified this in the revised version. The $^{15}$N tracing approach revealed that other processes than denitrification played no or only a minor role. However, to clarify, we changed the beginning of this section (4.3) as follows (l. 889 ff.): "As discussed above, all N$_2$O fluxes of modified SIRIN treatments of Soil 1, 3 and 4 were largely dominated by N$_2$O from non-inhibitable organisms or processes. This made it impossible to calculate SP$_{N2O}$ values for active bacteria or fungi (modified SIRIN B and C), also with Soil 2, where a relatively large N$_2$O production was observed with treatment D (Sutka et al., 2008; Rohe et al., 2014a; Maeda et al., 2015) (see section 4.4)."*

o L768: I don't think you do show this, because you had really large variability in your FD map values, and no clear quantitative answer for fFD because you had no clear endmember for soil water.

*This is correct, the variability among experiments (Soils) was large, although it was much smaller among samples from one experiment, with exception of Experiment 4 ($-C_2H_2$). In the revised version (section 4.5) we pointed out that in the presented application of $SP/\delta^{18}O$ Map we fitted $\delta^{18}O$ values of water by (l.1002 ff.) "Since the $\delta^{18}O_{H2O}$ value for the particular geographic region can be assessed based on the known isotopic signatures of meteoric waters (Lewicka-Szczebak et al., 2014; Stumpp et al., 2014; Lewicka-Szczebak et al., 2017; Buchen et al., 2018), the most plausible ranges of $\delta^{18}O_{H2O}$ values can be used to indicate the plausible ranges of $f_{FD\_MAP}$ values." However, calculation would be more precise when measuring $\delta^{18}O$ values of water during comparable experiments (l. 991 ff.): "Regarding the presented application of $SP/\delta^{18}O$ Map, calculation would be more precise when measuring $\delta^{18}O_{H2O}$ than from the fitted $\delta^{18}O_{H2O}$ values." This was also added to section 3.5 (l. 614 ff.): "Thus, in the presented application of $SP/\delta^{18}O$ Map, $\delta^{18}O_{H2O}$ values were fitted and it has to be pointed out that the precision of such calculations can be improved by measuring $\delta^{18}O_{H2O}$ instead."*

• Minor comments:

o L60: This description of denitrification should be the first sentence in the paragraph.

*We changed this in the revised version.*

o L149-156: This discussion of whether fungal soil and pure culture values agree seems logically to fit before the more detailed introduction to IEM and mixing line approaches. Overall the introduction is a little hard to follow – it would be good to really think about the logical flow of the concepts from least to most complex and structure the intro accordingly.

*We carefully revised the introduction. Thank you for the constructive comment.*

o L216-236: A table summarising the treatments and abbreviations used throughout would be very useful here. It is very confusing at the moment and needs to laid out much more clearly.

*Thanks for this constructive and helpful suggestion. We inserted such a table in the supplementary material of the revised version and provided a scheme showing approaches and different measured values in the material and method section (as shown above).*

o L239: The word "Experiment" here is confusing since it is really four different soils, right? It would be better to call the different soils "Soil 1" and so on. Also, why does Soil 4 get more fertiliser added?

*In the revised version the term "Experiment" was changed throughout the text, tables and figures to "Soil". The soil was adjusted to 80% water filled pore space (WFPS) with distilled water. Simultaneously to that the soil was sufficiently fertilized with $NO_3^-$ (varieties $-C_2H_2$, $+C_2H_2$, and traced). Soil 4 that was incubated prior to the other soils was amended with 60 mg N $kg^{-1}$ $NaNO_3$, while in agreement with other experiments conducted in our laboratory, Soil 1, 2 and 3 were amended with 50 mg N $kg^{-1}$ $KNO_3$. In variety traced $NO_3^-$ with a $^{15}N$-labeling of 50 atom% (at%) was used. This information was added to section 2.2.2 (l. 297 ff.).*

o L300: f rather than F would be a more common abbreviation for fraction. Also, this assumes no abiotic denitrification.

*Thank you for this advice. We changed it in the revised version. We also included the information that non-inhibitable organisms contributing to $N_2O$ production and abiotic processes are not included in the assumptions of Eq. 4.*

o S2.5.1 is very hard to follow because of the treatment designations. Again, a table earlier in the methods is needed, much more clearly linking each specific treatment combination to a clear abbreviation code.

*Thank you, as described above we inserted such a table.*

o L322: Product ratio is much too long to be used repeatedly as a variable, maybe just fred or P similar?

*Thank you for this advice. We changed it to $r_{15N}$, $r_{C2H2}$, and $r_{MAP}$ in the revised version.*

o S3.2.2: -C2H2 is basically a control compared to +C2H2 and it seems like logically it should be discussed first.

*You are right. We changed the order of 3.2.1 and 3.2.2 in the revised version.*

o L645: Partial pressure effects would potentially also be expected to affect N2O production, but you saw an increase in N2O production with time?

*Due to the inhibitor application, the incubation time was rather short and we assume that the $N_2O$ production was not largely affected since we found increasing production rates over time. Higher partial pressure in a closed system could result in lower diffusive emissions from the soil (Well et al., 2019). Unfortunately, we did not analyse the partial pressure during incubation. However, a numerical 3-D model for simulating gas diffusion emissions ($N_2O$ and $N_2$) in closed systems showed that denitrification might be underestimated after 6 hours by 30% (Well et al., 2019).*

o L4.2: Also potentially abiotic production.

*In the revised version we included information on this as follows (l. 783 f.): "Additionally, abiotic $N_2O$ production cannot be quantified with the experimental setup, but might be contributing to each inhibitor treatment."*

***Review response on** **"Comparing modified substrate induced respiration with selective inhibition (SIRIN) and N₂O isotope approaches to estimate fungal contribution to denitrification in three arable soils under anoxic conditions"***

*Anonymous Referee #2*
*We thank the reviewer for the critical comments on the structure of the manuscript and for the good comments that largely improved the manuscript.*
*The authors' answers are shown in italics.*

The manuscript submitted here presents an interesting combination of approaches for assessing the contribution of fungal denitrification to the N2O. By using some SIRIN and two isotopic techniques (endmember mixing) and SP/delta18O mapping, they conclude that the fungal contribution to N2O fluxes under anaerobic conditions in the three investigated soils is modest. In general, the manuscript is well written, and the methods are well elaborated. I however miss a clear rationale for the study. As a consequence, the reader is not guided through the work, so that it is hard to get the main conclusions of the work, and how the different pieces of the work fit together (i.e. how the methods compare to each other). Below, you can find some critical parts which should be revisited, with regard to objectives, experimental design and methods and discussion of the results. At the end, some more specific comments.

I think the manuscript would benefit from a more straightforward formulation of the objectives. As it is now, the three objectives (L155-162) are hard to differentiate from each other (ie, using three approaches to determine the fungal contribution to N2O efflux; compare the fungal contribution obtained by the

three, and evaluate the use of SPN2O values); even more importantly, in the abstract I don0t see a connection with such objectives. You can think on hypotheses -e.g. methods (do not) perform equal-, and ways to test them.

> *We revised the abstract and introduction. The abstract was revised by including (l. 29 ff.) "Three approaches were established (modified SIRIN approach, endmember mixing approach (IEM) and the SP/$\delta^{18}$O mapping approach (SP/$\delta^{18}$O Map) to independently investigate the fungal fraction contributing to N$_2$O from denitrification." and (l. 44 ff.) "All three approaches revealed a small fungal contribution to N$_2$O fluxes ($f_{FD}$) under anaerobic conditions in the soils tested. Quantifying the fungal fraction with modified SIRIN was not successful due to issues with inhibitors and pre-incubation effects. In only one soil, $f_{FD}$ of modified SIRIN could be estimated and resulted in 28±9 %, which was possibly overestimated as it was higher than the results obtained by IEM and SP/$\delta^{18}$O Map for this soil ($f_{FD}$ of below 15 and 20 %, respectively). As a consequence of the unsuccessful SIRIN approach, estimation of fungal SP$_{N2O}$ values was impossible."*

> *We included the following hypothesises to the introduction (l. 182 ff.): "We hypothesized that the fungal fraction contributing to N$_2$O from denitrification in different soils using a modified SIRIN approach and isotopic methods will be correlated but not match exactly due to limited inhibitability of microbial communities and variability in SP$_{N2O}$ endmember values. Furthermore, successful application of the modified SIRIN approach with determined fungal fraction contributing to N$_2$O from denitrification using SIRIN will yield fungal SP$_{N2O}$ endmember values within the range of values previously reported in the literature.". Thus, we changed the objectives of the study to (l. 189 ff.): "Therefore, this study aims at (i) determining the fungal contribution to N$_2$O production by denitrification under anoxic conditions and glucose addition using three arable soils and three approaches (modified SIRIN, IEM and the SP/$\delta^{18}$O Map) in order to assess the reliability in soil studies and thus assess factors of potential bias of the methods and (ii) to estimate the SP$_{N2O}$ values from a fungal soil communities and thus to evaluate the transferability of the pure culture range of the fungal SP$_{N2O}$ endmember values."*
>
> *According to this, the Conclusion section was changed to (l. 1055 ff.): "Here, the quantification of the fungal fraction with modified SIRIN could be done with one soil only due to inhibitor issues and was possibly overestimated when compared to the results of isotopic approaches."*

It is also not clear to me the reasoning for the selection of the three soils and how this relates to the objectives, so it is hard to evaluate the suitability of the approach. Is it just to get an idea of variability? In L165 you stated that the soils differ in texture to provide different conditions for denitrification, so that might be the reason behind. The experiments take place under anoxic conditions, so texture might not be that relevant, and one may argue that, for example, different C sources for denitrification maybe more important, or different proportion of fungal vs. bacterial microbial biomass. However, the variability across soils, or the potential role of texture on the results is not discussed at all. Furthermore, one of the soils was sampled twice. Why? What is the difference between experiment 1 and 4? Is it about seasonality? Which kind of information did you want to obtain, and which kind of lessons you learned in hand of the results? I am missing this information in the discussion.

> *Three different soils were assumed to provide various conditions for denitrification and thus also different environments for microbial communities. Thus, the three soils were not selected to analyse effects of land-use type or soil types, but we aimed to compare the different methods (modified SIRIN and isotope approaches) to analyse denitrification. Therefore, three soils harbouring different microbial communities were used to estimate differences in results of the used methods among soils (as described in section 2.1).*
>
> *The three soils differed in texture, but also in C content, C/N ratio and pH (Table 1) and we identified differing microbial biomass values. Thus, we assumed variable community structures and as a consequence, differences in fungal to bacterial abundances were assumed. We thus did not focus on textural effects on denitrification, but aimed to find differences in fungal abundance in soil. We included this as follows (l. 200 ff.): "All experiments were conducted with three arable soils differing in texture, C$_{org}$ content, C/N ratio and pH. There were chosen assuming*

*that the soils harbour different denitrifying communities, i.e., different fractions of bacteria and fungi contributing to denitrification. One of the soils was sampled during a second season to evaluate if the fungal fraction contributing to N₂O production is soil-specific or can be subject to seasonal change of microbial communities."*

As you said, the microbial inhibitors did not have the expected effects. This is evident not only for N2O, but also for CO2. I understand this kind of results are disappointing when investing large efforts in conducting the analysis. But this is a key issue which deserves more attention, since it has important implications for the relevance of the whole study. For example, L609-610 read: "the SIRIN results [: : :] were rather unsatisfactory and led to unsolved questions" (a similar statement at the end of the discussion, L827-830). This is quite a statement which, to be honest, it is not reflected in the abstract, which describes that, for the one soil where it was possible to quantify the fungal contribution, this was 28%, higher than what obtained by the other methods. Thus, as a reader I would infer that SIRIN might overestimate the fungal contribution to the N2O fluxes, which is quite a different conclusion compared to "SIRIN results were rather unsatisfactory". On the contrary, the conclusions have a totally different approach, focusing almost only in the caveats of the SIRIN approach.

> *We agree to clearly state in the manuscript that SIRIN was not successful, because we have the large amount of non-inhibitable production and the only result for Soil 2 is actually very unsure. As mentioned above we changed the abstract as follow (l. 44 ff.): "All three approaches tested revealed a small fungal contribution to N₂O fluxes (f$_{FD}$) under anaerobic conditions in the soils tested. Quantifying the fungal fraction with modified SIRIN was not successful. In only one soil, f$_{FD}$ of modified SIRIN was estimated and resulted in 28±9 %, which was possibly overestimated as results obtained by IEM and SP/δ18O Map for this soil resulted in f$_{FD}$ of below 15 and 20 %, respectively." This made it impossible to calculate SP$_{N2O}$ values for active bacteria or fungi using SIRIN treatments (modified SIRIN B-C).*

By the way, you said you tested the concentrations of inhibitors applied; thus, were the preliminary tests performing better than the "real runs"? How did you test the optimal concentration? In general, when presenting the results of the different methods and discussing them, I missed a profound analysis on which method should be applied, what the cons and pros are and whether methods provide complementary information, which would support the simultaneous use. As it is now, they are presented almost separately, thus failing in in the objective which can be derived from the tile: "comparing : : :approaches to estimate fungal contribution to denitrification : : :"). And I honestly consider this is a serial issue, especially because you are making use of some of the outputs of selected methods as input for the rest of the methods, making all of them dependent to each other. It is only a suggestion, but consider including some table or graph with the main features of each methods and the key info, so that a reader can get an overview at first glance

> *As described in section 2.2.1 the pre-experiments were conducted as described in the original method to analyse F:B ratio by substrate induced respiration with selective inhibition. Unfortunately, only CO₂ production was analysed under oxic conditions and we did not test the optimum conditions under anoxic conditions. Additionally, due to the oxic conditions, N₂O production was not measured in this pre-experiment. We described the differences between this pre-experiment and the incubation experiment presented in more detail in the method section. Regarding the different methods used (modified SIRIN and isotope approaches), we want to emphasize that both isotope approaches, IEM and SP/δ¹⁸O Map, were independent of results of the modified SIRIN approach. While SP$_{N2O}$ values from the acetylated treatment were used in IEM, the SP/δ¹⁸O Map was used to estimate the fungal contribution to N₂O production and N₂O reduction from the non-acetylated treatments.*
>
> *We revised the conclusion section and included (l. 1045 ff.): "Based on the presented results we conclude that the modified SIRIN approach presented here is not appropriate to estimate the contribution of selected communities (bacteria or fungi) on denitrification from soil. Here, the quantification of the fungal fraction with modified SIRIN could be done with one soil only and was possibly overestimated when compared the results of isotopic approaches. Both isotope*

*approaches (IEM and SP/δ¹⁸O Map) revealed similar results of the fungal fraction contributing to denitrification and thus could be recommended as equally suitable for future studies. The present study could show that consideration of N₂O reduction is indispensable. It has to be pointed out, however, that the fungal fraction estimated applies only for the soil under presented experimental conditions, i.e. anaerobic conditions, but not for the investigated soil in general. However, further studies would be needed to cross-validate methods, e. g. with improved inhibitor approaches or molecular-based methods."*

Specific comments:

L48: What do you mean by "under conditions ensuring larger fungal N2O fluxes"?

*When supplying C sources other than glucose, the fungal growth might be greater compared to that with glucose. Apart from that, in a future study one could analyse the microbial community first and by this identify soils with approved high fungal abundance or maybe even a high fungal denitrifier abundance before applying such experiments. We clarified this in the revised text by including "...by added fungal C substrates..." to the sentence in the abstract (l. 59 f.).*

L165-169: A more detailed description of the soil sampled will help. In general, the use of experiment/soil/treatment and variety is confusing, e.g. the same soil is used in two "experiments" (see above) and variety might refer to the use of c2h2 or 15N tracer.

*As described in a comment above the different soils were not chosen to analyse soil factors controlling denitrification, but were chosen to provide variable microbial communities with supposed variations in fungal and bacterial ratios contributing to the denitrifying community. We are sorry for confusion regarding the terms. In the revised version, the term "Experiment" was changed throughout the text, tables and figures to "Soil", treatments with or without C₂H₂ or ¹⁵N tracer were termed 'variety' and we tried to be consistent throughout. A figure was added to aid understanding of the set-up and wording.*

[Figure]

Table 1: inorganic N is expressed in mg/L, which is fine for solutions, but not for de-scribing soils. The usual unit for me is mg N /kg soil. Please check consistence of these numbers.

*Thanks for pointing this out. In the revised version, this was corrected.*

See my comments below, these values can have high temporal variability, with consequences on your analysis. Further to the table: what is "natural soil NO3-"? In general, how stable are these numbers for the Braunschweig soils, and for the rest of the sites? They are arable soils, probably subject to fertilization.

*This is correct. The measurements for Braunschweig soil were performed in samples from 2012. We included this information in the table caption of Table 2 as follows: "Except for NH₄⁺ and NO₃⁻, soil characteristics of loamy sand were only analysed once for samples collected in 2012.". Especially these values will vary within one year in arable soils. However, we amended the soil with C and N, thus changing the current state of the soil before incubation. Although*

*soil properties and microbial community or biomass may have changed over time, we thus assumed that pre-incubating the soil for seven days, applying C and N, and changing the environmental conditions during denitrification induced a rapid growth of specific organisms. Consequently, we were aware that the denitrifying community and the abundance of these organisms in incubation experiments may differ from the community in the field. We expanded the discussion on this point to section 4.4 in the discussion section as follows (l. 967 ff.): "Since environmental conditions may vary within one year in arable soils, soil pH, F:B ratio, or biomass as presented in Table 1 might be different for samples collected in summer 2011. However, as the soil was amended with C and N, the current state of the soil was changed before incubation in any case. Although soil properties, microbial community or biomass may have changed over time, we assumed pre-incubating the soil for seven days, applying C and N, and changing the environmental conditions during denitrification induced a rapid growth of specific organisms. It has to be presumed, that the denitrifying community and the abundance of these organisms in incubation experiments may differ from the community in the field.".*

L243: What is the rationale of having target soil densities? Do they correspond to the field bulk density?
*We did not analyse the bulk density of the tested soils. We repacked the soils according to the expected bulk densities based on texture, i.e. 1.6 g cm$^{-3}$ for a sandy soil and 1.3 g cm$^{-3}$ for a silt loam. We changed the respective sentence in the Material and Methods section as follows (l. 309 ff.):"During packing, the soil density was adjusted to an expected target soil density of 1.6 g cm$^{-3}$ in Soil 1, 2 and 4 and of 1.3 g cm$^{-3}$ in Soil 3 to imitate field conditions."*

L248/L378/Figure 1: How did you calculate the N2O (and CO2) fluxes? Since you flushed with N2, I presume that, for t=0, you used a background concentration of 0 for both N2O and CO2 and then calculated the rate of change after 6, 8 or 10 hours (or 2, 4 and 8 in experiment 4). However, you mentioned average production rates (L378, L390); thus, where does the average come from? As you show in Figure 1, rates vary in some cases by more than 100% depending on the incubation length. So, what is your view on this and, more importantly, what is your suggestion for future experiments?
*We calculated the N$_2$O production rates by averaging the measured N$_2$O production over 6, 8 or 10 hours (or 2, 4 and 8 in experiment 4). As you described, we calculated rates between the time point of flushing with N$_2$ (t=0) and 6, 8 or 10 hours (or 2, 4 and 8 in experiment 4). Thus, we did not calculate the difference in rates between two time points, but averaged over the time point of incubation (0-6, 0-8 or 0-10 hours or 0-2, 0-4 and 0-8 in experiment 4). However, although we calculated these average values, the production rates differed largely between the time points. This was included in section 2.3 as follows (l. 327 ff.): "CO$_2$ and N$_2$O production rates were calculated by averaging the measured N$_2$O production, i.e., between the time point of flushing with N$_2$ (t=0) and six, eight or ten hours (or two, four and eight hours with Soil 4)." As we modified the conditions for microorganisms directly before the incubation and accumulation of N$_2$O started (i.e. mixing the soil, adding water, nitrate, CO$_2$, flushing the headspace with N$_2$) we expected high activity of a large fraction of microorganisms. The antibiotics inhibit protein biosynthesis, and therefore we aimed to increase microbial growth by changing the environment. We explained this in more detail in section 4.1 (l. 735 ff.): "Thus, short-time incubation is recommended when conducting a modified SIRIN approach, as the incubation period should cause changes in conditions for microorganisms and initiate growth on the one hand, while it should avoid the use of inhibitors as C sources by organisms on the other.".*
*The incubation time of the presented study was relatively short (10 hours) compared to other studies focussing on denitrification. However, when using inhibitors it is absolutely necessary to keep incubation time as short as possible to prevent changes in microbial communities due to species development of non-inhibited organisms. This was already described by Anderson & Domsch 1973. The reason for this is that inhibitors can also be used as C sources for microbial growth. This was added to l. 733 ff.:"However, in accordance to Anderson and Domsch (1975) experimental duration should be as short as possible to ensure the CO$_2$ production of initially active organisms." In consequence, we would aim for improved inhibitor effectiveness in future*

*studies, but would recommend relatively short incubation times to avoid that microorganisms could use the inhibitors or dead cells as energy sources.*

L254: How can you have a fixed measurement precision across different incubation lengths? How was this calculated? The precision is some orders of magnitude lower than the calculated fluxes, but there are some large variations within the same treatment. Is this solely due to spatial variability? Or are you presenting only the analytical precision and leaving out some other sources of uncertainty?

> *The precision for GC and IRMS analysis presented here is the analytical precision of measurements derived from analysing laboratory standards of different concentrations. We described this in more detail in the revised version. Variations within treatments derive from spatial variations and replicate incubation that may differ in microbial activity and thus denitrifying activity.*

L289: I am not sure I understood this section. I suspect you used two approaches, but they are presented in a mixed way. With the IEM, one calculates the fungal contribution to N2O by solving the equation 4 using the SP of the N2O produced in the acetylene treatment (variety A) and the assumed SP for fungal and bacteria (33.6 for fungi, and -7.5 to +3.7 per mil for bacteria). The N2O from the acetylene is used to eliminate the distortion coming from N2O reduction in the non-acetylene treatment. But according to L304-307, you solve the equation for SP of fungi using FFD. Please clarify this, since it is highly misleading as it is now.

> *Sorry for the confusion. We tried to clarify by including "The $f_{FD\_SP}$ contributing to $N_2O$ production during denitrification was calculated using the measured $SP_{N2O}$ value from treatment A of variety $+C_2H_2$ as $SP_{prod}$ value (Eq. 4) in equation 4 that was solved for $f_{FD}$ ($f_{FD} = 1-((SP_{prod}-SP_{FD})/(SP_{BD}-SP_{FD}))$). By applying this equation, a range for $f_{FD\_SP}$ is received when using minimum and maximum $SP_{BD}$ values." in the Material and Method section (l. 380 ff.).*

L352: Where did you get the amount of N in unfertilized soils from? Table 1?

> *You are correct, these data are provided in Table 1. We included this information and added also $N_{min}$ data for the samples collected in summer 2011 from loamy sand (Braunschweig). We used the $N_{min}$ data that were analysed directly after sample collection for $N_{soil}$ (amount of N [mg] in unfertilized soil samples).*

L445: How did you assess the success of the acetylene blockage? A rough look to table 2 suggests that is experiment 2 which did not work. In general, this is a fundamental problem, since you don't know the n2o reduction rate a priori (it is precisely the info you want to obtain), unless e.g. application of 15N labelled substrate is combined with N2 isotopic analyses (what you did in section 3.2.3, but I don0t see results for the acetylene treatment, or for N2).

> *Unfortunately, Table 2 does not give information on completeness of blockage since in natural soils the product ratio can vary between the full range of 0 and 1. With $^{15}N$ tracing we did not conduct a treatment with acetylene, but only without acetylene to estimate the $N_2O$ reduction to $N_2$ by analysing $^{15}N$ in $N_2O$ and $N_2$. Nevertheless, we were able to assess the completeness of blockage of $N_2O$ reduction by $C_2H_2$ with the experimental setup. This was done by comparing product ratios calculated from $+C_2H_2$ and $-C_2H_2$ treatments with product ratio calculated from $^{15}N$ treatments (section 3.3 Table 3). To clarify this, we added "It was possible to assess the completeness of blockage of $N_2O$ reduction by $C_2H_2$ with the experimental setup. If $r_{15N}$ and $r_{C2H2}$ were in agreement, a complete blockage of $N_2O$ reduction could be assumed." to l. 396 ff.. Although estimated from parallel incubations, i. e. different incubation vessels, both product ratios ($^{15}N$ and $C_2H_2$) were in similar ranges for Experiment 2, 3, and 4, while only Experiment 1 revealed differences in the two calculated product ratios.*

L450: Significance level was established at 0.1 (L366) so p = 0.037 is significant

> *Thank you for the remark. We changed it in the revised version.*

L461: What are the implications for exp. 4? Was there a significant amount of NO3- available in the soil which may compromise your results? Interestingly, many replicates were not analysed; why? For

those values coming from two replicates, why didn't you include the standard deviation (as you did in Table 2)?

> *The soil of experiment 1 and 4 was investigated two times with the aim to identify differences in fungal and bacterial contribution of denitrification. Unfortunately we did not analyse microbial biomass in 2011 (Soil 4). We added the $N_{min}$ values for samples from summer 2011 and in the experiment $NO_3^-$ was supplied in excess. Additionally, we can assume a homogeneous distribution of $NO_3^-$ added due to the experimental procedure. Thus, $NO_3^-$ supply should not affect fungal or bacterial contribution to denitrification in this setup in any of the soils.*
>
> *Thank you for the remark. We included the standard deviation of the two samples in the revised version and indicated that only two samples were analysed. Some replicates were missing here due to logistical difficulties. This information was added to the Tables.*

L479: you probably mean eq. 8

> *Yes, thanks for spotting this. We corrected this in the revised version.*

Table 4 and 5: Why did you set the negative values to 0 in Table 5, but not in Table 4? Further to table 5: How are the ranges calculated? Are they coming from the different replicates, or from different SP and delta18O, or both?

> *We agree that this was imprecisely described and thus included information on the calculation of ranges in the Table description. Additionally, we uniformly showed calculated negative values for $f_{FD}$, but pointed out, that negative values are non-realistic and therefore discarded for further interpretation.*

Section 4.1 and 4.2 should be better streamlined. Actually, section 4.1 refers almost completely to inhibitors (which is section 4.2)

> *This was done in the revised version by combining section 4.1 and 4.2 and thoroughly editing the text.*

L642: It is not clear to me whether high partial pressure induces physiological changes or rather provokes methodological artifacts (or both). In the former case, respiratory effects might also influence denitrification activity. In the latter case, do you expect an effect on diffusion?

> *Higher partial pressure in a closed system could result in lower diffusive emissions from the soil (Well et al., 2019). Unfortunately, we did not analyse the partial pressure during incubation. The incubation time was rather short and thus we can assume from published values and own experience that $N_2O$ production was not largely affected, which is supported by increasing production rates measured over time. However, a numerical 3-D model for simulating gas diffusion emissions ($N_2O$ and $N_2$) in closed systems showed that denitrification might be underestimated after 6 hours by 30% (Well et al., 2019).*

L653: The role of abiotic processes should be briefly discussed here.

> *Thank you for this remark. Although it is known that abiotic denitrification may occur under the presented conditions it is not possible to quantify $N_2O$ produced from abiotic processes with the used setup. We included the possibility of co-occuring abiotic $N_2O$ production in section 4.1 as follows (l. 783 f.): "Additionally, abiotic $N_2O$ production cannot be quantified with the experimental setup, but might be contributing to each inhibitor treatment."*

L674-678: Experiments 1 and 4 were performed on the same soils, but you got completely different results. And this applies for the mismatch between tracers and acetylene method (only in treatment 1, as you say), but also for the tracers results per se (70-80% N2O production ratios in exp. 1, 50-60% in exp. 4). How do you explain this?

> *Samples were taken at different time points and microbial community may change over seasons during the year. We assume that variations in microbial communities and abundances may be the reasons for the differences in results. As described above we included this information in the discussion section.*
>
> *We already discussed possibilities of variations in microbial groups for differences in pathways between Soil 1 and 4 in section 4.4 as follows (l. 965 ff.): "The question arises, why hybrid $N_2O$*

*formation was only found when the loamy sand was sampled in summer (June, Soil 4) but not when it was sampled during winter (December, Soil 1). Since environmental conditions may vary within one year in arable soils, soil pH, F:B ratio, or biomass as presented in Table 1 might have been different for samples collected in summer 2011. However, as the soil was amended with C and N, the current state of the soil was changed before incubation in any case. Although soil properties, microbial community or biomass may have changed over time, we assumed pre-incubating the soil for seven days, applying C and N, and changing the environmental conditions during denitrification induced a rapid growth of specific organisms. It has to be presumed that the denitrifying community and the abundance of these organisms in incubation experiments may differ from the community in the field. Since these factors were not assessed in our study and their impact on co-denitrification is still poorly understood, it is currently not possible to give an answer here."*

L701: Include the papers you refer to.
    *We included the requested papers: (e. g. Laughlin and Stevens, 2002; Ladan and Jacinthe, 2016; Chen et al., 2014).*

L702-706: The explanation is right, but I suspect you have many uncertainties in the application of inhibitors. Take into account that CO2 release was not affected by the inhibitors, and N2O not as high as expected, so you may have a significant contribution of non-inhibitable organisms, so that the substrate effect on N2O reduction rates may not be that important
    *Thank you, as we discussed this point in section 4.5, we decided to delete this sentence.*

L724-727: The whole sentence is contradictory. Is the SP not an isotopic approach? Please, clarify
    *We apologize for the imprecise description. The sentence was changed as follows (l. 899 ff.):*
    *"Also in many soil incubation studies, $SP_{N2O}$ values (without reduction effects) within the range of bacterial pure cultures have been found (Lewicka-Szczebak et al., 2015; Lewicka-Szczebak et al., 2017; Senbayram et al., 2018).*

*Referee #3*

the named study is a valuable contribution. I support publication.

    *We thank the reviewer for the overall positive critique and for the good comments that will largely improve the manuscript.*
    *The authors' answers are shown in italics.*

Please find my review below:

The study by Rohe et al. is well written and sheds light on a timely topic, namely the role of fungal denitrification in N2O production. Though the results are limited to arable soils, the study has been devised wisely and brings together state-of-the art methods. It is a valuable contribution to the scientific community and well suited for a journal like BG.

I support publication and have only minor comments, which are given below.

Title

ok

Abstract

L41: the term in brackets doesn't add additional information on the mapping approach. I suggest deleting the brackets.

*It is correct, that the term (SP/δ¹⁸O Map) does not add additional information on the approach, but is used as the abbreviation for the SP/δ¹⁸O mapping approach in the following abstract and text. Thus, we decided to introduce this abbreviation in the abstract already as it is commonly used in the literature.*

L45: units unclear. I guess this is the fraction, but for clarity I suggest converting in % at this point.

*As suggested by the reviewer, we converted the quantity of microbial groups from unitless fraction to % in the abstract and in the whole manuscript. The mentioned paragraph (l. 45 ff.) was changed to: "Quantifying the fungal fraction with modified SIRIN was in most cases not successful. In only one soil, $f_{FD}$ of modified SIRIN was 28±9%, which was possibly overestimated as it was higher than the results obtained by IEM and SP/δ¹⁸O Map ($f_{FD}$ of 0 and 20%, respectively)."*

*In line 463, the conversion is addressed as follows: "In the following text all calculated fraction are presented in percent (%)."*

Introduction

L62: describes is inappropriate. I suggest: Denitrification is the stepwise reduction of nitrate to …

*This was changed as suggested.*

L70: sentence is incomplete: I guess it was …performing respiratory denitrification produce substantial amounts of N2O.

*This was changed as suggested.*

L118: I don't agree that the interpretation is more complex. In situations in which oxygen exchange with water is complete, this stabilizes d18O-NO3, since the 18O in water is more stable than in a nitrate pool that is replenished and consumed through nitrification and denitrification. Thus, an assumption of a constant endmember value becomes possible, which has helped immensely with regard to SP 18O-mapping. Please work out that exchange may stabilize 18O-NO3.

*This is correct; when O exchange between soil water and denitrification intermediates is (almost) complete, δ¹⁸O values of remaining $NO_3^-$ can be neglected and δ¹⁸O values of soil water can be used for interpretation of $δ^{18}O_{N2O}$. This was corrected in the revised version and the section was expanded as follows (l. 137 ff.): "Regarding $δ^{18}O_{N2O}$, a complete exchange of oxygen (O) between $NO_3^-$ and soil water can be assumed and consequently, one can use the δ¹⁸O values of soil water for interpretation of $δ^{18}O_{N2O}$ values (Lewicka-Szczebak et al., 2014; Kool et al., 2009; Snider et al., 2009). However, interpretation of $δ^{18}O_{N2O}$ values from different microbial groups may be more complex due to incomplete O exchange:, because variations in the extent of O exchange during denitrification between water and denitrification intermediatesN oxides altersaffect the final $δ^{18}O_{N2O}$ value differently (Garber and Hollocher, 1982; Aerssens et al., 1986; Kool et al., 2007; Rohe et al., 2014b; Rohe et al., 2017)."*

Materials and Methods

L191: the term "substrate induced growth inhibition" is confusing. Please clarify in how far substrate and not inhibitor is responsible for limiting growth.

*This term was corrected: "substrate induced respiration with growth inhibition"*

L282-291: How FFDmi is calculated depends on how well the calculated D of eq 1 and the measured D agree. In other words, at this point of the manuscript, one cannot assess if eq. 3 makes sense because D could be the sum of remaining N2o due to nitrification, uninhibited fungal denitrification, uninhibited bacterial denitrification and abiotic processes. Due to the experimental setup, nitrificatory contributions

and abiotic processes are likely to have little relevance, which is in agreement with the author's notion. But that's also why I don't understand why the denominator is A-D and not A. Why is it more sensible to calculate fungal contribution to denitrification with microbial inhibition compared to inhibitable fungal denitricfication ((A-C)/A)?

*Good point. Eq. 3 is based on the assumption that the fraction of $N_2O$ of treatment D is present in all other treatments as well (A, B, C), representing non-inhibitable sources. Thus calculating A-D as dominator enables to calculate the contribution of $N_2O$ production by bacteria or fungi to the proportion of $N_2O$ from bacteria plus fungi. The method is also based on the assumption that only/mainly bacteria and fungi contribute to $N_2O$ production.*

*As stated in the manuscript, the $N_2O$ production was large in treatment D of all Soils, but this approach is based on the premise that application of both inhibitors (treatment D) leads to a large extent of inhibition. Thus, the validity of Eq. 1 and Eq. 2, just like the requirement that the $N_2O$ production of treatment D is significantly smaller than $N_2O$ production of treatments A, B and C was essential to calculate $f_{FDmi}$. In that case, Eq. 3 could be applied as recommended in the original method. We referred to the original method, although it became clear that $N_2O$ production in treatment D was large. Thus, we decided to state (l. 711 ff): "The modified SIRIN approach was not successful, because large amounts of non-inhibitable $N_2O$ production were observed with all four Soils tested (Table 2, Table 3). The fungal fraction producing $N_2O$ during denitrification ($f_{FDmi}$) was only estimated for Soil 2,"*

L306: Please also give the mean SP_BD value, why is only the range given in contrast to SP_FD?

*Along with the previous point we decided to delete the attempt to calculate $SP_{FD}$ using the SIRIN results by resolving Eq. 4 for $SP_{FD}$ (section 2.5.2) since this yielded biased results. Consequently, section 3.6 ($SP_{N2O}$ values of $N_2O$ produced by the fungal soil community) was deleted and we were also not able to estimate associated $SP_{BD}$ values.*

L318-324: section is unclear what values were fitted? Please clarify

*Here the term "fitted values" describes the $f_{FD}$ contributing to $N_2O$ production from denitrification in soil samples was also estimated with the $SP/\delta^{18}O$ Map ($f_{FD\_MAP}$). We clarified this by including ($f_{FD\_MAP}$) in the respective sentence.*

Results

Section 3.1: Based on Eq. 1 and 2, the production rate in A = B+C-D. This should be used as a quality criterion for the assumptions met. From table 2, it seems that the assumptions made in deriving Eq 1 and 2 were not valid for N2O production rates. I suggest including the term B+C-D in table 2 and present this as result as well.

*As stated above we agreed to clearly state in the manuscript that SIRIN was not successful, because we have the large amount of non-inhibitable production and the only result for Soil 2 is actually very unsure. Thus we would rather clearly state that modified SIRIN was not successful and focus on difficulties in applying such a method and concurrent showing results of isotopic approaches than including other terms to table 2 as you suggested. Section 3.4 (l. 595 ff.) was expanded by "Taking the large ranges of $N_2O$ production rates of each treatment (minimum and maximum values) into account, for each Soil (A-D) was indistinguishable from ((B-D)+(C-D)) (Eq. 2), showing good agreement between Eqs. 1 and 2. However, $N_2O$ production in treatment D was large within all varieties. Only with Soil 2 of the variety $+C_2H_2$, the $N_2O$ production rates of treatment D were significantly smaller than those of the other three treatments."*

Discussion

L632: distribution or community? Please clarify.
*This was corrected to "community".*

L780: please define ap.
*The term ap was already described in the results section in line 1027 f. as follows: "the $^{15}N$ enrichment of the labeled N pool producing $N_2O$".*

L821: larger instead of "smaller than the SPn2o range …
*This section was deleted.*

**Associated editor:**

L. 167: Please provide the information on the sampling dates of the three different soils.
*This information was provided in section 2.1 "Soil 1 with loamy sand sampled in December 2012, Soil 2 with sand sampled in January 2013, Soil 3 with silt loam sampled in December 2012, and Soil 4 with loamy sand sampled in June 2011."*

L. 183: How can the SIR have been determined in summer 2010 and the experiments started in June 2011, if the soil was stored for a maximal duration of 2 months?
*We revised section 2.1. In the revised version we clearly stated that SIR and F:B ratio were analysed early to receive information on the soils tested.*

L. 253: Add "hours" after "two, four and eight".
*Done.*

L. 371: "...when replicates were n < 3.": Please indicate when that was the case.
*We added this information to the text as follows (l. 468 ff.): "For some ANOVAs treatments were excluded when replicates were n < 3. This was the case when only one or two samples out of three replicates could be analysed. This is denoted in the captions of tables (Table 2 and 3)."*

Figure 1/Table 2/Table 3: It appears unlikely that CO2 emissions were of the same order of magnitude (µg C kg-1 h-1) as N2O emissions. I assume it was mg C kg-1 h-1. Please check.
*Thanks for pointing this out. We carefully checked the calculations of $N_2O$ and $CO_2$ production rates and found, indeed, an error in the conversion of units. All $N_2O$ and $CO_2$ production rates must be increased by a factor of 100. We apologize for this error and would like to point out, that values for production rates are much higher in the revised version, but the correction did not affect interpretation of the results. Thus, the values $N_2O$ and $CO_2$ production rates were corrected throughout the revised text and in Figure 1, Table 2 and Table 3. However, the relation between $CO_2$ and $N_2O$ values was not affected by this mistake. The order of magnitude was similar, potentially due to the anaerobic incubation conditions.*

L. 643-646: One important factor for the decrease in CO2 emission rates could have been substrate (glucose) depletion. Please add this point, if you agree.
*We expanded this sentence as suggested: "With incubation time, production rates of $CO_2$ decreased, probably because experimental incubation conditions provoked unfavourable conditions and physiological changes, e.g. anaerobic conditions or local substrate depletion (e. g. C supplied as glucose)."*

L. 791: It would make more sense to start the sentence with: "The content of " instead of "Information on".

*Changed as suggested.*

**Other changes made:**

- *The affiliation of Dominika Lewicka-Szczebak has changed: The affiliation to the University of Göttingen was replaced by "Institute of Geological Sciences, University of Wrocław, pl. M. Borna 9, 50-204 Wrocław, Poland"*
- *Fraction was converted to percent in the whole text*
- *In order to enable a logical order of methods the order of section 2.5.2 and 2.5.3 was changed: Section 2.5.3 (Product ratio [$N_2O/(N_2+N_2O)$] of denitrification) now comes before section 2.5.2 (SP/$\delta^{18}O$ isotope mapping approach (SP/$\delta^{18}O$ Map)).*
- *The order of section 2.5.2 (SP/$\delta^{18}O$ Map) and 2.5.3 (product ratio) was changed to meet a logical order of methods applied.*
- *The discussion section was restructured; merged sections 4.1 and 4.2 as well as 4.5 and 4.7.*
  - *Reference was updated: Lewicka-Szczebak et al., 2020*
    Lewicka-Szczebak, D., Lewicki, M. P., and Well, R.: $N_2O$ isotope approaches for source partitioning of $N_2O$ production and estimation of $N_2O$ reduction – validation with the $^{15}N$ gas-flux method in laboratory and field studies, Biogeosciences, 17, 5513-5537, 10.5194/bg-17-5513-2020, 2020.

**Second round of review:**

Review of bg-2020-285:

*We thank both reviewers for the good comments that again largely improved the revised manuscript.*
*The authors' answers are shown in italics.*
*Most relevant changes in the revised manuscript are marked with a yellow background to highlight those to the previous revised version. We would like to draw your attention to the change of name from Anette Giesemann to Anette Goeske.*

**Reviewer #1**
Re-review of bg-2020-285:
Comparing modified substrate induced respiration with selective inhibition (SIRIN) and N2O isotope approaches to estimate fungal contribution to denitrification in three arable soils under anoxic conditions

This manuscript has benefitted greatly from the first review round, and now it is close to ready for publication, following the comments listed below. The premise and experimental work is good, but the manuscript is often confusingly written and not concise enough.
•Specific comments:

- Abstract: The abstract is confusing to read due to the complex set up and somewhat inconclusive results. There seems to be too much focus on SIRIN, with less focus on the other approaches. I would suggest to rewrite the abstract to (briefly!) answer the two main questions of this study: i) comparing the three approaches, first in method application and then in results, and ii) quantify importance of fungal denit. The current abstract is also perhaps too long; most of the first paragraph is intro and most of the last is conclusions –both should be shortened to just one concise sentence in the abstract.
  *The abstract was carefully revised and also shortened as suggested.*

O  L72:This is confusing –did L&S 2002 think fungal or co denitrification contributed 92%? Did they distinguish the two? The link here to codenitrification is not properly explained. Even if they often

co-occur, surely that doesn't mean codenitrification is an indicator fungal denitrification –co-denit can occur under many conditions without FD also.

*We apologize for the confusion. It was shown by Laughlin & Stevens 2002, that distinguishing between denitrification and co-denitrification was possible when using $^{15}N$ enriched electron acceptors. They could show that 8% of $N_2O$ resulted from denitrification, while 92% of $N_2O$ was produced during co-denitrification. We agree with the reviewer and deleted this section. This process is discussed in the discussion section.*

O Introduction: The introduction is quite hard to follow and still somewhat unclear. I recommend to restructure to something very easy to follow (considering the complexity of the topic), eg.:

▪an introduction to the topic as a whole,

▪followed by a single paragraph about each of the three methods (how the method is applied, examples from prev studies, strengths, risks and weaknesses),

▪followed by a synthesis of comparability of the methods as well as things like acetylene inhibition (it is currently very unclear how this relates to the three methods), and

▪finally a summary of what you hope to achieve, questions, hypotheses. The exact structure is of course up to you but at the moment it is very hard to follow, and for a complex topic such as this a clear intro is very important.

    *The introduction was carefully revised as suggested.*

• L165: This naming is super confusing. The soil sampled twice should be called 1.1 and 1.2, not 1 and 4, to clarify throughout that these are "more similar" that soils 2 and 3.

    *This was changed in Method section and changed throughout the manuscript. "As one this soil was sampled at two different time points, we conducted four experiments and named the different experiments "Soil 1.1", "Soil 1.2", "Soil 2", and "Soil 3" (section 2.1; l. 278 ff.): Soil 1.1 and Soil 1.2 with loamy sand sampled in December 2012 and in June 2011 respectively, Soil 2 with sand sampled in January 2013, and Soil 3 with silt loam sampled in December 2012 (Table 1)."*

• Figure 1: This table is an improvement but still extremely confusing. Why is acetylene only mentioned at the very right hand side? The formatting in the second-from-right boxes is really odd. The use of acronyms (esp in the second-from-right boxes) means the table is not a good overview for a reader who has not yet finished reading the paper. The linkages eg. between expt varieties is odd. This figure still needs a lot of work –it should function as an overview to guide a reader into the experiments and clarify exactly what was applied to which soils in which order. Using colour would probably help to separate different aspects.

    *Figure 1 has been revised and the workflow is now clearer represented and we hope that the reader can better follow this revised scheme. The figure caption has also been revised to better describe the experiments.*

[Figure]

***Figure 1: The methodical approach comprised a pre-experiment with substrate induced respiration (SIR) to estimate the optimal glucose concentration ($c_{opt}$(glucose) and the fungal-to-bacterial ration in the soil (F:B ratio), and the substrate induced respiration with selective inhibition approach (SIRIN) to determine the optimal inhibitor concentration ($c_{opt}$(streptomycin and $c_{opt}$(cycloheximide)). The initial soil status, i.e. ammonium and nitrate concentration of the soil ($c(NH_4^+)$ and $c(NO_3^-)$), respectively), was measured in $N_{min}$ extracts***

*and the isotopic signature of soil NO₃⁻ was analysed by the denitrifier method. The incubation experiment comprised the SIRIN approach with three experimental varieties: without acetylene (-$C_2H_2$), with $C_2H_2$ (+$C_2H_2$), and without $C_2H_2$ but with $^{15}N$ labelled NO₃⁻ (traced), while NO₃⁻ with natural isotopic composition was added to the other two varieties. Produced gas was analysed for its concentration (c($CO_2$) and c($N_2O$)) using gas chromatography (GC) and $N_2O$ was further analysed by isotope ratio mass spectrometry (IRMS) for its isotopic composition. Please refer to the Material & Methods section for more information.*

- L201-216: These results should be shown in a figure in the SI.
    *Respiration curves of representative examples were inserted in the SI.*

O L216: Is this a valid assumption, considering these are pretty different processes? What are the implications if conditions are not optimal for denitrification?
    *In our opinion this was the option to determine optimal concentrations. Growth is inhibited when sufficient concentrations are added to the soil. However, as stated in the manuscript, this modified SIRIN approach should be systematically assessed and checked if it is transferable to experiments under anaerobic and aerobic conditions to determine denitrification.*

- L250: Why more fertiliser for soil 4? And different sample timing (L264)?
    *The only reason for more NO₃⁻ with Soil 4 (new numbering: 1.2) than with the other three soils was the timing of experiments and development of our lab. For comparison reasons with other experiments in our working group we decided to change the concentration of NO₃⁻ in soil incubation experiments and thus we used 50 mg later instead of 60 mg N. Nevertheless, we decided to show the first results too, as this one soil was incubated twice (summer and wither time).*

- L307: I understand from your response to the initial review that A-D is conventionally used as the denominator in this equation, ie. you determine FD relative to FD + inhibitable BD, rather than as a proportion of total denitrification. All the same, I find that using just A as a denominator and calculating FD as a proportion of total denit would be more appropriate in this case, as it is more comparable to the isotope approaches, which are essentially finding FD as a proportion of total denit. I think it would be good to calculate f-FD with A as a denominator in addition to the more traditional A-D. It is well within the scope of this paper to try to improve these methods rather than just apply them as previously done.
    *With this approach a successful inhibition of gas fluxes of target groups is absolutely required. This includes treatment B and C, but also D, while only treatment D gives an indication of completeness of inhibition. In our opinion, this condition is the most important prerequisite of this method. The suggested calculation of $f_{FD}$ is not valid in our view, because we did not achieve satisfying inhibition with both inhibitors. We clearly have to state that SIRIN was not successful, because large $N_2O$ production was found with both inhibitors. Since we know that the inhibition of the presented study was not successful, it does not seem justifiable to us to relate $N_2O$ production of treatments B and/or C to the $N_2O$ production of treatment A. We agree that with successful inhibition, the proposed approach should be performed in addition to the original calculation.*

- S2.5.1: I think you could consider an addition to this section, which would allow you to also use the non-acetylene experiments with the IEM approach. FD, nitrification, and reduction all act to increase SP. Only denitrification (bacterial, codenit, nitrifier denit...) produce low SP N2O. Therefore, you can calculate a "minimum" denitrification contribution, and thus a "maximum" FD contribution. If you measure, for example, SP = 2 permil, with endmembers of 0 and 33 permil for BD and FD/nitrif (this is just a quick example, of course you want to consider uncertainties!), using Eq. 4 you get a minimum denitrification contribution of (2-0)/(33-0)=94% and thus a maximum FD of 6%. This approach gives a relatively strong constraint on fungal denitrification when you measure a low SP (can't have much FD) and very little or no constraint on FD when you measure a high SP.
    *It is true that this IEM cannot differentiate between $N_2O$ from fungal denitrification or nitrification. We have implemented your suggested calculation and expanded section 2.5.1 by (l. 464 ff.): "Based on $SP_{N2O}$ values from -$C_2H_2$ variety, it was possible to solve Eq. 4 also to estimate the maximum potential fungal contribution to denitrification ($f_{FD\_SPpot}$) assuming that we did not have any estimations for $N_2O$ reduction. While bacterial denitrification and nitrifier denitrification would result in low $SP_{N2O}$ values ($SP_{BD/ND}$=-10.7 to +3.7 ‰ (Frame and*

*Casciotti, 2010; Yu et al., 2020)), large $SP_{N2O}$ values would be expected from fungal denitrification and nitrification ($SP_{FB/N}=16$ –to 37 ‰ (Sutka et al., 2008; Decock and Six, 2013; Rohe et al., 2014a; Maeda et al., 2015; Rohe et al., 2017)). $N_2O$ reduction could have further increased the $SP_{prod}$ values. If the contribution of this process on SPprod values cannot be precisely estimated, by neglecting these effects we can determine the maximal potential fungal contribution. $f_{FD}$ calculated from Eq. 4 (variety -$C_2H_2$) would thus be lower if $N_2O$ reduction had occurred. However, assuming the impact of $N_2O$ reduction on $SP_{N2O}$ was negligible, this IEM enabled to calculate the maximum potential $f_{FD}$ as $f_{FD\_SPpot} = 1-((SP_{N2O}-SP_{FD/N})/(SP_{BD/ND}-SP_{FD/N}))$.”*

*Resulting ranges for this calculated measure were inserted to Table 5.*

o In S2.5.1 I'm also unsure why you completely discount nitrification. I know you have wettish soils and flush with N2but potentially there is still aerobic microsites, oxygen production in the soils from other processes, ... Do you have an estimate or previous study to show nitrification is truly negligible?

*Unfortunately, we did not perform pre-experiments to estimate the contribution of nitrification. However, with the $^{15}N$ approach we could show that $N_2O$ with Soil 1.1, Soil 2 and Soil 3 was produced from $^{15}N$-labelled $NO_3$. In case of nitrification, $^{15}N_{N2O}$ values in variety traced would be expected much smaller than measured values. As mentioned in Section 3.2.3 "Soil 1.2 is the only one showing a large discrepancy between measured (about 30 at%) and calculated $^{15}N_{N2O\_exp}$ (49 at%) in $N_2O$, whereas the other Soils showed close agreement (Table 3)". In section 4.4 it was discussed in detail that the discrepancy between measured and calculated $^{15}N_{N2O}$ values did not derive from dilution effects from labelled and unlabelled $^{15}N$ that could have been produced during nitrification.*

• L386: Why 0.1? This is a high significance level. And at L405, why do you sometimes report the P (eg. 0.002, 0.008, 0.027) and sometimes just the comparison to the sig level (P < 0.010)? It would be better to always report the P, unless it is very small, eg. P<0.001.

*We have standardized this in the revised version and used the significance level of 0.05.*

oTable 4: I would think that the errors causing negative values would also apply to positive values, eg. this simply reflects a high uncertainty of the mapping approach. Simply rejecting negative values will therefore bias any interpretations or aggregations towards an overestimation of fungal denitrification using this technique.

• This is also reflective of the fact that the mapping approach is extremely uncertain as your points are mostly very close together, providing little constraint on the gradient. The mapping approach would presumably be better suited only to samples with larger ranges (a couple of yours have more range but most not).

*The observed errors are mainly due to large ranges of possible endmember values (e.g. $SP_{BD}$). We do not know which values of the wide range are valid for our soils, so we calculated the scenarios with different endmember values covering the whole possible variations. For some values we got negative results, which indicates that the applied endmember value is not valid in that case. This does not mean that our positive results were free of any extra errors. Similarly, this may be due to various water isotopic signatures, that's why we indicate with bold font in Table 4 the most plausible results based on the most probable soil water isotopic signatures - these values were chosen without taking into account if the results are positive or negative. And we can see that the negative results are associated with implausible values for water isotopes (which was determined in our fitting procedure), hence these values should be anyway rejected - not because the results are negative, but because the implausible water isotope values indicate that the assumed $SP_{BD}$ value is for this case not valid.*

*However, it is of course true that for these low fungal contributions the method is uncertain - meaning the precision of $f_{FD}$ determination, however it is quite certain that this fungal contribution is not larger than the indicated range, which is emphasised in the text (section 3.5, l. 723 ff.): „The results obtained from SP/$\delta^{18}O$ Map show $f_{FD\_MAP}$ reaching up to 14, 20, 15, and 9 % for Soils 1.1, 1.2, 2, and 3, respectively (Figure 3, Table 4, Table 5)."*

• L605-619: SIRIN could be an overestimation simply by definition, as it does not include non-inhibitable in the definition, and FD-MAP could be an overestimation because of the rejection of negative values –so I would say the endmember approach gives the highest confidence.

*The methods are uncertain for a variety of reasons that are discussed in the Material & Methods and Discussion section. However, one must note that all methods rely on different assumptions, i.e. SIRIN assumes activity of non-target organisms is unaffected by inhibitors, and isotopic approaches rely on endmember signatures derived from pure culture experiments. Nevertheless, all successful methods presented indicated a bacterial dominance in $N_2O$ production over fungal $N_2O$ production.*

- S4.4: You have very little evidence at all relating to codenitrification and I think it could be more appropriate to remove this section, or at the very least reduce it to a short paragraph at the end of a different section.

  *In our opinion, this section is of great relevance for this manuscript as it highlights possibilities of co-occuring processes, which is confirmed by the fact of deviations of $^{15}N$-$N_2O$ and $^{15}N$-$NO_3$. We could show that hybrid $N_2O$ was produced and assumed production by co-denitrification. However, it is true that we cannot completely exclude any other processes. Therefore we changed the subtitle to "Potential influence of hybrid $N_2O$" and also revised and restructured this section 4.4. Now this section in less focussing on co-denitrification but still refers to its potential hybrid $N_2O$ formation.*

- Discussion: The discussion is much improved from the previous version, and provides a good overview now. The authors could still reread the discussion and ensure (like the introduction) that it is a clear and concise coverage of the results in relation to the research questions and hypotheses.

  *The discussion section was partly revised, two subtitles were changed.*

•Minor comments:
- L99: change to ...interferes with quantification of FD based on SP values...or something similar.

  *Changed as requested.*

- oL480: Change than of to than of

  *Changed as requested.*

- oS4.2: Should the title be with and not without?

  *The title was changed to "How important in $C_2H_2$ application to examine the fungal contribution to $N_2O$ production in soil? " in order to prevent the impression that SIRIN was successful. We rather discuss the importance of $C_2H_2$ application in soil incubation experiments to estimate $f_{FD}$.*

- L744: I think it would be more appropriate to change "revealed" to "were consistent with". Also, this sentence is long and a bit repetitive and could be improved.

  *Changed as requested.*

**Reviewer3:**

The revised version of this manuscript contains an improved description of the methodical approach, and the clear message that the application of the modified SIRIN method was not successful in this case. In summary, the manuscript has improved, and, as previously, I support publication if the following aspects are considered:

The first aspect relates to my previous comment on section 2.5.3 of the revised manuscript. Though this section is more clear now, there is a significant caveat for the procedure of fitting d18O-H2O values: This fitting involves the utilization of the product ratio r15N (eq.5). As a consequence, r15N and rMAP are not independent quantities any more, and this should also be indicated and discussed. To resolve this dependence between r15N and rMAP, the authors should (additionally, if you wish) make an assumption on the d18O-H2O value that makes the SP/18O mapping approach independent of the traced approach (usually, these values are between -12 to -7 per mil.).

*It is correct that the product ratio determined with the SP/$^{18}O$ mapping approach ($r_{MAP}$) strictly depends on the product ratio determined with the $^{15}N$ treatment ($r_{15N}$). Consequently, the determined $f_{FD\_MAP}$ value is based on both $^{15}N$ treatment and natural abundance isotopes SP and $\delta^{18}O$. This makes this value even more reliable than $f_{FD\_SP}$. We could assume a $\delta^{18}O_{H2O}$ value,*

*however taking the range from -12 to -7 ‰ will result in significant variations. The values which we indicate as the most plausible (Table 4, bolded values) are actually in a similar range from -11 to -6 ‰. If we just assume one value without knowing it, the results may be biased. There is no need in this study for independence of the values for $r_{15N}$ and $r_{MAP}$, because the aim is to compare $f_{FD\_MAP}$, $f_{FD\_SP}$ and $f_{FD\_SIRIN}$ (Table 5). We have added the information that the determined $f_{FD\_MAP}$ in this case also depends on $r_{15N}$ results in Section 3.5 (l. 725 ff.):*
*"Importantly, due to the fitting procedure applied the estimations of $f_{FD\_MAP}$ values are based not only on $SP_{N2O}$ and $\delta^{18}O_{N2O}$ values but also on the results obtained in the $^{15}N$ treatment ($r_{15N}$ values)."*
*This information was also added to the footnotes to Table 5:*
*"[d]Fungal fraction of $N_2O$ production calculated by $SP/\delta^{18}O$ Map with assuming most probable $SP_{N2O}$ values from bacterial denitrification (according to Table 4). Using the minimum and maximum $SP_{N2O}$ values known for bacteria and ranges of fitted $\delta^{18}O_{H2O}$ values (the fitting is based also on results obtained in $^{15}N$ treatment) resulted in a $f_{FD\_MAP}$ range. "*

The second aspect is that in the context of the very clear communication that the modified SIRIN approach was not successful, section 4.2 could be shortened significantly as it now reads as a rather hypothetical discussion of a situation in which the inhibition of bacteria and fungi had worked. At the moment it sounds like SIRIN still is an option for the authors, but this collides with the authors' conclusions. Please elaborate on your opinion here because it is not clear whether or not you still advocate the SIRIN approach.

*The subtitle was changed to "Is $C_2H_2$ application a suitable and necessary treatment for examining the fungal contribution to $N_2O$ production in soil? " in order not to leave the impression that SIRIN was successful. We revised this section to rather discuss the importance of $C_2H_2$ application in soil incubation experiments to estimate $f_{FD}$ and pointed out, why this is of particular importance also for other studies, i.e when SIRIN treatments effectively inhibit the selective groups. In this section we clearly state that the SIRIN approach presented did not work. In the conclusion, we summarize that additional work on the methods is needed.*

**Comments to the Associate Editor:**

*We took great care to revise the manuscript as you suggested with your comments (.pdf file). These changes relate besides some other corrections to the following points:*

- *changing "variety" and "varieties" to "variant" and "variants"*
- *changing the name of soil taken in June 2011 to "Soil 1.1", and the name of soil taken in December 2012 to "Soil 1.2"*
- *section 2.5.1:"assuming that we did not have any estimations for $N_2O$ reduction" was changed to "assuming that there was no contribution of $N_2O$ reduction".*